# FEATURE AVERAGING: AN IMPLICIT BIAS OF GRADIENT DESCENT LEADING TO NON-ROBUSTNESS IN NEURAL NETWORKS

**Binghui Li**[1,*]  **Zhixuan Pan**[2,*]  **Kaifeng Lyu**[3]  **Jian Li**[2,†]
[1]Center for Machine Learning Research, Peking University
[2]Institute for Interdisciplinary Information Sciences, Tsinghua University
[3]Simons Institute, UC Berkeley
`libinghui@pku.edu.cn,  pzx20@mails.tsinghua.edu.cn`
`kaifenglyu@berkeley.edu,  lapordge@gmail.com`

## ABSTRACT

In this work, we investigate a particular implicit bias in gradient descent training, which we term "Feature Averaging," and argue that it is one of the principal factors contributing to the non-robustness of deep neural networks. We show that, even when multiple discriminative features are present in the input data, neural networks trained by gradient descent tend to rely on an average (or a certain combination) of these features for classification, rather than distinguishing and leveraging each feature individually. Specifically, we provide a detailed theoretical analysis of the training dynamics of two-layer ReLU networks on a binary classification task, where the data distribution consists of multiple clusters with mutually orthogonal centers. We rigorously prove that gradient descent biases the network towards feature averaging, where the weights of each hidden neuron represent an average of the cluster centers (each corresponding to a distinct feature), thereby making the network vulnerable to input perturbations aligned with the negative direction of the averaged features. On the positive side, we demonstrate that this vulnerability can be mitigated through more granular supervision. In particular, we prove that a two-layer ReLU network can achieve optimal robustness when trained to classify individual features rather than merely the original binary classes. Finally, we validate our theoretical findings with experiments on synthetic datasets, MNIST, and CIFAR-10, and confirm the prevalence of feature averaging and its impact on adversarial robustness. We hope these theoretical and empirical insights deepen the understanding of how gradient descent shapes feature learning and adversarial robustness, and how more detailed supervision can enhance robustness.

## 1 INTRODUCTION

Deep learning has achieved unprecedented success across a wide range of application domains, including many safety-critical systems such as autonomous driving and diagnostic assistance technologies. Despite these successes, a landmark study by Szegedy et al. (2013) exposed that deep neural networks are extremely vulnerable to adversarial attacks. These attacks involve adding nearly imperceptible and carefully chosen perturbations to input data to confound deep learning models into making incorrect predictions. The perturbed inputs are termed adversarial examples, and their existence has attracted significant attention from the research community. Since then, various attacks (Biggio et al., 2013; Szegedy et al., 2013; Goodfellow et al., 2014; Madry et al., 2018) and defenses (Goodfellow et al., 2014; Madry et al., 2018; Shafahi et al., 2019; Pang et al., 2022) were developed, but the issue of adversarial robustness is still far from being resolved.

Previous attempts to explain the adversarial robustness of neural networks have been made from various theoretical perspectives. Daniely & Shacham (2020); Bubeck et al. (2021a); Bartlett et al.

---

[*]Equal contribution, alphabet ordering.
[†]Corresponding author.

(2021); Montanari & Wu (2023) proved the existence of adversarial examples for neural networks with random weights across various architectures. Tsipras et al. (2019); Zhang et al. (2019) analyzed the fundamental trade-off between robustness and accuracy. Bubeck et al. (2021b); Bubeck & Sellke (2021); Li et al. (2022a); Li & Li (2023) proved that having a large model size is necessary for achieving robustness in many settings. Ilyas et al. (2019); Tsilivis & Kempe (2022); Kumano et al. (2024); Li & Li (2024) studied the relationship between adversarial examples and the presence of non-robust but predictive features in the data distribution.

A related line of work in deep learning theory studies the implicit bias of gradient descent to explain why neural networks generalize so well. Training deep neural networks is a highly non-convex and over-parametrized optimization problem, in which there are many solutions that fit the training data correctly. Recent studies suggest that, without explicit regularization, gradient descent seems to implicitly bias towards solutions that enjoy favorable properties, particularly good generalization. Hence, characterizing various implicit biases in favor of better generalization has been extensively studied in recent years (Gunasekar et al., 2017; Soudry et al., 2018; Arora et al., 2019b; Lyu & Li, 2020; Blanc et al., 2020). However, good generalization properties do not necessarily imply good robustness with respect to inputs. Indeed, even well-trained neural networks are vulnerable to adversarial examples. In fact, recent studies by Vardi et al. (2022) and Frei et al. (2024) proved that the implicit bias of gradient descent can be a "double-edged sword", in the sense that it leads to generalizable solutions with perfect clean accuracy, but being non-robust (susceptible to small adversarial $\ell_2$-perturbations), even though there exist robust networks with perfect robust accuracy. Under a similar data setup, Min & Vidal (2024) further conjectured that the weight vectors of a two-layer ReLU network trained by gradient flow converge to an average of the cluster centers.

In this paper, we perform a detailed analysis of the training dynamics of gradient descent on two-layer ReLU networks (under data distributions similar to Vardi et al. (2022), Frei et al. (2024) and Min & Vidal (2024), and the detailed discussion about the connection between their works and our paper is deferred to Section 2), and rigorously prove that the learned weights exhibit a particular implicit bias, which we term *feature averaging*. In our setting, feature averaging refers to a particularly simple form: the network trained by gradient descent tends to learn the average of useful features, in the sense that the weight vector associated with each hidden-layer neuron is a weighted average of feature vectors. This also resolves the conjecture made by Min & Vidal (2024). Further, one can easily show that such an average is more susceptible to small adversarial perturbations than individual features, rendering the learned solution non-robust.

In our experiments, we empirically observe similar phenomena in several other settings. We argue that feature averaging is a key factor contributing to the non-robustness of deep neural networks, and demonstrate its close relationship with several known phenomena and theoretical models in adversarial robustness research. These include the observation that neural networks tend to leverage both robust and non-robust features for classification (Tsipras et al., 2019; Ilyas et al., 2019; Allen-Zhu & Li, 2022; Tsilivis & Kempe, 2022; Li & Li, 2024), the connection between model Lipschitzness (smoothness) and over-parameterization (Bubeck et al., 2021b; Bubeck & Sellke, 2021; Li et al., 2022a; Li & Li, 2023), the simplicity bias of gradient descent that leads to non-robustness (Shah et al., 2020; Lyu et al., 2021), and the dimpled manifold hypothesis (Shamir et al., 2021). Beyond the linear average behavior studied in this work, we conjecture that feature averaging may appear in more complex forms in real-world settings. For example, the neural network may tend to combine many localized, semantically meaningful (hence more robust (Ilyas et al., 2019; Tsilivis & Kempe, 2022)) features into one discriminative but non-robust feature.

In light of the feature averaging phenomena, we propose to enhance the robustness by learning individual features. In particular, we explore a natural and simple, yet less explored method in the study of adversarial robustness, which is to provide more granular supervised information related to individual features and force the model to learn the individual features. Theoretically, we prove that training a two-layer ReLU network with feature-level labels leads to a binary classifier with optimal robustness. Empirically, we design several experiments, using synthetic and real datasets, and the experimental results demonstrate that feature-level supervised information can be very effective in enhancing the robustness of the model (even with standard training). These results are consistent with the empirical findings by Sitawarin et al. (2022); Li et al. (2024), which showed that incorporating fine-grained annotations, such as part-level segmentation, can substantially enhance the adversarial robustness of object recognition systems. See Appendix A for a more detailed discussion of these

connections, including the relationship of feature averaging to existing robustness phenomena, and how more granular supervision may improve adversarial robustness.

Our technical contributions can be summarized as follows:

1. (Section 4.1) Under certain multi-cluster data distributions (similar to that in Frei et al. (2024)), we prove that two-layer ReLU networks trained by gradient descent converge to feature-averaging solutions (Theorem 4.5). In particular, we show that the weight vector associated with each hidden-layer neuron converges to the average of cluster-center features and the feature-averaging solution is non-robust w.r.t. the radius $\Omega(\sqrt{d/k})$, while there exist solutions with optimal robust radius $O(\sqrt{d})$ (where $d$ is the data dimension, $k$ is the number of clusters, and the existence of such optimal robust solutions is shown in Theorem G.3). This result also solves the conjecture of Min & Vidal (2024) under our settings (Theorem 4.6).

2. (Section 4.2) We show that if the model is provided with the feature level labels (in fact a multi-class classification problem in our multi-cluster data distribution setting), a two-layer network can learn the individual features, which furthermore can induce a robust model with optimal robust radius $O(\sqrt{d})$ (Theorem 4.7).

3. (Section 5) We validate our theoretical results on synthetic data and real-world datasets such as MNIST and CIFAR-10. We empirically show that gradient descent learns averaged features. Our experiments also demonstrate enhanced robustness through the incorporation of fine-grained supervisory information.

## 2 RELATED WORK

**Implicit Bias of Gradient Descent.** The implicit bias of gradient descent has been studied from various perspectives. The most prominent line of works establishes an equivalence between neural networks in certain training regimes to kernel regression with Neural Tangent Kernel (NTK) (Du et al., 2019b;a; Allen-Zhu et al., 2019a; Zou et al., 2020; Chizat et al., 2019; Arora et al., 2019b; Ji & Telgarsky, 2020b; Cao & Gu, 2019), but the generalization of kernel regression is usually worse than that of real-world neural networks. Other works prove other types of implicit biases beyond this NTK regime, including margin maximization (Soudry et al., 2018; Nacson et al., 2019; Lyu & Li, 2020; Ji & Telgarsky, 2020a), parameter norm minimization (Gunasekar et al., 2017; 2018; Arora et al., 2019a) and sharpness reduction (Blanc et al., 2020; Damian et al., 2021; HaoChen et al., 2021; Li et al., 2022b; Lyu et al., 2022; Gu et al., 2023). All these works focus on implicit biases that may lead to good generalization except that Vardi et al. (2022) and Frei et al. (2024) connected the line of works on margin to the non-robustness of neural networks, which we discuss shortly.

**Feature Learning Theory for Two-Layer Networks.** The feature learning theory of two-layer neural networks as proposed in various recent studies (Wen & Li, 2021; Allen-Zhu & Li, 2022; Chen et al., 2022; Cao et al., 2022; Zhou et al., 2022; Chidambaram et al., 2023; Allen-Zhu & Li, 2023; Kou et al., 2023a; Simsek et al., 2023) aims to explore how features are learned in deep learning. This theory extends the theoretical optimization analysis beyond the scope of the neural tangent kernel (NTK) theory (Jacot et al., 2018; Du et al., 2019b;a; Allen-Zhu et al., 2019b; Arora et al., 2019b). Among these feature learning works, there exist various data assumptions about feature-noise structure. Based on the data assumption of sparse coding model, Wen & Li (2021) study feature learning process of self-supervised contrastive learning, and Allen-Zhu & Li (2022) propose a principle called feature purification to explain the workings of adversarial training. Allen-Zhu & Li (2023) utilize multi-view-based patch-structured data assumption to understand the benefits of ensembles in deep learning. Following the multi-view data proposed in Allen-Zhu & Li (2023), Chidambaram et al. (2023) show that data mix-up algorithm can provably learn diverse features to improve generalization. Cao et al. (2022); Kou et al. (2023a) explore the benign overfitting phenomenon of two-layer convolutional neural networks by leveraging a technique of signal-noise decomposition. Zhou et al. (2022) study feature condensation and prove that, for two-layer network with small initialization, input weights of hidden neurons condense onto isolated orientations at the initial training stage. Simsek et al. (2023) focus on the regression setting and study the compression of the teacher network, and they find that weight vectors, whether copying an individual teacher vector or averaging a set of teacher vectors, are critical points of the loss function.

**Comparisons with Vardi et al. (2022), Frei et al. (2024) and Min & Vidal (2024).** Recently, Vardi et al. (2022) and Frei et al. (2024) demonstrated that for two-layer ReLU networks, any KKT solution to the maximum margin program (it is known that gradient flow converges to such KKT solution (Lyu & Li, 2020; Ji & Telgarsky, 2020a)) leads to non-robust solutions under the assumption of synthetic cluster data, and Min & Vidal (2024) further conjectured that the weight vectors of two-layer ReLU network converge to an average of cluster-center vectors. Their finding highlights the significance of the optimization process in the (non)robustness of neural networks. Our theoretical results are inspired by theirs, but differ from theirs in the following important aspects: (1) Conceptually, feature averaging is arguably more intuitive and concrete (in the feature level) than the set of KKT properties. Moreover, feature averaging (or its nonlinear extensions) may appear in more complex and general setting even when the solution is far from a KKT point. (2) Technically, we perform a detailed and finite-time analysis of the gradient descent dynamics, in contrast to their result about limiting behavior of gradient descent. In particular, our analysis of gradient descent dynamics reveals the feature learning process. Furthermore, we comment that the time complexity converging from an initialization point to a KKT solution can be slow (i.e., $\Omega(1/\log(t))$ proven in Soudry et al. (2018); Lyu & Li (2020); Kou et al. (2023b)). (3) Our analysis of the GD dynamics requires small initialization, whereas their results depend on starting from a solution that already correctly classifies the training set (an assumption made in (Lyu & Li, 2020) for achieving KKT points). (4) Our result (Theorem 4.5) solves the conjecture proposed by Min & Vidal (2024), where we show that the weight vector associated with each neuron aligns with a weighted average of cluster features, and the ratio between weights of distinct clusters is close to 1.

## 3 PROBLEM SETUP

In this section, we introduce some useful notations and concepts, including the multi-cluster data distribution, the two-layer neural network learner and the gradient descent algorithm.

**Notations.** We use bold-face letters to denote vectors, e.g., $\boldsymbol{x} = (x_1, \ldots, x_d)$. For $\boldsymbol{x} \in \mathbb{R}^d$, we denote by $\|\boldsymbol{x}\|$ the Euclidean ($\ell_2$) norm. We denote by $\mathbb{1}(\cdot)$ the standard indicator function. We denote $\mathrm{sgn}(z) = 1$ if $z > 0$ and $-1$ otherwise. For integer $n \geq 1$, we denote $[n] = \{1, \ldots, n\}$. We denote by $\mathcal{N}\left(\mu, \sigma^2\right)$ the normal distribution with mean $\mu \in \mathbb{R}$ and variance $\sigma^2$, and by $\mathcal{N}(\boldsymbol{\mu}, \boldsymbol{\Sigma})$ the multivariate normal distribution with mean vector $\boldsymbol{\mu}$ and covariance matrix $\boldsymbol{\Sigma}$. The identity matrix of size $d$ is denoted by $\boldsymbol{I}_d$. We use $\mathrm{Unif}(A)$ to denote the uniform distribution on the support set $A$. We use standard asymptotic notation $O(\cdot)$ and $\Omega(\cdot)$ to hide constant factors, and $\tilde{O}(\cdot), \tilde{\Omega}(\cdot)$ to hide logarithmic factors.

### 3.1 DATA DISTRIBUTION

Following Vardi et al. (2022); Frei et al. (2024), we consider binary classification on the following data distribution with multiple clusters.

**Definition 3.1** (Multi-Cluster Data Distribution). Given $k$ vectors $\boldsymbol{\mu}_1, \ldots, \boldsymbol{\mu}_k \in \mathbb{R}^d$, called the *cluster features*, and a partition of $[k]$ into two disjoint sets $J_\pm = (J_+, J_-)$, we define $\mathcal{D}(\{\boldsymbol{\mu}_j\}_{j=1}^k, J_\pm)$ as a data distribution on $\mathbb{R}^d \times \{-1, 1\}$, where each data point $(\boldsymbol{x}, y)$ is generated as follows:

1. Draw a cluster index as $j \sim \mathrm{Unif}([k])$;

2. Set $y = +1$ if $j \in J_+$; otherwise $j \in J_-$ and set $y = -1$;

3. Draw $\boldsymbol{x} := \boldsymbol{\mu}_j + \boldsymbol{\xi}$, where $\boldsymbol{\xi} \sim \mathcal{N}(\boldsymbol{0}, \boldsymbol{I}_d)$.

For convenience, we write $\mathcal{D}$ instead of $\mathcal{D}(\{\boldsymbol{\mu}_j\}_{j=1}^k, J_\pm)$ if $\{\boldsymbol{\mu}_j\}_{j=1}^k$ and $J_\pm$ are clear from the context. For $s \in \{\pm 1\}$, we write $J_s$ to denote $J_+$ if $s = +1$ and $J_-$ if $s = -1$.

To ease the analysis, we make the following simplifying assumptions on the distribution.

**Assumption 3.2** (Orthogonal Equinorm Cluster Features). The cluster features $\{\boldsymbol{\mu}_j\}_{j=1}^k$ satisfy the properties that (1) $\|\boldsymbol{\mu}_j\| = \sqrt{d}$ for all $j \in [k]$; and (2) $\boldsymbol{\mu}_i \perp \boldsymbol{\mu}_j$ for all $1 \leq i < j \leq k$.

**Assumption 3.3** (Nearly Balanced Classification). The partition $J_\pm$ satisfies $c^{-1} \leq \frac{|J_+|}{|J_-|} \leq c$ for some absolute constant $c \geq 1$.

Our data distribution is similar to that in Vardi et al. (2022) and Frei et al. (2024). In particular, Vardi et al. (2022) consider a setting where data are comprised of $k$ nearly orthogonal data points in $\mathbb{R}^d$. This assumption is further relaxed in Frei et al. (2024), where they assume $k$ clusters with nearly orthogonal cluster means $\{\boldsymbol{\mu}_i\}_{i=1}^k$ (i.e., they have that $\frac{|\langle \boldsymbol{\mu}_i, \boldsymbol{\mu}_j \rangle|}{\|\boldsymbol{\mu}_i\|\|\boldsymbol{\mu}_j\|} = O\left(\frac{1}{k}\right)$ holds for all $i \neq j$). For simplicity, our work focuses on the setting with clusters exactly orthogonal to each other.

## 3.2 NEURAL NETWORK LEARNER

A training dataset $\mathcal{S} := \{(\boldsymbol{x}_i, y_i)\}_{i=1}^n \subseteq \mathbb{R}^d \times \{-1, 1\}$ of size $n$ is randomly sampled from the data distribution $\mathcal{D}(\{\boldsymbol{\mu}_j\}_{j=1}^k, J_\pm)$ and is used to train a two-layer neural network.

**Network Architecture.** We focus on learning two-layer ReLU networks. Such networks are usually defined as $f_{\boldsymbol{\theta}}(\boldsymbol{x}) := \sum_{j=1}^M a_j \operatorname{ReLU}(\langle \boldsymbol{w}_j, \boldsymbol{x} \rangle + b_j)$, where $\boldsymbol{\theta} := \left(\{a_j\}_{j=1}^M, \{\boldsymbol{w}_j\}_{j=1}^M, \{b_j\}_{j=1}^M\right)$ are the parameters of the network, and $\operatorname{ReLU}(\cdot)$ is the ReLU activation function defined as $\operatorname{ReLU}(z) = \max(0, z)$.

For the sake of simplicity, we consider the case where $M = 2m$ is even and fix the second layer as $a_j = \frac{1}{m}$ for $1 \leq j \leq m$ and $a_j = -\frac{1}{m}$ for $m+1 \leq j \leq 2m$, which is a widely adopted setting in the literature of feature learning theory (Allen-Zhu & Li, 2022; Cao et al., 2022; Kou et al., 2023a). With this simplification, we focus on training only the first layer $(\{\boldsymbol{w}_j\}_{j=1}^M, \{b_j\}_{j=1}^M)$ and rewrite the network as

$$f_{\boldsymbol{\theta}}(\boldsymbol{x}) := \frac{1}{m} \sum_{r \in [m]} \operatorname{ReLU}(\langle \boldsymbol{w}_{+1,r}, \boldsymbol{x} \rangle + b_{+1,r}) - \frac{1}{m} \sum_{r \in [m]} \operatorname{ReLU}(\langle \boldsymbol{w}_{-1,r}, \boldsymbol{x} \rangle + b_{-1,r}),$$

where $\boldsymbol{\theta} = \left(\{\boldsymbol{w}_{+1,r}\}_{r=1}^m, \{b_{+1,r}\}_{r=1}^m, \{\boldsymbol{w}_{-1,r}\}_{r=1}^m, \{b_{-1,r}\}_{r=1}^m\right)$ are the trainable parameters, and $\boldsymbol{w}_{+1,r}$ and $b_{+1,r}$ correspond to the neurons with $a_r = \frac{1}{m}$, while $\boldsymbol{w}_{-1,r}$ and $b_{-1,r}$ correspond to the neurons with $a_r = -\frac{1}{m}$.

**Training Objective and Gradient Descent.** The neural network $f_{\boldsymbol{\theta}}(\cdot)$ is trained to minimize the following empirical loss on the training dataset $\mathcal{S}$: $\mathcal{L}(\boldsymbol{\theta}) := \frac{1}{n} \sum_{i=1}^n \ell(y_i f_{\boldsymbol{\theta}}(\boldsymbol{x}_i))$, where $\ell(q) := \log(1 + e^{-q})$ is the logistic loss. We apply gradient descent to minimize this loss:

$$\boldsymbol{\theta}^{(t+1)} = \boldsymbol{\theta}^{(t)} - \eta \nabla \mathcal{L}(\boldsymbol{\theta}^{(t)}), \tag{1}$$

where $\boldsymbol{\theta}^{(t)}$ denotes the parameters at $t$-th iteration for all $t \geq 0$, and $\eta > 0$ is the learning rate. We specify the derivative of ReLU activation as $\operatorname{ReLU}'(z) = \mathbb{1}(z \geq 0)$ in backpropagation. At initialization, we set $\boldsymbol{w}_{s,r}^{(0)} \sim \mathcal{N}(\boldsymbol{0}, \sigma_{\mathrm{w}}^2 \boldsymbol{I}_d)$ and $b_{s,r}^{(0)} \sim \mathcal{N}(0, \sigma_{\mathrm{b}}^2)$ for some $\sigma_{\mathrm{w}}, \sigma_{\mathrm{b}} > 0$.

**Clean Accuracy and Robust Accuracy.** For a given data distribution $\mathcal{D}$ over $\mathbb{R}^d \times \{-1, 1\}$, the clean accuracy of a neural network $f_{\boldsymbol{\theta}} : \mathbb{R}^d \to \mathbb{R}$ on $\mathcal{D}$ is defined as

$$\operatorname{Acc}_{\mathrm{clean}}^{\mathcal{D}}(f_{\boldsymbol{\theta}}) := \mathbb{P}_{(\boldsymbol{x},y) \sim \mathcal{D}} \left[\operatorname{sgn}(f_{\boldsymbol{\theta}}(\boldsymbol{x})) = y\right].$$

In this work, we focus on the $\ell_2$-robustness. The $\ell_2$ $\delta$-robust accuracy of $f_{\boldsymbol{\theta}}$ on $\mathcal{D}$ is defined as

$$\operatorname{Acc}_{\mathrm{robust}}^{\mathcal{D}}(f_{\boldsymbol{\theta}}; \delta) := \mathbb{P}_{(\boldsymbol{x},y) \sim \mathcal{D}} \left[\forall \boldsymbol{\rho} \in \mathbb{B}_\delta : \operatorname{sgn}(f_{\boldsymbol{\theta}}(\boldsymbol{x} + \boldsymbol{\rho})) = y\right],$$

where $\mathbb{B}_\delta := \{\boldsymbol{\rho} \in \mathbb{R}^d : \|\boldsymbol{\rho}\| \leq \delta\}$ is the $\ell_2$-ball centered at the origin with radius $\delta$. We say that a neural network $f_{\boldsymbol{\theta}}$ is $\delta$-robust if $\operatorname{Acc}_{\mathrm{robust}}^{\mathcal{D}}(f_{\boldsymbol{\theta}}; \delta) \geq 1 - \epsilon(d)$ for some function $\epsilon(d)$ that vanishes to zero, i.e., $\epsilon(d) \to 0$ as $d \to \infty$.

**Robust Networks Exist.** In a very similar setting to ours, Frei et al. (2024) show that there exists a two-layer ReLU network that can achieve nearly $100\%$ clean accuracy and $\Omega(\sqrt{d})$-robust accuracy on their data distribution. In our setting, we can also construct a similar network that achieves nearly $100\%$ clean accuracy and $\Omega(\sqrt{d})$-robust accuracy. In particular, such network utilizes one hidden neuron to capture one feature/cluster (i.e., the neural is activated only if the input point is from the corresponding cluster). See Theorem G.3 in Appendix G.2 for the details and Figure 1 for an illustration. However, we will soon show that, despite such $\Omega(\sqrt{d})$-robust network exists, gradient descent is incapable of learning such a robust network, but instead converges to a very different solution with a robust radius that is $\Theta(\sqrt{k})$ times smaller.

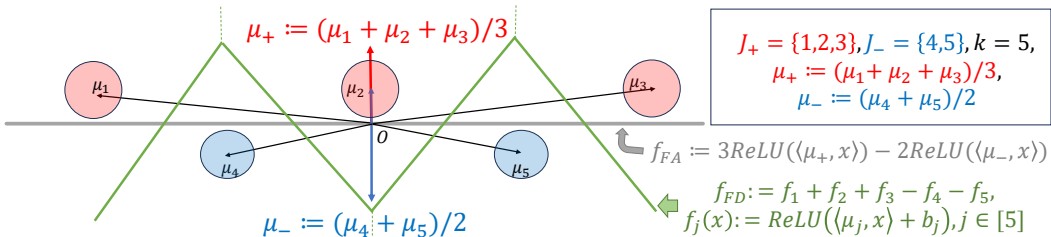

Figure 1: **Schematic illustration of feature-averaging and feature-decoupling**: We consider a dataset with 5 clusters. The first three clusters belong to $J_+$, and the other two to $J_-$. Denote $\boldsymbol{\mu}_+ := (\boldsymbol{\mu}_1 + \boldsymbol{\mu}_2 + \boldsymbol{\mu}_3)/3, \boldsymbol{\mu}_- := (\boldsymbol{\mu}_4 + \boldsymbol{\mu}_5)/2$. For ease of illustration, we assume that $\sum_{j=1}^5 \boldsymbol{\mu}_j = \mathbf{0}$. The feature-averaging classifier $f_{\text{FA}}$ leverages two neurons with averaged features to classify all data, which corresponds to a linear classifier (the gray line). The feature-decoupling classifier $f_{\text{FD}}$ leverages individual features and has more complex polyhedral decision boundary (green lines). Note that the instance is high dimensional and this is only a schematic illustration. The distance between data points and the decision boundary of $f_{\text{FD}}$ (green lines) is much larger than that of $f_{\text{FA}}$ (gray line), which implies that the feature-decoupling classifier is more robust than the feature-averaging one.

## 4 MAIN RESULTS

In this section, we present our main technical results. In Section 4.1, we first present the main result (Theorem 4.5) regarding feature averaging, that is standard gradient descent training finds feature averaging solutions for the data distribution $\mathcal{D}$ and such feature averaging solution is non-robust. In Section 4.2, we demonstrate that if more supervisory information can be obtained (specific cluster categories rather than just binary classification labels), we can achieve feature decoupling via gradient descent on a similar two layer multi-class network. Consequently, we can obtain a binary classification network with optimal robust perturbation radius (Theorem 4.7).

### 4.1 NETWORK LEARNER PROVABLY LEARNS FEATURE-AVERAGING SOLUTION

The prior work by Frei et al. (2024) has showed that, under certain conditions, training a two-layer ReLU network for infinite time converges to a network that can achieve nearly $100\%$ clean accuracy on $\mathcal{D}$ but is only $o(\sqrt{d/k})$-robust. A subsequent work by Min & Vidal (2024) conjectured that the network converges to a specific form of solution, which we refer to as the *feature-averaging network*.

**Definition 4.1** (Feature-Averaging Network). We define $f_{\text{FA}}(\boldsymbol{x})$ as the following function:

$$f_{\text{FA}}(\boldsymbol{x}) := |J_+| \cdot \text{ReLU}\left(\langle \boldsymbol{\mu}_+, \boldsymbol{x} \rangle\right) - |J_-| \cdot \text{ReLU}\left(\langle \boldsymbol{\mu}_-, \boldsymbol{x} \rangle\right),$$

where $\boldsymbol{\mu}_+ := \frac{1}{|J_+|} \sum_{j \in J_+} \boldsymbol{\mu}_j$ is the average of cluster centers in the positive class, and similarly $\boldsymbol{\mu}_- := \frac{1}{|J_-|} \sum_{j \in J_-} \boldsymbol{\mu}_j$ is that for the negative class. We say that a two-layer ReLU network $f_{\boldsymbol{\theta}}(\boldsymbol{x})$ is a *feature-averaging network* if $f_{\boldsymbol{\theta}}(\boldsymbol{x}) = C \cdot f_{\text{FA}}(\boldsymbol{x})$ for some $C > 0$.

**Remark 4.2.** *The feature-averaging network uses the first neuron to process all data within positive clusters, and the second neuron negative clusters. Thus, it can correctly classify clean data. However, it fails to robustly classify perturbed data for a radius larger than $\Omega(\sqrt{d/k})$: in particular, consider the attack vector $\boldsymbol{\rho}$ that aligns with the negative direction of the averaged features, i.e., $\boldsymbol{\rho} \propto -\sum_{j \in J_+} \boldsymbol{\mu}_j + \sum_{j \in J_-} \boldsymbol{\mu}_j$. One can easily check that with $\|\boldsymbol{\rho}\| = \delta = \Omega(\sqrt{d/k})$, the attack is successful, i.e., $\text{sgn}(f_{\text{FA}}(\boldsymbol{x} + \boldsymbol{\rho})) \neq \text{sgn}(f_{\text{FA}}(\boldsymbol{x}))$ due to the linearity of $f_{\text{FA}}(\boldsymbol{x} + \boldsymbol{\rho})$ over $\boldsymbol{\rho}$. See Appendix G.1 for the details, and see Figure 1 for an illustration.*

Our first main result is a non-asymptotic analysis of the training dynamics that explicitly characterizes the solution learned by gradient descent on distribution $\mathcal{D}$ after a finite number of iterations. For theoretical analysis, we make the following assumptions about the hyper-parameters.

**Assumption 4.3** (Choices of Hyper-Parameters). We assume that:

$$d = \Omega(k^{10}) \qquad c = \Theta(1) \qquad n \in [\Omega(k^7), \exp(O(\log^2(d)))]$$
$$m = \Theta(k) \qquad \eta = O(d^{-2}) \qquad \sigma_b^2 = \sigma_w^2 = O(\eta k^{-5}).$$

**Remark 4.4** (Discussion of Hyper-Parameter Choices). *We make specific choices of hyper-parameters for the sake of calculations, and we emphasize that these may not be the tightest possible choices. In particular, we need the data dimension $d$ to be significantly larger than the number of clusters $k$ to ensure all $k$ cluster features are orthogonal within $\mathbb{R}^d$. We further require that the number of samples $n$ is a large polynomial of $k$ to ensure that the network can learn all $k$ cluster features. We assume the learning rate $\eta$ and the initialization magnitude $\sigma_w, \sigma_b$ are sufficiently small, which helps the network to be trained in the feature learning regime (Lyu et al., 2021; Cao et al., 2022; Allen-Zhu & Li, 2023; Kou et al., 2023a).*

Now, everything is ready to state the first main theorem of our paper, which characterizes the weights of the learned network and shows that after a certain number of iterations, the network can be closely approximated by the *feature-averaging network* (defined in Definition 4.1).

**Theorem 4.5.** *In the setting of training a two-layer ReLU network on the binary classification problem $\mathcal{D}(\{\boldsymbol{\mu}_j\}_{j=1}^k, J_{\pm})$ as described in Section 3, under Assumptions 3.2, 3.3 and 4.3, for some $\gamma = o(1)$, after $\Omega(\eta^{-1}) \leq T \leq \exp(\tilde{O}(k^{1/2}))$ iterations, with probability at least $1 - \gamma$, the neural network satisfies the following properties:*

1. *The clean accuracy is nearly perfect:* $\mathrm{Acc}_{\mathrm{clean}}^{\mathcal{D}}(f_{\boldsymbol{\theta}^{(T)}}) \geq 1 - \exp(-\Omega(\log^2 d))$.

2. *Gradient descent leads the network to the feature-averaging regime: there exists a time-variant coefficient $\lambda^{(T)} \in [\Omega(1), +\infty)$ such that for all $s \in \{\pm 1\}$, $r \in [m]$, the weight vector $\boldsymbol{w}_{s,r}^{(T)}$ can be approximated as*

$$\left\| \boldsymbol{w}_{s,r}^{(T)} - \lambda^{(T)} \sum_{j \in J_s} \|\boldsymbol{\mu}_j\|^{-2} \boldsymbol{\mu}_j \right\| \leq o(d^{-1/2})$$

*and the bias terms are sufficiently small, i.e., $\left| b_{s,r}^{(T)} \right| \leq o(1)$.*

3. *Consequently, the network is non-robust: for perturbation radius $\delta = \Omega(\sqrt{d/k})$, the $\delta$-robust accuracy is nearly zero, i.e., $\mathrm{Acc}_{\mathrm{robust}}^{\mathcal{D}}(f_{\boldsymbol{\theta}^{(T)}}; \delta) \leq \exp(-\Omega(\log^2 d))$.*

We provide a proof sketch for Theorem 4.5 in Appendix C (see the full proof in Appendix E.2). Theorem 4.5 suggests that the weight vector aligns with the average of cluster features: the direction of the weight vector associated with a positive neuron converges to the average of positive cluster features $\boldsymbol{\mu}_+$, and that associated with a negative neuron to the average of negative cluster features $\boldsymbol{\mu}_+$. Moreover, the above feature-averaging property of learned network implies non-robustness, i.e., the learned network is only $o(\sqrt{d/k})$-robust although an $\Omega(\sqrt{d})$-robust solution exists as we proved in Section 3.

As a corollary of Theorem 4.5, we resolve the conjecture proposed in Min & Vidal (2024) in our setting.

**Theorem 4.6** (Conjecture 1 from Min & Vidal (2024)). *In the setting of Theorem 4.5, we have that $\inf_{C>0} \sup_{\boldsymbol{x} \in \mathbb{R}^d : \|\boldsymbol{x}\|_2 = \sqrt{d}} |Cf_{\mathrm{FA}}(\boldsymbol{x}) - f_{\boldsymbol{\theta}^{(T)}}(\boldsymbol{x})| = o(1)$, where $f_{\mathrm{FA}}(\boldsymbol{x})$ is the feature-averaging network (Definition 4.1).*

Under a similar orthogonal cluster data assumption, Min & Vidal (2024) conjecture that two-layer neural network converges to the feature-averaging solution via gradient flow training with small initialization. They empirically validate the conjecture via experiments on synthetic datasets. Theorem 4.6 provides a rigorous proof for the conjecture, although the original conjecture is stated under a slightly different setting from ours. In their setting, the second layer of the network is also trainable, but we fix the second layer for simplicity. We also require certain assumptions on the hyperparameters, which has been discussed in details in Assumption 4.3 and Remark 4.4.

## 4.2 FINE-GRAINED SUPERVISION IMPROVES ROBUSTNESS

We have shown that gradient descent is unable to differentiate individual cluster features, which causes non-robustness. Hence, a natural question is: if we provided more fine-grained feature-level supervision, can gradient descent learn a robust solution? We show that this is indeed possible in the case where each data point is labeled with the cluster it belongs to, rather than just a binary label.

**Fine-Grained Supervision.**   Following the setting in Section 3, we consider the binary classification task with data distribution $\mathcal{D}(\{\boldsymbol{\mu}_j\}_{j=1}^k, J_\pm)$. But instead of training the model directly to predict the binary labels, we assume that we are able to label each data point with the cluster $\hat{y} \in [k]$ it belongs to, and then we train a $k$-class classifier to predict the cluster labels. More specifically, we first sample a training set $\mathcal{S} := \{(\boldsymbol{x}_i, y_i)\}_{i=1}^n \subseteq \mathbb{R}^d \times \{\pm 1\}$ from $\mathcal{D}$, along with the cluster labels $\{\tilde{y}_i\}_{i=1}^n$ for all data points. Then a $k$-class neural network classifier is trained on $\tilde{\mathcal{S}} := \{(\boldsymbol{x}_i, \tilde{y}_i)\}_{i=1}^n \subseteq \mathbb{R}^d \times [k]$.

**Multi-Class Network Classifier.**   We train the following two-layer neural network for the $k$-class classification mentioned above: $\boldsymbol{F_\theta}(\boldsymbol{x}) := (f_1(\boldsymbol{x}), f_2(\boldsymbol{x}), \ldots, f_k(\boldsymbol{x})) \in \mathbb{R}^k$, where $f_j(\boldsymbol{x}) := \frac{1}{h} \sum_{r=1}^h \mathrm{ReLU}(\langle \boldsymbol{w}_{j,r}, \boldsymbol{x} \rangle)$, $\boldsymbol{\theta} := (\boldsymbol{w}_{1,1}, \boldsymbol{w}_{1,2}, \ldots, \boldsymbol{w}_{k,h}) \in \mathbb{R}^{khd}$ are trainable weights, and $h = \Theta(1)$. One can think $\{\mathrm{ReLU}(\langle \boldsymbol{w}_{j,r}, \boldsymbol{x} \rangle)\}_{j \in [k], r \in [h]}$ as $kh = \Theta(k)$ neurons partitioned into $k$ groups, where the corresponding second layer weights are set in a way that the $j$-th group only contributes to the $j$-th output $f_j(\boldsymbol{x})$ of the network. The output $\boldsymbol{F_\theta}(\boldsymbol{x})$ is converted to probabilities using the softmax function, namely $p_j(\boldsymbol{x}) := \frac{\exp(f_j(\boldsymbol{x}))}{\sum_{i=1}^k \exp(f_i(\boldsymbol{x}))}$ for $j \in [k]$. For predicting the binary label for the original binary classification task on $\mathcal{D}$, we take the difference of the probabilities of the positive and negative classes, i.e., $F_{\boldsymbol{\theta}}^{\mathrm{binary}}(\boldsymbol{x}) := \sum_{j \in J_+} p_j(\boldsymbol{x}) - \sum_{j \in J_-} p_j(\boldsymbol{x})$. The clean accuracy $\mathrm{Acc}_{\mathrm{clean}}^{\mathcal{D}}(F_{\boldsymbol{\theta}}^{\mathrm{binary}})$ and $\delta$-robust accuracy $\mathrm{Acc}_{\mathrm{robust}}^{\mathcal{D}}(F_{\boldsymbol{\theta}}^{\mathrm{binary}}; \delta)$ are then defined similarly as before.

**Training Objective and Gradient Descent with Fine-Grained Supervision.** We train the multi-class network $\boldsymbol{F_\theta}(\boldsymbol{x})$ to minimize the cross-entropy loss $\mathcal{L}_{\mathrm{CE}}(\boldsymbol{\theta}) := -\frac{1}{n} \sum_{i=1}^n \log p_{\tilde{y}_i}(\boldsymbol{x}_i)$. Similar to Section 3, we use gradient descent to minimize the loss function $\mathcal{L}_{\mathrm{CE}}(\boldsymbol{\theta})$ with learning rate $\eta$, i.e., $\boldsymbol{\theta}^{(t+1)} = \boldsymbol{\theta}^{(t)} - \eta \nabla_{\boldsymbol{\theta}} \mathcal{L}_{CE}(\boldsymbol{F}_{\boldsymbol{\theta}^{(t)}})$. At initialization, we set $\boldsymbol{w}_{j,r}^{(0)} \sim \mathcal{N}(0, \sigma_{\mathrm{w}}^2 \boldsymbol{I}_d)$ for some $\sigma_{\mathrm{w}} > 0$.

**GD Finds Robust Networks.** In contrast to the feature-averaging implicit bias in our previous setting (Theorem 4.5), the following theorem shows that with fine-grained supervision, gradient descent converges to a neural network that learns decoupled features, i.e., the weight of each neuron is aligned with one cluster feature.

**Theorem 4.7.** *In the setting of training a multi-class network on the multiple classification problem $\tilde{\mathcal{S}} := \{(\boldsymbol{x}_i, \tilde{y}_i)\}_{i=1}^n \subseteq \mathbb{R}^d \times [k]$ as described in the above, under Assumptions 3.2, 3.3 and 4.3, for some $\gamma = o(1)$, after $\Omega(\eta^{-1} k^8) \leq T \leq \exp(\tilde{O}(k^{1/2}))$ iterations, with probability at least $1 - \gamma$, the neural network satisfies the following properties:*

1. *The clean accuracy is nearly perfect: $\mathrm{Acc}_{\mathrm{clean}}^{\mathcal{D}}(F_{\boldsymbol{\theta}^{(T)}}^{\mathrm{binary}}) \geq 1 - \exp(-\Omega(\log^2 d))$.*

2. *The network converges to the feature-decoupling regime: there exists a time-variant coefficient $\lambda^{(T)} \in [\Omega(\log k), +\infty)$ such that for all $j \in [k]$, $r \in [h]$, the weight vector $\boldsymbol{w}_{j,r}^{(T)}$ can be approximated as*

$$\left\| \boldsymbol{w}_{j,r}^{(T)} - \lambda^{(T)} \|\boldsymbol{\mu}_j\|^{-2} \boldsymbol{\mu}_j \right\| \leq o(d^{-1/2}).$$

3. *Consequently, the corresponding binary classifier achieves optimal robustness: for perturbation radius $\delta = O(\sqrt{d})$, the $\delta$-robust accuracy is also nearly perfect, i.e., $\mathrm{Acc}_{\mathrm{robust}}^{\mathcal{D}}(F_{\boldsymbol{\theta}^{(T)}}^{\mathrm{binary}}; \delta) \geq 1 - \exp(-\Omega(\log^2 d))$.*

The detailed proof can be found in Appendix F.3. Theorem 4.7 manifests that the multi-class network learns the decoupled features, and the induced binary classifier achieves optimal robustness. See Figure 1 for an illustration. Instead of leveraging the bias term to filter out cluster noise as the feature-decoupling classifier $f_{\mathrm{FD}}$ that we illustrated in Figure 1 and Theorem G.3, the soft-max operator of $F_{\boldsymbol{\theta}}^{\mathrm{binary}}$ plays a similar role here.

It can be easily verified that $F_{\boldsymbol{\theta}^{(T)}}^{\mathrm{binary}}$ achieves optimal robustness radius (up to constant factor) since the distance between distinct cluster centers is at most $\Theta(\sqrt{d})$ (i.e. $\|\boldsymbol{\mu}_i - \boldsymbol{\mu}_j\| = \Theta(\sqrt{d}), \forall i \neq j$).

**Convergence to Robust Networks Requires Implicit Bias.** In fact, adding more fine-grained supervision signals does not trivially lead to decoupled features and robustness, since the above network found by gradient descent is not the only solution that can achieve $100\%$ clean accuracy. As a counterexample, we show that there exists a multi-class network that achieves perfect clean accuracy but is not $\Omega(\sqrt{d/k})$-robust, which is formally given in the following proposition.

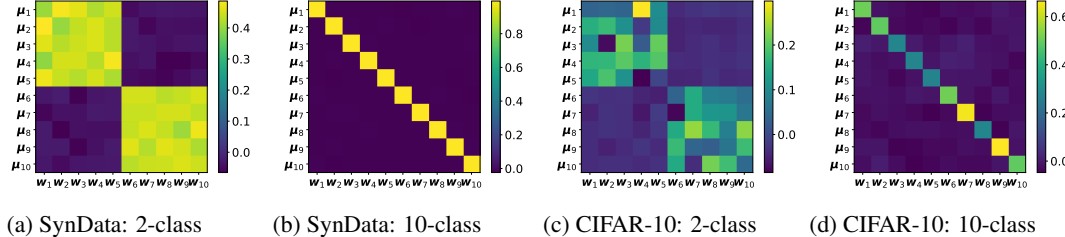

|  (a) SynData: 2-class | (b) SynData: 10-class | (c) CIFAR-10: 2-class | (d) CIFAR-10: 10-class |

Figure 2: Illustration of feature averaging and feature decoupling on synthetic dataset (a,b) and CIFAR-10 dataset (c,d). Figure (a) and Figure (c) correspond to models trained using 2-class labels, and Figure (b) and Figure (d) correspond to models trained using 10-class labels, respectively. Each element in the matrix, located at position $(i, j)$, represents the average cosine value of the angle between the feature vector $\boldsymbol{\mu_i}$ of the $i$-th feature and the equivalent weight vector $\boldsymbol{w}_j$ of the $f_j(\cdot)$.

**Proposition 4.8.** *Consider the following multi-class network $\boldsymbol{F}_{\tilde{\boldsymbol{\theta}}}$: for all $j \in [k]$, the sub-network $f_j$ has only single neuron ($h = 1$) and is defined as $f_j(\boldsymbol{x}) = \mathrm{ReLU}\left(\langle \boldsymbol{\mu}_j + \sum_{l \in J_s} \boldsymbol{\mu}_l, \boldsymbol{x} \rangle\right)$, where cluster $j$ has binary label $s \in \{\pm 1\}$. With probability at least $1 - \exp(-\Omega(\log^2 d))$ over $\tilde{S}$, we have that $\mathcal{L}_{\mathrm{CE}}(\tilde{\boldsymbol{\theta}}) \leq \exp(-\Omega(d)) = o(1)$, where $\tilde{\boldsymbol{\theta}}$ denotes the weights of $\boldsymbol{F}_{\tilde{\boldsymbol{\theta}}}$. Moreover, $\mathrm{Acc}_{\mathrm{clean}}^{\mathcal{D}}(F_{\tilde{\boldsymbol{\theta}}}^{\mathrm{binary}}) \geq 1 - \exp(-\Omega(\log^2 d))$, $\mathrm{Acc}_{\mathrm{robust}}^{\mathcal{D}}(F_{\tilde{\boldsymbol{\theta}}}^{\mathrm{binary}}; \Omega(\sqrt{d/k})) \leq \exp(-\Omega(\log^2 d))$.*

## 5 EXPERIMENTS

### 5.1 FEATURE AVERAGING AND FEATURE DECOUPLING

To validate our theoretical results, we conduct experiments on the synthetic dataset as we mentioned in Section 3 and the CIFAR-10 dataset, described as follows:

- **Synthetic Dataset.** We generate the synthetic data following the data distribution in Section 3. Specifically, we choose the hyper-parameters as $k = 10, d = 3072, m = 5, n = 1000, \alpha = \sigma = 1, \eta = 0.001, \sigma_w = \sigma_b = 0.00001, T = 100$. For simplicity, we denote the weights of the two-layer network as $\boldsymbol{w}_1, \boldsymbol{w}_2, \ldots, \boldsymbol{w}_{10}$ (where the first five weights correspond positive neurons and the other five weights correspond negative neurons). We also set the first five clusters as positive and the others as negative. Additionally, we provide an ablation study for other choices of hyper-parameters (see the details in Appendix B).

- **Binary Classification on CIFAR-10.** We create a binary classification task from the CIFAR-10 dataset by merging the first five classes into one class and the other five classes into the other class. We use the normal 10-classification on CIFAR-10 as the 10-class task.

- **Pre-trained Feature Extractor.** We utilize a ResNet18 model pre-trained on CIFAR-10 and replaced the model's final layer with a two-layer ReLU neural network as described in our theory (fixed second layer as diagonal form, i.e., $f_j(\boldsymbol{z}) := \frac{1}{h}\sum_{r=1}^{h} \mathrm{ReLU}(\langle \boldsymbol{w}_{j,r}, \boldsymbol{z} \rangle), \forall j \in [10]$, where $\boldsymbol{z}$ denotes the hidden representation of the penultimate layer). We only train the last two layers. We choose the width of the first layer to be 30 ($h = 3$) to ensure that the accuracy of the pre-trained model was not compromised. Then, inspired by the theoretical study about neuron collapse (Papyan et al., 2020), we calculate the average value of the penultimate layer output of the neural network for each class as the corresponding feature $\boldsymbol{\mu}_i$ of that class $i \in [10]$. For 10-classification, we use $\boldsymbol{w}_j := \frac{1}{h}\sum_{r=1}^{h} \boldsymbol{w}_{j,r}$ as the equivalent weight of $f_j$. For the 15 positive weights and 15 negative weights in the binary classification network, we equally divide them into 5 positive classes and 5 negative classes to ensure a fair comparison , which ensures that two models both have the same form $\boldsymbol{F} := (f_1, f_2, \ldots, f_{10}) \in \mathbb{R}^{10}$ and each sub-network $f_j$ corresponds to a weight vectors $\boldsymbol{w}_j$.

**Experiment Results.** The results are shown in Figure 2, which demonstrates our theoretical findings: 2-classification model learns feature-averaging solution while 10-classification model learns the feature-decoupling solution. Concretely, Figure 2a and Figure 2c correspond to our feature-averaging

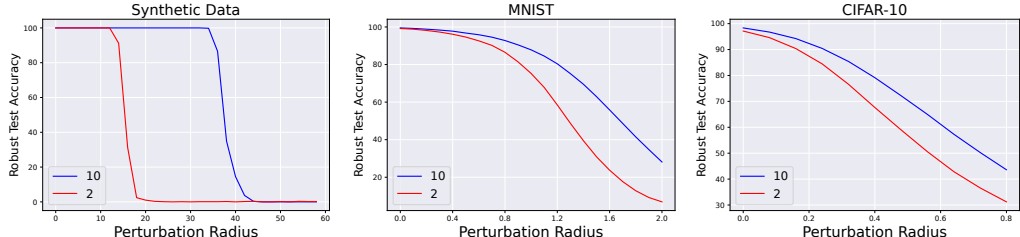

Figure 3: **Verifying robustness improvement:** We compare adversarial robustness between model trained by 2-class labels (red line) and model trained by 10-class labels (blue line) on synthetic data (the left), MNIST (the middle) and CIFAR-10 (the right).

regime in Theorem 4.5, where the correlations between each weight vector of positive (negative) neuron and all positive (negative) cluster features are nearly-equally larger than those between each weight vector of positive (negative) neuron and all negative (positive) cluster features; Figure 2b and Figure 2d correspond to our feature-decoupling regime in Theorem 4.7, where the correlation matrix is nearly diagonal. Additionally, we also provide a transfer learning experiment based on the CLIP model (Radford et al., 2021) to further verify our theory (see the details in the Appendix H).

## 5.2 ROBUSTNESS IMPROVEMENT FROM FINE-GRAINED SUPERVISION INFORMATION

Moreover, we aim to verify whether the model trained with fine-grained supervision information (i.e., 10-class labels) is more robust compared to the model trained with only binary (2-class) labels.

**Experiment Settings.** To ensure fairness in the comparison, we sum the logits corresponding to the 5 positive classes and subtracted the sum of the logits corresponding to the 5 negative classes from the 10-class model's output. This result is used as the binary classification output for the 10-class model. The robust accuracy is measured by using the standard PGD attacks (Madry et al., 2018) with different $\ell_2$-pertubation radius. We run experiments in the following datasets:

- **Synthetic Dataset.** We generate synthetic data as the same as that in Section 5.1.
- **Binary Classification on MNIST and CIFAR-10.** To further verify our theory in deep neural networks, on both MNIST and CIFAR-10 datasets, we train ResNet18 models from scratch with normal 10-classification labels and 2-classification labels (MNIST: parity-classification; CIFAR-10: binary-classification as that we mentioned in Section 5.1).

**Experiment Results.** The results are presented in Figure 3. With the perturbation radius increasing, we can see that the models trained with 10-class labels have higher robust test accuracy than those trained with 2-class labels in all datasets. This collaborates with our theoretical results (Theorem 4.5 and Theorem 4.7) that the models can achieve better robustness with more supervised information.

## 6 CONCLUSION

This paper exposes "Feature Averaging" as an implicit bias in gradient descent that may compromise the robustness of deep neural networks. Theoretical insights from a two-layer ReLU network reveal a tendency for gradient descent to average/combine individually meaningful features, which can lead to a loss of distinct discriminative information. We demonstrate that with more detailed feature-level supervision, the networks can learn to differentiate these features, enhancing model robustness. This is supported by empirical evidence from both synthetic and real-world data, including MNIST and CIFAR-10. Our findings not only deepen our understanding of adversarial examples in deep learning but also suggest that fine-grained supervision can enhance the robustness of deep neural networks against adversarial attacks.

### ACKNOWLEDGMENTS

Binghui Li is partially supported by the Elite Ph.D. Program in Applied Mathematics for PhD Candidates in Peking University. We thank anonymous reviewers for their valuable suggestions.

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

CONTENTS

# A    CONNECTIONS OF OUR RESULTS WITH OTHER EXPLANATIONS OF ADVERSARIAL EXAMPLES

(1) **Approximate Linearity of the Model:** Earlier hypothesis about the origin of adversarial examples (e.g., Goodfellow et al. (2014)) had proposed the idea that the existence of adversarial examples is related to the fact the model $f_{\boldsymbol{\theta}}(\boldsymbol{x})$ is approximately linear. Subsequently, there is a sequence of theoretical studies showing that adversarial examples exist abundantly in the input space for neural networks with random weights (*without training*), and a main insight is that such random networks are approximately linear and with high probability an input point is close to the decision boundary (by isoperimetry argument) (see e.g., (Gilmer et al., 2018; Bubeck et al., 2021a; Bartlett et al., 2021; Montanari & Wu, 2023)). Our Theorem 4.5 proves similar approximate linearity. See details in the proof intuition of Theorem G.2. Think of the special case that the weight vector corresponding to each neuron is exactly the average of the cluster means ($\boldsymbol{\mu}_+ := \frac{1}{|J_+|} \sum_{j \in J_+} \boldsymbol{\mu}_j$ or $\boldsymbol{\mu}_- := \frac{1}{|J_-|} \sum_{j \in J_-} \boldsymbol{\mu}_j$) and $\sum_{j=1}^k \boldsymbol{\mu}_j = \boldsymbol{0}$. In this case, the two-layer network reduces to a simple linear model w.r.t. the perturbation. Theorem 4.5 also shows it leads to adversarial examples for *trained* neural network (albeit with different data distribution from the aforementioned work). Our result is also related to the *dimpled manifold hypothesis* (Shamir et al., 2021), which proposed that during training neural network first finds a simple decision boundary that is close to most training points.

(2) **Non-Robust Features:** Another appealing point of view was developed in Ilyas et al. (2019), which proposed that adversarial examples are related to the presence of non-robust features. Here, a feature refers to an individual measurable property or characteristic of the data used by the model to make predictions. Robust features refer to those that are not easily disturbed or affected by variations or noise in the data, allowing the model to maintain stable performance. Non-robust features, on the other hand, are features derived from patterns in the data distribution that are highly predictive but fragile, making them often incomprehensible to humans. They showed empirically that neural networks learn both robust and non-robust features that are useful to classify clean images. In image classification tasks, Ilyas et al. (2019), Engstrom et al. (2019), Tsilivis & Kempe (2022) and Li & Li (2024) visualized both robust and non-robust features. While robust features are more perceptually meaningful for human, non-robust features resemble noise and artifacts. Interestingly, they showed that non-robust feature can be leveraged to construct adverserial examples for DNN. Our paper presents a theoretical setting in which neural networks provably learn non-robust features (due to feature averaging), despite the existence of more robust features. Moreover, we prove that the learned non-robust feature ($\boldsymbol{\mu}_+$ or $\boldsymbol{\mu}_-$) can be utilized to attack the feature-averaging network.

(3) **Relation to the Lower Bound Examples in Li et al. (2022a):** From the perspective of expressivity, Li et al. (2022a) constructed a lower bound example (see an illustration in Figure 4), for which there is non-robust linear classifier, but the set of robust solutions requires a hypothesis class of a much larger (in fact exponentially large) VC-dimension. This partially explains why neural networks are non-robust (unless they are exponentially large). The construction of our data distribution (as well as that in (Vardi et al., 2022; Frei et al., 2024)) echoes the essence of this lower bound example in spirit, and our results can be seen as an explanation from the perspective of optimization.

(4) **Relation to Frequency Bias in Xu et al. (2019) and Xu et al. (2024):** These works refer to the phenomena that deep neural networks generally learn lower-frequency features first, and then the higher-frequency ones. This can be seen as a particular form of simplicity bias. The feature averaging bias studied is also a form of simplicity bias: under our theoretical setup, the simplicity refers to the linear combination of cluster features, and it is closely related to the approximate linearity of the decision boundary, as discussed in (1). Hence, both studies assert that neural networks tend to favor simplicity during the initial stages of training, sharing a similar underlying spirit.

(5) **Relation to Superposition in Gandelsman et al. (2024):** Similar "averaging" or "superposition" effects are also observed in the work of Gandelsman et al. (2024). In particular, they proposed an automated interpretation of individual neurons in CLIP models (Radford et al., 2021) by generating textual descriptions of their functions. They observed that in CLIP models, each neuron may encode several distinct and unrelated concepts, and they leveraged this effect to construct "semantic" adversarial examples. Specifically, they decomposed a second-order effect of each neuron into a sparse set of word directions in the joint text-image space, and co-occurring words in these sets can be used for the mass generation of semantic adversarial images. For example, the text prompt "A cat lounging in the sun, with a group of elephants in the background and a value sign in the foreground"

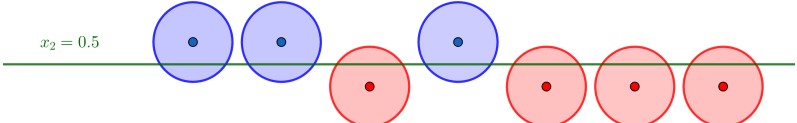

Figure 4: **A schematic illustration of the construction in Li et al. (2022a)**: The positive class consists of blue points and the negative class the red points. In their lower bound, there are in fact exponentially blue points slightly above the hyperplane and exponentially many red ones slightly below it. The hyperplane has perfect clean accuracy but is non-robust, while a more robust classifier exists (by classifying the blue balls from the red balls). One can observe the conceptual similarity with Figure 1.

generates an image where CLIP's prediction assigns a 65% probability to the label "dog" and a 35% probability to the label "cat." Similarly, in our setting, neurons also encode multiple features, and our adversarial example is constructed by combining multiple features together.

(6) **Relation to Fine-Grained Annotation in Sitawarin et al. (2022) and Li et al. (2024):** These works have demonstrated that combining human prior knowledge with end-to-end learning can enhance the robustness of deep neural networks for object classification through a part-based model that makes predictions by recognizing the parts of the object in a bottom-up manner. By utilizing richer annotations, this approach helps networks learn more robust features without requiring additional samples or larger models. Specifically, part-based models, which provide detailed segmentations of objects, allow the network to focus on discriminative object parts, thereby improving its ability to resist adversarial attacks. Consistent with these findings, our theory and experiments show that fine-grained supervisory information can similarly improve the performance of robust classification. We believe that incorporating additional supervision signals to further enhance robustness could be an important direction for future research.

## B ADDITIONAL SYNTHETIC EXPERIMENTS

### B.1 ABLATION STUDY ON SYNTHETIC DATASETS

We conducted several additional experiments on synthetic datasets, as an ablation study for choices of hyper-parameters. The goal is to show that feature averaging happens in different settings.

**Baseline Setting.** We choose the hyper-parameters as $k = 10, d = 3072, m = 5, n = 1000, \alpha = \sigma = 1, \eta = 0.001, \sigma_w = \sigma_b = 0.00001, T = 100$. We denote the weights of the two-layer network as $\boldsymbol{w}_1, \boldsymbol{w}_2, \ldots, \boldsymbol{w}_{10}$ (where the first five weights correspond positive neurons and the other five weights correspond negative neurons). We also set that the first five clusters are positive and the others are negative. Each element in the matrix, located at position $(i, j)$, represents the average cosine value of the angle between the feature vector $\boldsymbol{\mu_i}$ and the weight vector $\boldsymbol{w}_j$. The experiment result under baseline setting is presented as Figure 5 (a), Figure 6 (b), Figure 7 (c) and Figure 8 (b).

**Effect of the number of samples.** We vary the number of samples as $n = 1000, 10000, 50000$. See results in Figure 5. It shows that feature-averaging can not be mitigated via more training data.

**Effect of the learning rate.** We vary the learning rate as $\eta = 0.01, 0.001, 0.0001$. See results in Figure 6. It shows that the assumption about small learning rate is necessary for feature averaging.

**Effect of the initialization magnitude.** We vary the initialization magnitude as $\sigma_w = \sigma_b = 0.001, 0.0001, 0.00001$. See results in Figure 7. It shows that the assumption about small initialization is necessary for feature averaging.

**Effect of the signal-to-noise ratio.** We vary the signal-to-noise ratio as $\mathrm{SNR} := \alpha/\sigma = 0.5, 1, 2$. See results in Figure 8. It shows that our results can also apply to $\mathrm{SNR} = \Theta(1)$ case.

**Effect of the orthogonal condition.** We vary the cosine value of the angle between different cluster center features as $\cos(\boldsymbol{\mu}_i, \boldsymbol{\mu}_j) = 0.00001, 0.001, 0.09, \forall i \neq j$. See results in Figure 9. It shows that the exact orthogonal condition can be relaxed to a nearly orthogonal setting, under which feature averaging still happens.

**Effect of the equinorm condition.** We vary the minimal and maximal norms of cluster centers as $\min_i \|\boldsymbol{\mu}_i\|/\max_i \|\boldsymbol{\mu}\| = 1.0/1.0, 0.8/1.2, 0.6/1.4$ (from the smallest norm to the largest norm, an arithmetic progression is formed). See results in Figure 10. It shows that the exact equinorm condition can be relaxed to general non-equinorm setting, where feature averaging still occurs.

### B.2 ADVERSARIAL TRAINING ON SYNTHETIC DATASETS

To investigate the relationship between our theoretical results and adversarial training, we conducted the following experiments on synthetic data.

**Experiment Settings.** We employ hyper-parameters that are largely consistent with the baseline settings: $k = 10, d = 3072, m = 50, n = 1000, \alpha = \sigma = 1, \eta = 0.001, \sigma_w = \sigma_b = 0.00001$, and $T = 100$. The only modification is increasing the network width $m$, as adversarial training requires a wider network for optimal performance. We utilize PGD-based adversarial training. During adversarial training, we need to select the $\ell_2$ radius for adversarial attacks. Through experiment, we find that training with an attack radius of 40 achieved the best robustness. We visualize the weights and feature correlations of the network trained with this radius using the same methods.

Additionally, we compare the robustness of the network obtained via adversarial training with that of networks trained using binary and 10-class labels. To illustrate the effect of different attack radii on the robustness of networks trained with adversarial Training, we include results for a network trained with an attack radius of 20 in the robustness experiments.

**Experiment Results.** See experiment results in Figure 11. The results indicate that networks trained with adversarial training do exhibit a tendency to learn feature decoupling solutions. However, the degree of decoupling is less pronounced compared to networks trained with fine-grained supervision. In terms of robustness, adversarial training does provide significant improvements, but it remains slightly inferior to the robustness achieved through fine-grained supervision.

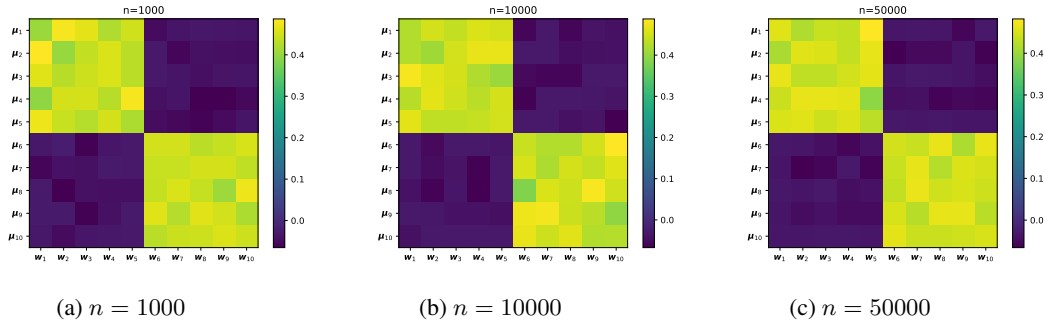

(a) $n = 1000$        (b) $n = 10000$        (c) $n = 50000$

Figure 5: Illustration of feature averaging on synthetic dataset, when varying the number of samples $n$.

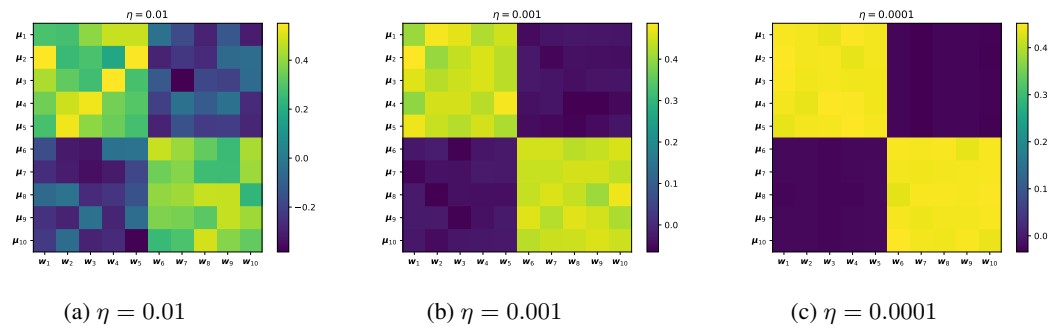

(a) $\eta = 0.01$        (b) $\eta = 0.001$        (c) $\eta = 0.0001$

Figure 6: Illustration of feature averaging on synthetic dataset, when varying learning rate $\eta$.

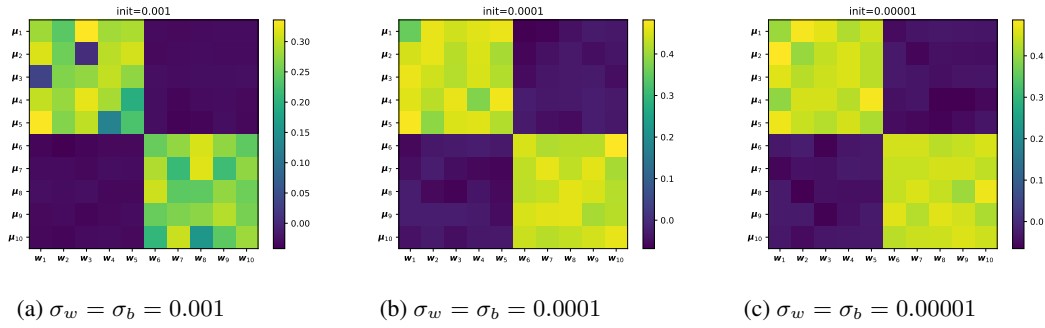

(a) $\sigma_w = \sigma_b = 0.001$     (b) $\sigma_w = \sigma_b = 0.0001$     (c) $\sigma_w = \sigma_b = 0.00001$

Figure 7: Illustration of feature averaging on synthetic dataset, when varying the initialization magnitude $(\sigma_b^2, \sigma_w^2)$.

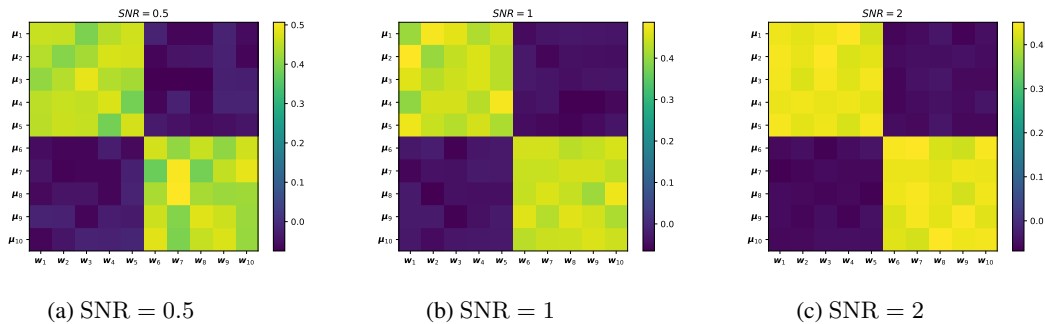

(a) SNR $= 0.5$      (b) SNR $= 1$      (c) SNR $= 2$

Figure 8: Illustration of feature averaging on synthetic dataset, when varying the signal-to-noise ratio (SNR).

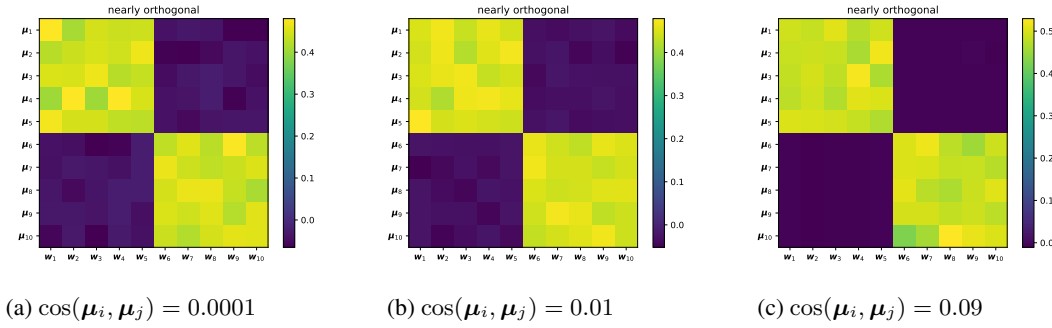

(a) $\cos(\boldsymbol{\mu}_i, \boldsymbol{\mu}_j) = 0.0001$     (b) $\cos(\boldsymbol{\mu}_i, \boldsymbol{\mu}_j) = 0.01$     (c) $\cos(\boldsymbol{\mu}_i, \boldsymbol{\mu}_j) = 0.09$

Figure 9: Illustration of feature averaging on synthetic dataset, when varying the orthogonal condition.

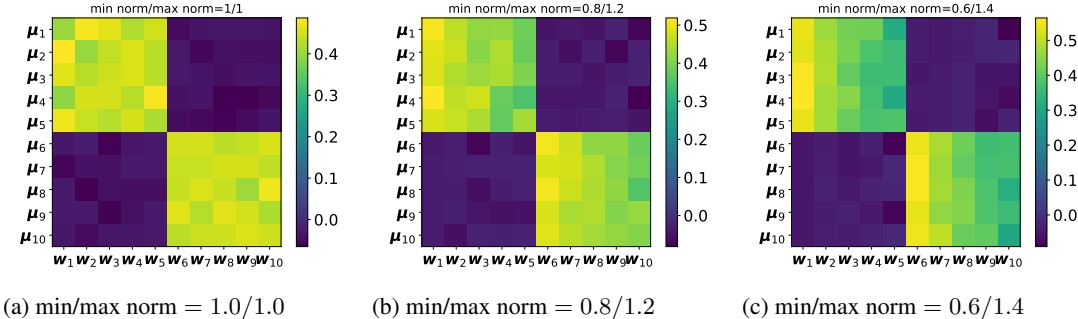

(a) min/max norm $= 1.0/1.0$      (b) min/max norm $= 0.8/1.2$      (c) min/max norm $= 0.6/1.4$

Figure 10: Illustration of feature averaging on synthetic dataset, when varying the equinorm condition.

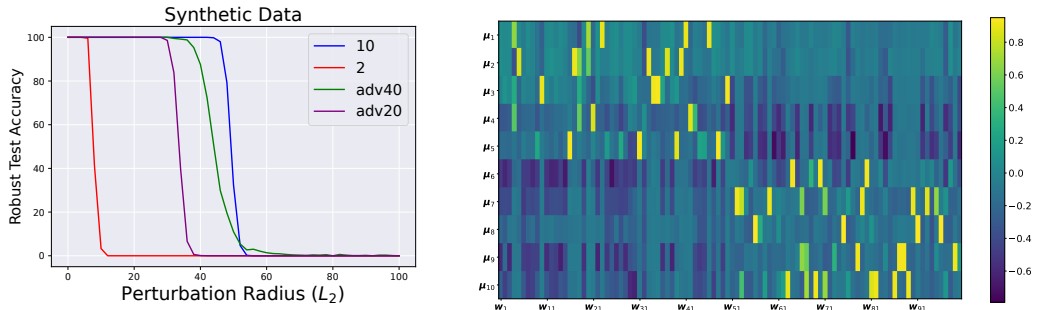

Figure 11: **Adversarial Training on Synthetic Datasets.** The left: we compare adversarial robustness among model trained by 2-class labels (red line), models trained by adversarial training with perturbation radius $= 20$ (purple) and $= 40$ (green) and model trained by 10-class labels (blue line) on synthetic data. The right: each element in the matrix, located at position $(i, j)$, represents the average cosine value of the angle between the feature vector $\boldsymbol{\mu_i}$ of the $i$-th feature and the equivalent weight vector $\boldsymbol{w}_j$ of the $f_j(\cdot)$.

## C    ANALYSIS OF TRAINING DYNAMICS FOR FEATURE-AVERAGING REGIME

In this section, we present a proof sketch of Theorem 4.5, where we provide a detailed analysis of training dynamics in feature-averaging regime.

### C.1    DERIVING DYNAMICS OF COEFFICIENTS FROM GRADIENT DESCENT

By rigorously analyzing the gradient descent iterations, we know that each neuron is situated within a span that encompasses the collective cluster features and the intrinsic noise of the training data points. This span is explicitly characterized by the weight-feature correlations, which is shown as:

**Lemma C.1** (Weight Decomposition). *During the training dynamics, there exists the following normalized coefficient sequences $\lambda_{s,r,j}^{(t)}$ and $\sigma_{s,r,i}^{(t)}$ for each pair $s \in \{-1, +1\}, r \in [m], j \in [k], i \in [n]$ such that*

$$\boldsymbol{w}_{s,r}^{(t)} = \boldsymbol{w}_{s,r}^{(0)} + \sum_{j \in [k]} \lambda_{s,r,j}^{(t)} \|\boldsymbol{\mu}_j\|^{-2} \boldsymbol{\mu}_j + \sum_{i \in [n]} \sigma_{s,r,i}^{(t)} \|\boldsymbol{\xi}_i\|^{-2} \boldsymbol{\xi}_i.$$

Then, we give the restatement of Theorem 4.5 as follows.

**Theorem C.2** (Restatement of Theorem 4.5). *In the setting of training a two-layer ReLU network on the binary classification problem $\mathcal{D}(\{\boldsymbol{\mu}_j\}_{j=1}^k, J_\pm)$ as described in Section 3, under Assumptions 3.2, 3.3 and 4.3, for some $\gamma = o(1)$, after $\Omega(\eta^{-1}) \leq T \leq \exp(\tilde{O}(k^{1/2}))$ iterations, with probability at least $1 - \gamma$, the neural network satisfies the following properties:*

    *1. The clean accuracy is nearly perfect: $\mathrm{Acc}_{\mathrm{clean}}^{\mathcal{D}}(f_{\boldsymbol{\theta}^{(T)}}) \geq 1 - \exp(-\Omega(\log^2 d))$.*

2. *Gradient descent leads the network to the feature-averaging regime: there exists a time-variant coefficient $\lambda^{(T)} \in [\Omega(1), +\infty)$ such that for all $s \in \{\pm 1\}$, $r \in [m]$, the weight vector $\boldsymbol{w}_{s,r}^{(T)}$ can be approximated as*

$$\left\| \boldsymbol{w}_{s,r}^{(T)} - \lambda^{(T)} \sum_{j \in J_s} \|\boldsymbol{\mu}_j\|^{-2} \boldsymbol{\mu}_j \right\| \leq o(d^{-1/2})$$

*and the bias terms are sufficiently small, i.e., $\left| b_{s,r}^{(T)} \right| \leq o(1)$.*

3. *Consequently, the network is non-robust: for perturbation radius $\delta = \Omega(\sqrt{d/k})$, the $\delta$-robust accuracy is nearly zero, i.e., $\mathrm{Acc}_{\mathrm{robust}}^{\mathcal{D}}(f_{\boldsymbol{\theta}^{(T)}}; \delta) \leq \exp(-\Omega(\log^2 d))$.*

In light of Lemma C.1 and the second item of the above theorem indicates that $\boldsymbol{w}_{s,r}^{(t)}$ is approximately proportional to the average of features in $J_s$ (the coefficients from the same class are large and approximately the same, and those from the opposite class are small).

In order to deal with the behavior of ReLU activation, we define $S_{s,i}^{(t)} := \{j \in [m] : \langle \boldsymbol{w}_{s,j}^{(t)}, \boldsymbol{x}_i \rangle + b_{s,j}^{(t)} > 0\}$, for $s \in \{-1, +1\}$ and $i \in [n]$, denoting the set of indices of neurons in positive or negative class (determined by $s$) which is activated by training data point $\boldsymbol{x}_i$ at time step $t$. Then, we apply Lemma C.1 to the gradient descent iteration (1), deriving the following result.

**Lemma C.3** (Updates of Coefficients $\lambda_{s,r,j}^{(t)}, \sigma_{s,r,i}^{(t)}$). *For each pair $s \in \{-1, +1\}, r \in [m], j \in [k], i \in [N]$ and time $t \geq 0$, we have the following update equations:*

$$\lambda_{s,r,j}^{(t+1)} = \lambda_{s,r,j}^{(t)} - \frac{s\eta}{nm} \cdot \sum_{i \in I_j} \ell_i'^{(t)} \|\boldsymbol{\mu_j}\|^2 \mathbb{1}\left(r \in S_{s,i}^{(t)}\right), \tag{2}$$

$$\sigma_{s,r,i}^{(t+1)} = \sigma_{s,r,i}^{(t)} - \frac{s\eta}{nm} \cdot \ell_i'^{(t)} \|\boldsymbol{\xi_i}\|^2 \mathbb{1}\left(r \in S_{s,i}^{(t)}\right), \tag{3}$$

*where $\ell_i'^{(t)} := \ell'(y_i f_{\boldsymbol{\theta}^{(t)}}(\boldsymbol{x}_i))$ denotes the point-wise loss derivative at point $\boldsymbol{x}_i$, and $I_j := \{i \in [n] : \boldsymbol{x}_i$ in cluster $j\}$ denotes the set of the training points in the $j$-th cluster.*

According to equations (2) and (3) from Lemma C.3, we know

$$\lambda_{s,r,j}^{(t)} = \sum_{i \in I_j} \frac{\|\boldsymbol{\xi}_i\|^2}{\|\boldsymbol{\mu}_j\|^2} \sigma_{s,r,i}^{(t)} \approx \sum_{i \in I_j} \sigma_{s,r,i}^{(t)}, \tag{4}$$

where we also use $\lambda_{s,r,j}^{(0)} = \sigma_{s,r,i}^{(0)} = 0$ and the fact that, w.h.p., we have $\|\boldsymbol{\xi}_i\| \approx \sqrt{d} = \|\boldsymbol{\mu}_j\|$. It suggests that we only need to focus on the dynamics of the noise coefficients $\sigma_{s,r,i}^{(t)}$ (i.e., equation (3)).

## C.2 Two Key Techniques about Loss Derivative and Activation Region

It seems that the main difficulty in analyzing the iteration (3) is addressing the time-variant loss derivative $\ell_i'^{(t)}$ and ReLU activation region $S_{s,i}^{(t)}$. To overcome these two challenges, we provide two corresponding key techniques (Lemma C.4 and Lemma C.6) as follows, which can usefully simplify the analysis of noise coefficients' dynamics.

**Key Technique 1: Bounding Loss Derivative Ratio.** We will establish the connection between loss derivative ratio $y_i \ell_i'^{(t)} / y_j \ell_j'^{(t)}$ and the training data margin gap $\Delta_q^{(t)}(i,j) := q_i^{(t)} - q_j^{(t)}$, where $q_i^{(t)}$ denotes the margin of the $i$-th training data at iteration $t$ defined as $q_i^{(t)} := y_i f_{\boldsymbol{\theta}^{(t)}}(\boldsymbol{x}_i)$. Then, we have:

**Lemma C.4** (Training data margins are balanced during training dynamics). *There exists a time threshold $T_0$ such that, for any time $1 \leq t \leq T_0$ and distinct data points $(\boldsymbol{x}_i, \boldsymbol{x}_j)$, it holds that*

$$\Delta_q^{(t)}(i,j) \leq \epsilon(k), \tag{5}$$

*where we use $\epsilon(k)$ to denote a time-independent error term satisfying $\epsilon(k) \to 0$ as $k \to \infty$.*

According to Lemma C.4, for any distinct training data points $\boldsymbol{x}_i$ and $\boldsymbol{x}_j$ with the same label, the loss derivative ratio can be bounded as:

$$y_i \ell_i'^{(t)} / y_j \ell_j'^{(t)} \approx \exp(\Delta_q^{(t)}(j,i)) \approx 1 + \Delta_q^{(t)}(j,i) \stackrel{(5)}{=} 1 \pm o(1), \tag{6}$$

where the first approximation holds due to $\ell'(z) = 1/(1 + \exp(z))$ and we use the fact $e^z \approx 1 + z$ for small $z$ in the second approximation.

**Remark C.5.** *This method was initially proposed by Chatterji & Long (2021) in the context of benign overfitting for linear classification and was subsequently extended to networks with non-linear activation (Frei et al., 2022; Kou et al., 2023a). In this paper, we extend the auto-balance technique of Kou et al. (2023a) from the single-feature case to our multi-cluster scenario to prove Lemma C.4.*

**Key Technique 2: Analyzing ReLU Activation Regions.** Then, we turn to the analysis of the activation regions $S_{s,i}^{(t)}$. In fact, after the first gradient descent update, the set of activated neurons can be described in the following lemma.

**Lemma C.6** (Each training data can activate all its corresponding neurons)**.** *For the same time threshold $T_0$ as that in Lemma C.4 and all time $1 \leq t \leq T_0$, it holds that $S_{1,i}^{(t)} = [m]$ for all $i \in I_+$ and $S_{-1,i}^{(t)} = [m]$ for all $i \in I_-$, where $I_+ := \{i : i \in I, y_i = 1\}$ and $I_- := \{i : i \in I, y_i = -1\}$.*

We rewrite our model as $f_{\boldsymbol{\theta}^{(t)}} = f_1^{(t)} + f_{-1}^{(t)}$, where $f_s^{(t)} := \frac{s}{m} \sum_{r \in [m]} \mathrm{ReLU}(\langle \boldsymbol{w}_{s,r}^{(t)}, \boldsymbol{x} \rangle + b_{s,r}^{(t)})$, $s \in \{-1, 1\}$. Then, Lemma C.6 manifests that $f_s^{(t)}$ is linear in the training data point $(\boldsymbol{x}_i, y_i)$ with label $y_i = s$. If we show $f_{-y_i}^{(t)}$ keeps small, we will have the linearization $f_{\boldsymbol{\theta}^{(t)}} \approx f_{y_i}^{(t)} = \frac{y_i}{m} \sum_{r \in [m]} (\langle \boldsymbol{w}_{y_i,r}^{(t)}, \boldsymbol{x}_i \rangle + b_{y_i,r}^{(t)})$, which allows us to approximate the data margin by noise coefficients (applying Lemma C.1 and (4)).

**Remark C.7.** *Indeed, we use induction to prove Lemma C.4 and Lemma C.6 together (see Lemma E.7 in the appendix), where we show the case when $t = 1$ by our small initialization assumption and use the auto-balance technique to complete the inductive step (see the full proof in Appendix E.1).*

## C.3   PROOF SKETCH OF THEOREM C.2

Now, based on the two key techniques above, we provide a proof sketch of Theorem C.2, which consists of five steps.

**Step 1: Proving that feature coefficient ratio $\lambda_{s_1,r_1,j_1}^{(T)} / \lambda_{s_2,r_2,j_2}^{(T)} (j_1 \in J_{s_1}, j_2 \in J_{s_2})$ is close to 1.**

By Lemma C.6, for all $s \in \{-1, 1\}, i \in I_s$, we know

$$\sigma_{s,r,i}^{(t+1)} = \sigma_{s,r,i}^{(t)} - \frac{s\eta}{nm} \ell_i'^{(t)} \|\boldsymbol{\xi}_i\|^2. \tag{7}$$

Combined with the loss derivative ratio bound (6), it furthermore implies that the noise coefficient ratio is close to 1, i.e., for any $r_1, r_2 \in [m], i_1, i_2 \in I$, we have

$$\sigma_{y_{i_1},r_1,i_1}^{(t)} / \sigma_{y_{i_2},r_2,i_2}^{(t)} \stackrel{(7)}{\approx} \sum_{t'=0}^{t} \ell_{i_1}'^{(t')} / \sum_{t'=0}^{t} \ell_{i_2}'^{(t')} \stackrel{(6)}{\approx} 1 \pm o(1). \tag{8}$$

Thus, for any $s_1, s_2 \in \{-1, 1\}, j_1 \in J_{s_1}, j_2 \in J_{s_2}$ and time $t \leq T$, we can derive

$$\lambda_{s_1,r_1,j_1}^{(t)} / \lambda_{s_2,r_2,j_2}^{(t)} \stackrel{(4)}{\approx} \sum_{i_1 \in I_{j_1}} \sigma_{s_1,r_1,i_1}^{(t)} / \sum_{i_2 \in I_{j_2}} \sigma_{s_2,r_2,i_2}^{(t)} \stackrel{(8)}{\approx} |I_{j_1}| / |I_{j_2}| = 1 \pm o(1).$$

**Step 2: Proving that $\lambda_{s,r,j}^{(T)}$ attains $\Omega(1)$ for $j \in J_s$, and keeps $o(1)$ for $j \in J_{-s}$.**

By induction, we can show that both bias terms $b_{s,r}^{(t)}$ and $\lambda_{s,r,j}^{(t)} (j \in J_{-s})$ keep $o(1)$-order during the learning process (Lemma E.10 and Corollary E.16), which thereby implies the following approximation, i.e., for any $s \in \{-1, 1\}, r \in [m], i \in I_j$, we have

$$\langle \boldsymbol{w}_{s,r}^{(t)}, \boldsymbol{x}_i \rangle + b_{s,r}^{(t)} \approx \lambda_{s,r,j}^{(t)} + \sigma_{s,r,i}^{(t)}, \tag{9}$$

where we also need time $t$ satisfying $t = \exp(\tilde{O}(k^{0.5}))$ (see details in Lemma E.13).

Then, for any $s \in \{-1, 1\}, i \in I_j$ and data point $(\boldsymbol{x}_i, y_i)$ satisfying $y_i = -s$, we know

$$\text{ReLU}(\langle \boldsymbol{w}_{s,r}^{(t)}, \boldsymbol{x}_i \rangle + b_{s,r}^{(t)}) \overset{(9)}{=} \text{ReLU}(\lambda_{s,r,j}^{(t)} + \sigma_{s,r,i}^{(t)} + o(1)) \leq o(1), \tag{10}$$

where the last inequality holds due to $\lambda_{s,r,j}^{(t)}, \sigma_{s,r,i}^{(t)} \leq 0$ (Lemma E.5).

Next, we approximate the model output for training data point $(\boldsymbol{x}_i, y_i)$ belonging to the $j$-th cluster as

$$f_{\boldsymbol{\theta}^{(t)}}(\boldsymbol{x}_i) = \sum_{s \in \{-1,1\}} \sum_{r \in [m]} \frac{s}{m} \text{ReLU}(\langle \boldsymbol{w}_{s,r}^{(t)}, \boldsymbol{x}_i \rangle + b_{s,r}^{(t)}) \overset{(10)}{\approx} \frac{y_i}{m} \sum_{r \in [m]} (\langle \boldsymbol{w}_{y_i,r}^{(t)}, \boldsymbol{x}_i \rangle + b_{y_i,r}^{(t)})$$

$$\overset{(9)}{\approx} \frac{y_i}{m} \sum_{r \in [m]} (\lambda_{y_i,r,j}^{(t)} + \sigma_{y_i,r,i}^{(t)}) \overset{(4)}{\approx} \frac{y_i}{m} \sum_{r \in [m]} \left( \sum_{i' \in I_j} \sigma_{y_i,r,i'}^{(t)} + \sigma_{y_i,r,i}^{(t)} \right) \overset{(8)}{\approx} y_i(|I_j| + 1)\sigma_{y_i,1,i}^{(t)}. \tag{11}$$

Therefore, we derive the following approximate update w.r.t. $\sigma_{y_i,1,i}^{(t)}$, i.e., for any iteration $t \in [0, \exp(\tilde{O}(k^{1/2}))]$, we have $\sigma_{y_i,1,i}^{(t+1)} \overset{(7)(11)}{\approx} \sigma_{y_i,1,i}^{(t)} + \frac{\eta d}{nm} \exp\left(-\frac{n}{k} \sigma_{y_i,1,i}^{(t)}\right)$ (Lemma E.21). By leveraging $\log(z + 1) - \log(z) \approx \frac{1}{z}$, we inductively prove $\sigma_{y_i,1,i}^{(t)} \approx \frac{k}{n} \log(\eta t)$ (Lemma E.7) and $\lambda_{s,r,j}^{(t)} \approx \log(\eta t), j \in J_s$ (Lemma E.23). When the assumption $T = \Omega(\eta^{-1})$ holds, we have $\lambda_{s,r,j}^{(T)} = \Omega(1), j \in J_s$.

**Step 3: Gradient descent leads the network to the feature-averaging regime.**

We can choose $\lambda^{(T)} = \lambda_{1,1,j_0}^{(T)}$ for some $j_0 \in J_+$ as the representative of $\Lambda^{(T)} = \{\lambda_{s,r,j}^{(T)} : s \in \{-1, +1\}, r \in [m], j \in J_s\}$.

Combining the result in **Step 1 and Step 2**, we know that for any $\lambda_{s,r,j}^{(T)} \in \Lambda^{(T)}$,

$$\lambda^{(T)} \approx \lambda_{s,r,j}^{(T)} \approx \log(\eta T).$$

We can also prove that the $\boldsymbol{w}_{s,r}^{(T)}$ is minimally affected by the coefficient $\sigma_{s,r,i}^{(T)}$ in weight decomposition. Thus, we have (Lemma E.25)

$$\left\| \boldsymbol{w}_{s,r}^{(T)} - \lambda^{(T)} \sum_{j \in J_s} \|\boldsymbol{\mu}_j\|^{-2} \boldsymbol{\mu}_j \right\| \leq o(d^{-1/2})$$

.

**Step 4: Proving that the clean accuracy is perfect.**

For a randomly-sampled test data point $(\boldsymbol{x} = \boldsymbol{\mu}_j + \boldsymbol{\xi}, y) \sim \mathcal{D}$ within cluster $j \in J_y$, we can prove that, with probability at least $1 - \exp(\Omega(\log^2 d))$, it holds that $|\langle \boldsymbol{w}_{s,r}^{(T)}, \boldsymbol{\xi} \rangle| = o(1)$ for all $s \in \{\pm 1\}, r \in [m]$ (Lemma E.26). Then, for data satisfying the above condition, we can calculate the data margin as $y f_{\boldsymbol{\theta}^{(T)}}(\boldsymbol{x}) \approx \frac{1}{m} \sum_{r \in [m]} \lambda_{s,r,j}^{(T)} = \Omega(1) > 0$, which implies that $\text{Acc}_{\text{clean}}^{\mathcal{D}}(f_{\boldsymbol{\theta}^{(T)}}) \geq 1 - \exp(-\Omega(\log^2 d))$.

**Step 5: Proving that the robust accuracy is poor.**

We consider the perturbation $\boldsymbol{\rho} = -\frac{2(1+c)}{k} \left( \sum_{j \in J_+} \boldsymbol{\mu}_j - \sum_{j \in J_-} \boldsymbol{\mu}_j \right)$. By applying Lemma E.26 again, we can derive that $\text{sgn}(f_{\boldsymbol{\theta}^{(T)}}(\boldsymbol{x} + \boldsymbol{\rho})) \neq \text{sgn}(f_{\boldsymbol{\theta}^{(T)}}(\boldsymbol{x}))$, which means $\text{Acc}_{\text{robust}}^{\mathcal{D}}(f_{\boldsymbol{\theta}^{(T)}}; 2(1+c)\sqrt{d/k}) \leq \exp(-\Omega(\log^2 d))$ and finishes the proof of Theorem C.2.

# D  PRELIMINARY PROPERTIES

In this section, we provide some useful properties of our training dataset and neural network learner at the initialization. These properties hold with high probability under our assumptions. Our subsequent proofs will be based on the validity of these properties. The proofs of the inlined claims are concluded with the ∎ symbol, while the proofs of the overarching results are concluded with the □ symbol.

## D.1  DETAILED DATA MODEL AND ASSUMPTIONS

First, we recall the definition of multi-class feature data distribution that we defined in Section 3.

**Definition D.1** (Multi-Cluster Data Distribution). Given $k$ vectors $\boldsymbol{\mu}_1, \ldots, \boldsymbol{\mu}_k \in \mathbb{R}^d$, called the *cluster features*, and a partition of $[k]$ into two disjoint sets $J_\pm = (J_+, J_-)$, we define $\mathcal{D}(\{\boldsymbol{\mu}_j\}_{j=1}^k, J_\pm)$ as a data distribution on $\mathbb{R}^d \times \{-1, 1\}$, where each data point $(\boldsymbol{x}, y)$ is generated as follows:

1. Draw a cluster index as $j \sim \mathrm{Unif}([k])$;

2. Set $y = +1$ if $j \in J_+$; otherwise $j \in J_-$ and set $y = -1$;

3. Draw $\boldsymbol{x} := \boldsymbol{\mu}_j + \boldsymbol{\xi}$, where $\boldsymbol{\xi} \sim \mathcal{N}(\mathbf{0}, \boldsymbol{I}_d)$.

For convenience, we write $\mathcal{D}$ instead of $\mathcal{D}(\{\boldsymbol{\mu}_j\}_{j=1}^k, J_\pm)$ if $\{\boldsymbol{\mu}_j\}_{j=1}^k$ and $J_\pm$ are clear from the context. For $s \in \{\pm 1\}$, we write $J_s$ to denote $J_+$ if $s = +1$ and $J_-$ if $s = -1$.

To ease the analysis, we make the following simplifying assumptions on the distribution.

**Assumption D.2** (Orthogonal Equinorm Cluster Features). The cluster features $\{\boldsymbol{\mu}_j\}_{j=1}^k$ satisfy the properties that (1) $\|\boldsymbol{\mu}_j\| = \sqrt{d}$ for all $j \in [k]$; and (2) $\boldsymbol{\mu}_i \perp \boldsymbol{\mu}_j$ for all $1 \leq i < j \leq k$.

**Assumption D.3** (Nearly Balanced Classification). The partition $J_\pm$ satisfies $c^{-1} \leq \frac{|J_+|}{|J_-|} \leq c$ for some absolute constant $c \geq 1$.

Next, we summarize the assumptions of these hyper-parameters that we mentioned in the main text, as listed below.

**Assumption D.4** (Choices of Hyper-Parameters). We state the range of parameters for our proofs in the appendix to hold.

- $d = \Omega(k^{10})$                                       (recall $d$ is the data dimension)

- $c = \Theta(1)$                                            (recall $c$ is the balance ratio)

- $n \in [\Omega(k^7), \exp(O(\log^2(d)))]$                    (recall $n$ is the number of samples)

- $m = \Theta(k)$                           (recall $2m$ is the width of network learner)

- $\eta \leq O(d^{-2})$                                   (recall $\eta$ is the learning rate)

- $\sigma_b^2 = \sigma_w^2 \leq \eta k^{-5}$,     (recall $\boldsymbol{w}_{s,r}^{(0)} \sim \mathcal{N}(\mathbf{0}, \sigma_w^2 \boldsymbol{I}_d), b_{s,r}^{(0)} \sim \mathcal{N}(0, \sigma_b^2)$ give the initialization)

**Remark D.5** (Discussion of Hyper-Parameter Choices). *In this paper, we make specific choices of hyper-parameters for the sake of calculations (and we emphasize that these may not be the tightest possible choices), which is a widely-applied simplicity in the literature of feature learning works (Wen & Li, 2021; Chen et al., 2022; Allen-Zhu & Li, 2022; 2023; Chidambaram et al., 2023). Namely, we need the data dimension $d$ to be a significantly larger polynomial in the number of clusters $k$ to ensure all $k$ cluster features $\boldsymbol{\mu}_1, \boldsymbol{\mu}_2, \ldots, \boldsymbol{\mu}_k$ can be orthogonal within the space $\mathbb{R}^d$. The balance ratio $c$ is an absolute constant that is independent with $d$ and $k$. Our results can be extended to $\boldsymbol{x} := \alpha\boldsymbol{\mu}_j + \sigma\boldsymbol{\xi}$ for some parameters $\alpha = \Theta(1)$ and $\sigma = \Theta(1)$, but here we set $\alpha = \sigma = 1$ for simplicity. And we further require that $n$ is a large polynomial in $k$ due to our choice of large signal-noise-ratio (recall that we have $\alpha/\sigma = \Theta(1)$, which implies that $\|\boldsymbol{\mu}_j\| = \sqrt{d} \approx \|\boldsymbol{\xi}\|$ with high probability). We need $m \in [\max\{|J_+|, |J_-|\}, k]$ for the existence of robust solution (Theorem G.3). We assume the learning rate $\eta$ and the initialization magnitude $\sigma_w, \sigma_b$ are sufficiently small, which helps the network to be trained in the feature learning regime (Lyu et al., 2021; Cao et al., 2022; Allen-Zhu & Li, 2023; Kou et al., 2023a).*

### D.2 USEFUL PROPERTIES OF THE TRAINING DATASET

Now, we introduce some useful notations, for simplifying our proof.

- Denote $I = \{i : i = 1, 2, \cdots, n\}$ as the set of indices of all training data points.
- Define $c(\cdot)$ as the map $I \to J$ where $c(i)$ represents the index of the cluster to which point $\boldsymbol{x}_i$ belongs.
- $I_j = \{i : i \in I, c(i) = j\}$ denotes the set of the training points in the $j$-th cluster.
- $I_+ = \{i : i \in I, c(i) \in J_+\}$ and $I_- = \{i : i \in I, c(i) \in J_-\}$ denote the index sets of all positive class data points and negative class data points respectively.

Then, under Assumption D.2,D.3,D.4, we show that Proposition D.6 of the training dataset hold with high probability. It is worth noting that most of the proofs of Proposition D.6 follow standard methodologies and closely resemble those presented in Frei et al. (2024), as our data distribution is similar to theirs.

Recall the cumulative distribution function (CDF) of the standard normal distribution, usually denoted as $\Phi(x)$, which is defined as the integral

$$\Phi(x) := \frac{1}{\sqrt{2\pi}} \int_{-\infty}^{x} e^{-t^2/2} dt = \frac{1}{\sqrt{2\pi}} \int_{-x}^{\infty} e^{-t^2/2} dt.$$

Additionally, we have the following commonly used bounds on the tail probabilities of the standard normal distribution.

$$\left( \frac{x^2 - 1}{\sqrt{2\pi} x^3} \right) \exp(-x^2/2) \leq 1 - \Phi(x) \leq \frac{1}{\sqrt{2\pi} x} \exp(-x^2/2) \qquad (12)$$

**Proposition D.6.** *Let $\Delta = 4\sqrt{d} \ln(d)$ and $\delta = 8n^2 d^{-\frac{\ln(d)}{2}} + 2k \left(n/k\right)^{-\ln(n/k)}$. With probability at least $1 - \delta$ over sampled training dataset $\mathcal{S} \sim \mathcal{D}^n$, we have the following properties:*

1. *For every $i \in I$ we have $\sqrt{d} - 2\ln(d) \leq \|\boldsymbol{\xi}_i\| \leq \sqrt{d} + 2\ln(d)$.*

2. *For every $i \in I$ we have $\|\boldsymbol{x}_i\| \leq 3\sqrt{d}$.*

3. *For every $i, j \in I, i \neq j$ we have $|\langle \boldsymbol{\xi}_i, \boldsymbol{\xi}_j \rangle| \leq \Delta$.*

4. *For every $i \in I$ and $j \in J$ we have $|\langle \boldsymbol{\mu}_j, \boldsymbol{\xi}_i \rangle| \leq \Delta$.*

5. *For every $i, j \in I$ with $c(i) \neq c(j)$ we have $|\langle \boldsymbol{x}_i, \boldsymbol{x}_j \rangle| \leq \Delta$.*

6. *For every $i, j \in I$ with $c(i) = c(j)$ we have $\frac{1}{2}d \leq \langle \boldsymbol{x}_i, \boldsymbol{x}_j \rangle \leq 2d$.*

7. *For every $j \in J$ we have $\frac{n}{k} - \sqrt{\frac{2n}{k}} \ln(\frac{n}{k}) \leq |I_j| \leq \frac{n}{k} + \sqrt{\frac{2n}{k}} \ln(\frac{n}{k})$.*

**Remark D.7.** *Property 1 and Property 2 show that the data-wise noise and training data point are bounded, i.e., $\|\boldsymbol{\xi}_i\| \approx \sqrt{d}$ and $\|\boldsymbol{x}_i\| = O(\sqrt{d})$. Property 3 and Property 4 show that the correlation between different noises (or between cluster center feature and random noise) is very small. Property 5 and Property 6 suggest that the correlation between training data points of different clusters is very small, but the correlation between training data points within the same cluster is very large. Property 7 manifests that the training dataset $\mathcal{S}$ approximately includes $\frac{n}{k} (= \Omega(k^6))$ examples from each cluster.*

*Proof of Proposition D.6.* Now, we prove Property 1-7 one by one.

**Property 1:** We notice that $\|\boldsymbol{\xi}\|^2$ follows the Chi-squared distribution.

The concentration bound in Lemma 1 by Laurent & Massart (2000) implies that for all $t \geq 0$, we have

$$\Pr\left[ \|\boldsymbol{\xi}\|^2 - d \geq 2\sqrt{dt} + 2t \right] \leq e^{-t},$$
$$\Pr\left[ \|\boldsymbol{\xi}\|^2 - d \leq -2\sqrt{dt} \right] \leq e^{-t}.$$

Plugging in $t = \ln^2(d)$, we can see that

$$\Pr\left[\|\boldsymbol{\xi}\|^2 \geq (\sqrt{d} + 2\ln(d))^2\right] \leq \Pr\left[\|\boldsymbol{\xi}\|^2 - d \geq 2\sqrt{d}\ln(d) + 2\ln^2(d)\right] \leq d^{-\ln(d)},$$

$$\Pr\left[\|\boldsymbol{\xi}\|^2 \leq (\sqrt{d} - 2\ln(d))^2\right] \leq \Pr\left[\|\boldsymbol{\xi}\|^2 - d \leq -2\sqrt{d}\ln(d)\right] \leq d^{-\ln(d)}.$$

Thus, we have

$$\Pr\left[\sqrt{d} - 2\ln(d) \leq \|\boldsymbol{\xi}_i\| \leq \sqrt{d} + 2\ln(d)\right] \leq 2d^{-\ln(d)}. \tag{13}$$

Then by union bound, we have Property 1 holds for every $i \in I$ with probability at least $1 - 2nd^{-\ln(d)}$.

**Property 2:** When Property 1 holds, by triangle inequality, we know Property 2 holds:

$$\|\boldsymbol{x}_i\| \leq \|\boldsymbol{\mu}_{c(i)}\| + \|\boldsymbol{\xi}_i\| \leq \sqrt{d} + \sqrt{d} + \ln(d) \leq 3\sqrt{d}.$$

The proofs for other properties require calculations pertaining to Gaussian distribution. We first introduce a useful lemma below.

**Lemma D.8.** *Let* $\boldsymbol{\xi} \sim \mathrm{N}(\boldsymbol{0}, I_d)$. *For any* $\boldsymbol{x} \in \mathbb{R}^d$ *we have*

$$\Pr\left[|\langle \boldsymbol{x}, \boldsymbol{\xi} \rangle| \geq \|\boldsymbol{x}\|\ln(d)\right] \leq 2d^{-\frac{\ln(d)}{2}}.$$

*Proof of Lemma D.8.* Note that $\left\langle \dfrac{\boldsymbol{x}}{\|\boldsymbol{x}\|}, \boldsymbol{\xi} \right\rangle$ has the distribution $\mathcal{N}(0, 1)$.

By standard Gaussian tail bound, we have for every $t \geq 0$ that $\Pr\left[\left|\left\langle \frac{\boldsymbol{x}}{\|\boldsymbol{x}\|}, \boldsymbol{\xi} \right\rangle\right| \geq t\right] \leq 2\exp\left(-\dfrac{t^2}{2}\right)$.

Plugging in $t = \ln(d)$, we can see that

$$\Pr\left[\left|\left\langle \frac{\boldsymbol{x}}{\|\boldsymbol{x}\|}, \boldsymbol{\xi} \right\rangle\right| \geq \ln(d)\right] \leq 2\exp\left(-\dfrac{\ln^2(d)}{2}\right) = 2d^{-\frac{\ln(d)}{2}}.$$

$\blacksquare$

**Property 3:** Next, we prove Property 3 using the result in Lemma D.8.

Noting that if $|\langle \boldsymbol{\xi}_i, \boldsymbol{\xi}_j \rangle| \geq \sqrt{2d}\ln(d)$, we have that at least one of the following holds:

1. $\|\boldsymbol{\xi}_i\| \geq \sqrt{2d}$;

2. $\left|\left\langle \frac{\boldsymbol{\xi}_i}{\|\boldsymbol{\xi}_i\|}, \boldsymbol{\xi}_j \right\rangle\right| \geq \ln(d)$.

Now we bound the probabilities of these two events separately. By Property 1, we have

$$\Pr[\|\boldsymbol{\xi_i}\| \geq \sqrt{2d}] \leq 2d^{-\ln(d)}.$$

Next, by Lemma D.8, we have

$$\Pr\left[\left|\left\langle \frac{\boldsymbol{\xi}_i}{\|\boldsymbol{\xi}_i\|}, \boldsymbol{\xi}_j \right\rangle\right| \geq \ln(d)\right] \leq 2d^{-\frac{\ln(d)}{2}}.$$

Then, by union bound, we know that,

$$\Pr\left[|\langle \boldsymbol{\xi}_i, \boldsymbol{\xi}_j \rangle| \geq \sqrt{2d}\ln(d)\right] \leq 2d^{-\frac{\ln(d)}{2}} + 2d^{-\ln(d)} \leq 4d^{-\frac{\ln(d)}{2}}$$

Then, applying the union bound for all pairs $i, j \in I, i \neq j$, we have $|\langle \boldsymbol{\xi}_i, \boldsymbol{\xi}_j \rangle| \leq \sqrt{2d}\ln(d)^2$ holds with probability at least $1 - 4n^2 d^{-\frac{\ln(d)}{2}}$.

**Property 4:** Applying Lemma D.8, we have

$$\Pr\left[|\langle \boldsymbol{\mu_j}, \boldsymbol{\xi}\rangle| \geq \sqrt{d}\ln(d)\right] \leq 2d^{-\frac{\ln(d)}{2}}.$$

Then for all pairs $i \in I, j \in J$, applying union bound we have that $|\langle \boldsymbol{\mu_j}, \boldsymbol{\xi_i}\rangle| \leq \sqrt{d}\ln(d)$ for all $i \in I, j \in J$ holds with probability at least $1 - 2n^2 d^{-\frac{\ln(d)}{2}}$.

**Property 5:** By using the results above, we have

$$
\begin{aligned}
|\langle \boldsymbol{x}_i, \boldsymbol{x}_j\rangle| &\leq \left|\langle \boldsymbol{\mu}_{c(i)}, \boldsymbol{\mu}_{c(j)}\rangle\right| + \left|\langle \boldsymbol{\mu}_{c(i)}, \boldsymbol{\xi}_j\rangle\right| + \left|\langle \boldsymbol{\mu}_{c(j)}, \boldsymbol{\xi}_i\rangle\right| + |\langle \boldsymbol{\xi}_i, \boldsymbol{\xi}_j\rangle| \\
&= \left|\langle \boldsymbol{\mu}_{c(i)}, \boldsymbol{\xi}_j\rangle\right| + \left|\langle \boldsymbol{\mu}_{c(j)}, \boldsymbol{\xi}_i\rangle\right| + |\langle \boldsymbol{\xi}_i, \boldsymbol{\xi}_j\rangle| \\
&\leq 4\sqrt{d}\ln(d) = \Delta.
\end{aligned}
$$

**Property 6:** By using the results above and noting that $\langle \mu_{c(i)}, \mu_{c(j)}\rangle = d$ for $i, j$ with $c(i) = c(j)$, we have

$$
\begin{aligned}
|\langle \boldsymbol{x}_i, \boldsymbol{x}_j\rangle - d| &\leq \left|\langle \boldsymbol{\mu}_{c(i)}, \boldsymbol{\xi}_j\rangle\right| + \left|\langle \boldsymbol{\mu}_{c(j)}, \boldsymbol{\xi}_i\rangle\right| + |\langle \boldsymbol{\xi}_i, \boldsymbol{\xi}_j\rangle| \\
&\leq 4\sqrt{d}\ln(d) = \Delta.
\end{aligned}
$$

Thus, we have

$$\frac{1}{2}d \leq \langle \mathbf{x}_i, \mathbf{x}_j\rangle \leq 2d.$$

**Property 7:** We define $X_i$ for $i \in I$ as the indicator random variable that the $i$-th point is in the $j$-th cluster. It takes value 1 with probability $\frac{1}{k}$, and 0 with probability $1 - \frac{1}{k}$.

Then we know that $|I_j| = \sum\limits_{i \in I} X_i$. Applying Chernoff bound, we have that

$$\Pr\left[\frac{n}{k} - \sqrt{\frac{2n}{k}}\ln(\frac{n}{k}) \leq |I_j| \leq \frac{n}{k} + \sqrt{\frac{2n}{k}}\ln(\frac{n}{k})\right] \geq 1 - 2\exp(-\ln^2(n/k)) = 1 - 2(n/k)^{-\ln(n/k)}.$$

Then for all $j \in J$, applying union bound, we have Property 7 holds with probability at least $1 - 2k(n/k)^{-\ln(n/k)}$.

Combining all of the above together, we have Proposition D.6 holds with probability at least $1 - 8n^2 d^{-\frac{\ln(d)}{2}} - 2k(n/k)^{-\ln(n/k)}$. $\qquad\square$

## D.3 USEFUL PROPERTIES OF THE NETWORK INITIALIZATION

The proofs for properties of the network initialization require the range of the maximum value obtained from multiple independent samples drawn from a Gaussian distribution. We first present the following useful lemma.

**Lemma D.9** (Concentration of Maximum of Gaussians). *Let $X_i \sim \mathcal{N}(0, 1), 1 \leq i \leq l$ be i.i.d. random variables. Denote $X_{max} = \max_{1 \leq i \leq l}\{X_i\}, X_{min} = \min_{1 \leq i \leq l}\{X_i\}$. For any $t \geq 0$, we have*

- $\Pr\left[X_{min} \leq -\sqrt{2\log(l)} - t\right] \leq \frac{1}{2}\exp\left(-t^2/2\right),$

- $\Pr\left[X_{max} \leq \sqrt{2\log(l) - t}\right] \leq \exp\left(-\frac{e^{t/2}}{\sqrt{2\pi}(\sqrt{2\log(l)}+1)}\right).$

*Proof of Lemma D.9.* The proof is standard and similar to Proposition A.1 and A.2 from Chidambaram et al. (2023). We include it for convenience of the readers.

For any $a \geq 0$, we have

$$\Pr\left[X_{\min} \geq -a\right] = (\Phi(a))^l = (1 - Q(a))^l,$$

where $Q(x) = 1 - \Phi(x)$. By using $(1-x)^n \geq 1 - nx$ for any $x \in [0,1]$ and $n \in \mathbb{N}$, we then get

$$\Pr[X_{\min} \geq -a] \geq 1 - lQ(a).$$

Now we use the elementary inequality for the tail of the normal distribution:

$$Q(a) \leq \frac{1}{2} e^{-\frac{a^2}{2}},$$

so that

$$\Pr[X_{\min} \geq -a] \geq 1 - \frac{1}{2} l e^{-\frac{a^2}{2}}.$$

Plugging $a = \sqrt{2\log(l)} + t$, we get

$$\Pr\left[X_{\min} \geq -\sqrt{2\log(l)} - t\right] \geq 1 - l \exp\left(-\frac{(\sqrt{2\log(l)} + t)^2}{2}\right)$$

$$= 1 - \frac{1}{2} l \exp\left(-\frac{2\log(2l) + t^2}{2}\right)$$

$$= 1 - \frac{1}{2} e^{-\frac{t^2}{2}}.$$

Similar to the previous proof, we know that

$$\Pr[X_{\max} \leq a] = (\Phi(a))^l \leq \left(1 - \frac{a}{\sqrt{2\pi}(a^2+1)} \exp\left(-\frac{a^2}{2}\right)\right)^l.$$

Plugging $a = \sqrt{2\log(l)} - t$,

$$\Pr\left[X_{\max} \leq \sqrt{2\log(l)} - t\right] \leq \left(1 - \frac{\sqrt{2\log(l)} - t}{\sqrt{2\pi}l(2\log(l) - t + 1)} \exp\left(\frac{t}{2}\right)\right)^l$$

$$\leq \left(1 - \frac{\exp(t/2)}{\sqrt{2\pi}l(\sqrt{2\log(2l)} + 1)}\right)^l$$

$$\leq \exp\left(-\frac{\exp(t/2)}{\sqrt{2\pi}(\sqrt{2\log(2l)} + 1)}\right)$$

$\square$

By applying Proposition D.6, we can derive the following result, which gives the range of network parameters at the initialization.

**Proposition D.10.** *With probability at least $1 - 4md^{-\ln(d)} - 2m^{-3}$, we have the following properties for our network initialization:*

- *For any $s \in \{-1, +1\}, r \in [m]$, we have $\sigma_w\left(\sqrt{d} - 2\ln(d)\right) \leq \|\boldsymbol{w}_{s,r}^{(0)}\| \leq \sigma_w\left(\sqrt{d} + 2\ln(d)\right)$.*

- *For any $s \in \{-1, +1\}, r \in [m]$, we have $|b_{s,r}^{(0)}| \leq 2\sigma_b\sqrt{2\ln(m)}$.*

*Proof of Proposition D.10.* For $\boldsymbol{w}_{s,r}^{(0)}$, reusing the same argument as the proof of Property (1) in Proposition D.6, we know that

$$\sigma_w\left(\sqrt{d} - 2\ln(d)\right) \leq \|\boldsymbol{w}_{s,r}^{(0)}\| \leq \sigma_w\left(\sqrt{d} + 2\ln(d)\right)$$

holds for all $s \in \{-1, +1\}, r \in [m]$ with probability at least $1 - 4md^{-\ln(d)}$.

For $b_{s,r}^{(0)}$, by standard Gaussian tail bound, we know that

$$\Pr\left[|b_{s,r}^{(0)}| \geq 2\sigma_b\sqrt{2\ln(m)}\right] \leq \exp\left(-4\ln(m)\right) = m^{-4}.$$

Then using union bound, we know that $|b_{s,r}^{(0)}| \geq 2\sigma_b\sqrt{2\ln(m)}$ holds for all $s \in \{-1,+1\}, r \in [m]$ with probability at least $1 - 2m^{-3}$.

In conclusion, we know that these properties hold with probability at least $1 - 2md^{-\ln(d)} - 2m^{-3}$. $\quad\square$

We then show that each neuron is activated by at least one training data point in each cluster upon initialization with high probability. We first formally define the notion of activation region as follows.

**Definition D.11** (Activation Region over Data Input). Let $T_{s,r,j} := \{i \in I_j : \langle \boldsymbol{w}_{s,r}^{(0)}, \boldsymbol{x}_i \rangle + b_{s,r}^{(0)} \geq 0\}$ be the set of indices of training data points in the $j$-th cluster which can activate the $r$-th neuron with weight $\boldsymbol{w}_{s,r}$ at time step 0.

Then, we give the following result about the activation region $T_{s,r,j}$.

**Proposition D.12.** *Assuming Proposition D.6 and Proposition D.10 holds. Then with probability at least $1 - m^{-0.01} - 2mk\exp\left(-\frac{n}{9km^2}\right)$, for all $s \in \{-1,+1\}, r \in [m], j \in J$, we have*

$$|T_{s,r,j}| \geq \frac{n}{3km^2}.$$

*Proof of Proposition D.12.* Given $\boldsymbol{\mu}_j$ for $j \in J$, we have $\langle \boldsymbol{w}_{s,r}^{(0)}, \boldsymbol{\mu}_j \rangle \sim \mathcal{N}(0, \sigma_w\sqrt{d})$.

Using the conclusion in Lemma D.9, with probability at least $1 - m^{0.01}$, we have

$$\min_{s \in \{-1,+1\}, r \in J} \langle \boldsymbol{w}_{s,r}^{(0)}, \boldsymbol{\mu}_j \rangle \geq -1.1\sigma_w\sqrt{d}\sqrt{2\ln(2m)}.$$

In the following proof, we assume that the above conclusion holds.

Given $\boldsymbol{w}_{s,r}^{(0)}$, we have $\langle \boldsymbol{w}_{s,r}^{(0)}, \boldsymbol{\xi}_i \rangle \sim \mathcal{N}(0, \|\boldsymbol{w}_{s,r}^{(0)}\|)$.

Then by Gaussian tail bound (12), we have

$$\begin{aligned}
\Pr\left[\langle \boldsymbol{w}_{s,r}^{(0)}, \boldsymbol{\xi}_i \rangle \geq 1.2\sigma_w\sqrt{d}\sqrt{2\ln(2m)}\right] &= 1 - \Phi\left(\frac{1.2\sigma_w\sqrt{d}\sqrt{2\ln(2m)}}{\|w_{s,r}^{(0)}\|}\right) \\
&\geq 1 - \Phi(\sqrt{3\ln(2m)}) \\
&\geq \frac{1}{2\sqrt{3\ln(2m)}}\exp\left(-\frac{3\ln(2m)}{2}\right) \\
&\geq m^{-2}.
\end{aligned}$$

We denote $X_{s,r,i} = \mathbb{1}\left(\langle \boldsymbol{w}_{s,r}^{(0)}, \boldsymbol{\xi}_i \rangle \geq 1.2\sigma_w\sqrt{d}\sqrt{2\ln(2m)}\right), T'_{s,r,j} = \sum_{i \in I_j} X_{s,r,i},$

$Y_{s,r,i} = \mathbb{1}\left(\langle \boldsymbol{w}_{s,r}^{(0)}, \boldsymbol{x}_i \rangle + b_{s,r}^{(0)} \geq 0\right)$ and we know that $|T_{s,r,j}| = \sum_{i \in I_j} Y_{s,r,i}$.

If $\langle \boldsymbol{w}_{s,r}^{(0)}, \boldsymbol{\xi}_i \rangle \geq 1.2\sigma_w\sqrt{d}\sqrt{2\ln(2m)}$, then

$$\begin{aligned}
\langle \boldsymbol{w}_{s,r}^{(0)}, \boldsymbol{x}_i \rangle + b_{s,r}^{(0)} &\geq \min_{s \in \{-1,+1\}, r \in J} \langle \boldsymbol{w}_{s,r}^{(0)}, \boldsymbol{\mu}_p \rangle + \langle \boldsymbol{w}_{s,r}^{(0)}, \boldsymbol{\xi}_i \rangle - 2\sigma_b\sqrt{2\ln(m)} \\
&\geq 0.1\sigma_w\sqrt{d}\sqrt{2\ln(2m)} - 2\sigma_b\sqrt{2\ln(m)} \geq 0.
\end{aligned}$$

That is to say $Y_{s,r,i} = 1$ if $X_{s,r,i} = 1$. Thus

$$\Pr\left[|T_{s,r,j}| \geq \frac{n}{2m^2k}\right] \geq \Pr\left[T'_{s,r,j} \geq \frac{n}{2m^2k}\right]$$

For any given $s \in \{-1, +1\}, r \in [m]$ and $i \in I_j$, we know that $X_{s,r,i}$ are i.i.d. and $\mathbb{E}[X_{s,r,i}] \geq m^{-2}$. Then by Chernoff bound, we have

$$\Pr\left[T'_{s,r,j} \geq \frac{|I_j|}{2m^2}\right] \geq 1 - \exp\left(\frac{|I_j|}{8m^2}\right) \geq 1 - \exp\left(\frac{n}{9km^2}\right)$$

Then

$$
\begin{aligned}
\Pr\left[|T_{s,r,j}| \geq \frac{n}{3km^2}\right] &\geq \Pr\left[T'_{s,r,j} \geq \frac{n}{3km^2}\right] \\
&\geq \Pr\left[T'_{s,r,j} \geq \frac{|I_j|}{2m^2}\right] \\
&\geq 1 - \exp\left(\frac{n}{9km^2}\right).
\end{aligned}
$$

Then by union bound, we know that $|T_{s,r,j}| \geq \dfrac{n}{3km^2}$ holds for all $s \in \{-1, +1\}, r \in [m], p \in J$ with probability at least $1 - 2mk\exp\left(\dfrac{n}{9km^2}\right)$.

Combing the above together, with probability at least $1 - m^{-0.01} - 2mk\exp\left(\dfrac{n}{9km^2}\right)$, we have $|T_{s,r,j}| \geq \dfrac{n}{3km^2}$. $\qquad\square$

Next, we show that the pre-activation output of the network is very small, at the initialization.

**Lemma D.13.** *For any $i \in I, r \in [m], s \in \{-1, +1\}$, we have $|\langle \boldsymbol{w}_{s,r}^{(0)}, \boldsymbol{x}_i \rangle + b_{s,r}^{(0)}| \leq \dfrac{\eta\sqrt{d}}{nm}$.*

*Proof of Lemma D.13.* By Property (2) in Proposition D.6 and Lemma D.8, we have

$$\|\boldsymbol{x}_i\| \leq 3\sqrt{d}, \|\boldsymbol{w}_{s,r}^{(0)}\| \leq \sigma_w\left(\sqrt{d} + 2\ln(d)\right), \|b_{s,r}^{(0)}\| \leq 2\sigma_b\sqrt{2\ln(2m)}.$$

Therefore, by triangle inequality, we know that

$$|\langle \boldsymbol{w}_{s,r}^{(0)}, \boldsymbol{x}_i \rangle + b_{s,r}^{(0)}| \leq \|\boldsymbol{x}_i\|\|\boldsymbol{w}_{s,r}^{(0)}\| + \|b_{s,r}^{(0)}\| \leq 3\sqrt{d} \cdot 2\sigma_w\sqrt{d} + 2\sigma_b\sqrt{2\ln(2m)} \leq \frac{\eta\sqrt{d}}{nm}$$

$\qquad\square$

Finally, we present the following two lemmas about the range of the loss derivative.

Denote $\ell_i'^{(t)} := \nabla_{f_{\boldsymbol{\theta}^{(t)}}(x_i)}\ell(y_i f_{\boldsymbol{\theta}^{(t)}}(x)) = -\dfrac{y_i\exp\left(-y_i f_{\boldsymbol{\theta}^{(t)}}(\boldsymbol{x}_i)\right)}{1 + \exp\left(-y_i f_{\boldsymbol{\theta}^{(t)}}(\boldsymbol{x}_i)\right)} = -\dfrac{y_i}{1 + \exp\left(y_i f_{\boldsymbol{\theta}^{(t)}}(\boldsymbol{x}_i)\right)}$.

**Lemma D.14.** *For each $i \in I$, we have $-\dfrac{2}{3} \leq y_i\ell_i'^{(0)} \leq -\dfrac{1}{3}$.*

*Proof of Lemma D.14.* By applying Lemma D.13, we know

$$|f_{\boldsymbol{\theta}^{(0)}}(\boldsymbol{x}_i)| \leq \frac{1}{m}\left(\sum_{s \in \{-1,+1\}}\sum_{r \in [m]} |\langle \boldsymbol{w}_{s,r}^{(0)}, \boldsymbol{x}_i \rangle + b_{s,r}^{(0)}|\right) \leq \frac{2\eta\sqrt{d}}{nm} \leq \ln 2$$

Then, we have $1/2 \leq \exp\left(y_i f_{\boldsymbol{\theta}^{(0)}}(\boldsymbol{x}_i)\right) \leq 2$, and we can derive that

$$-\frac{2}{3} \leq y_i\ell_i'^{(0)} = -\frac{1}{1 + \exp\left(y_i f_{\boldsymbol{\theta}^{(0)}}(\boldsymbol{x}_i)\right)} \leq -\frac{1}{3}.$$

$\qquad\square$

**Lemma D.15.** *For each $i \in I$ and any time step $t$, we have $-1 \leq y_i\ell_i'^{(t)} \leq 0$.*

*Proof of Lemma D.15.* It can be easily checked as follows.

$$-1 \leq y_i \ell_i'^{(t)} = -\frac{1}{1 + \exp\left(y_i f_{\boldsymbol{\theta}^{(t)}}(\boldsymbol{x}_i)\right)} \leq 0.$$

$\square$

# E  PROOF FOR SECTION 4: FEATURE-AVERAGING REGIME

In this section, we provide the proof of Theorem 4.5. We analyze the training dynamics of gradient descent, which constitutes the main part of our proof.

## E.1  ANALYSIS OF TRAINING DYNAMICS

We first assume that all the properties and lemmas mentioned in Appendix D hold with high probability over the sampled training dataset and the network initialization.

Now, we introduce some useful notations. Denote $S_{s,i}^{(t)}$ as the set of indices of neurons in positive or negative class (determined by $s$) which has been activated by training data point $x_i$ at time step $t$. Formally, we define it as $S_{s,i}^{(t)} = \{r \in [m] : \langle w_{s,r}^{(t)}, x_i \rangle + b_{s,r}^{(t)} \geq 0\}$ for $s \in \{-1, +1\}$ and $i \in I$. The following lemma describes the set of activated neurons after the first gradient descent update.

**Lemma E.1.** $S_{1,i}^{(1)} = [m]$ for all $i \in I_+$ and $S_{-1,i}^{(1)} = [m]$ for all $i \in I_-$.

*Proof of Lemma E.1.* Without loss of generality, we consider the case when $x_i$ belongs to the positive class. We show that $\langle w_{1,r}^{(1)}, x_i \rangle + b_{1,r}^{(1)} \geq 0$ for all $r \in [m]$. By applying the gradient descent update and Lemma D.13, we have

$$
\begin{aligned}
\langle w_{1,r}^{(1)}, x_i \rangle + b_{1,r}^{(1)} =& \langle w_{1,r}^{(0)}, x_i \rangle + b_{1,r}^{(0)} - \eta \left( \langle \nabla_{w_{1,r}} \mathcal{L}(\theta^{(0)}), x_i \rangle + \nabla_{b_{1,r}} \mathcal{L}(\theta^{(0)}) \right) \\
\geq& \frac{\eta}{nm} \left( -d^{1/2} - mn \left( \langle \nabla_{w_{1,r}} \mathcal{L}(\theta^{(0)}), x_i \rangle + \nabla_{b_{1,r}} \mathcal{L}(\theta^{(0)}) \right) \right)
\end{aligned}
$$

First, we examine the update of linear terms as follows:

$$
\begin{aligned}
&- mn \langle \nabla_{w_{1,r}} \mathcal{L}(\theta^{(0)}), x_i \rangle \\
=& - \langle \sum_{p \in I_+} \ell_p'^{(0)} \mathbb{1} \left( \langle w_{1,r}^{(0)}, x_p \rangle + b_{1,r}^{(0)} \geq 0 \right) x_p - \sum_{p \in I_-} \ell_p'^{(0)} \mathbb{1} \left( \langle w_{1,r}^{(0)}, x_i \rangle + b_{1,r}^{(0)} \geq 0 \right) x_p, x_i \rangle \\
\geq& - \sum_{p \in I_{c(i)}} \ell_p'^{(0)} \mathbb{1} \left( \langle w_{1,r}^{(0)}, x_p \rangle + b_{1,r}^{(0)} \geq 0 \right) \langle x_p, x_i \rangle + \sum_{p \notin I_{c(i)}} \ell_p'^{(0)} \mathbb{1} \left( \langle w_{1,r}^{(0)}, x_p \rangle + b_{1,r}^{(0)} \geq 0 \right) |\langle x_p, x_i \rangle| \\
\geq& \frac{1}{3} \sum_{p \in I_{c(i)}} \mathbb{1} \left( \langle w_{1,r}^{(0)}, x_p \rangle + b_{1,r}^{(0)} \geq 0 \right) \langle x_p, x_i \rangle - \frac{2}{3} \sum_{p \notin I_{c(i)}} \mathbb{1} \left( \langle w_{1,r}^{(0)}, x_i \rangle + b_{1,r}^{(0)} \geq 0 \right) |\langle x_p, x_i \rangle| \\
\geq& \frac{d}{6} \sum_{p \in I_{c(i)}} \mathbb{1} \left( \langle w_{1,r}^{(0)}, x_p \rangle + b_{1,r}^{(0)} \geq 0 \right) - \frac{2\Delta}{3} \sum_{p \notin I_{c(i)}} \mathbb{1} \left( \langle w_{1,r}^{(0)}, x_p \rangle + b_{1,r}^{(0)} \geq 0 \right) \quad \left( Recall \ \Delta = 4\sqrt{d} \ln(d) \right) \\
=& \frac{1}{6} d |T_{1,r,c(i)}| - \frac{2\Delta}{3} \sum_{l \in r/\{c(i)\}} |T_{1,r,l}| \\
\geq& \frac{nd}{18km^2} - \frac{2n\Delta}{3} \quad \text{(By Proposition D.12)} \\
\geq& n\Delta
\end{aligned}
\tag{14}
$$

Next, we examine the update of the bias term as follows:

$$
\begin{aligned}
-mn \nabla_{b_{1,r}} \mathcal{L}(\theta^{(0)}) =& - \sum_{p \in I_+} \ell_p'^{(0)} \mathbb{1} \left( \langle w_{1,r}^{(0)}, x_p \rangle + b_{1,r}^{(0)} \geq 0 \right) + \sum_{p \in I_-} \ell_p'^{(0)} \mathbb{1} \left( \langle w_{1,r}^{(0)}, x_p \rangle + b_{1,r}^{(0)} \geq 0 \right) \\
\geq& - \frac{2}{3} \sum_{p \in I} \mathbb{1} \left( \langle w_{1,r}^{(0)}, x_p \rangle + b_{1,r}^{(0)} \geq 0 \right) \\
\geq& - \frac{2n}{3}
\end{aligned}
\tag{15}
$$

Combining (14) and (15) together, we know

$$\langle \boldsymbol{w}_{1,r}^{(1)}, \boldsymbol{x}_i \rangle + b_{1,r}^{(1)} \geq \frac{\eta}{nm} \left( -d^{1/2} - mn \left( \left\langle \nabla_{\boldsymbol{w}_{1,r}} \mathcal{L}(\boldsymbol{\theta}^{(0)}), \boldsymbol{x}_i \right\rangle + \nabla_{b_{1,r}} \mathcal{L}(\boldsymbol{\theta}^{(0)}) \right) \right)$$
$$\geq \frac{\eta}{nm} (-d^{1/2} + n\Delta - \frac{2n}{3})$$
$$\geq 0$$

Since this inequality holds for all $r \in [m]$, we have that $S_{1,i}^{(1)} = [m]$. For the case when $\boldsymbol{x}_i$ belongs to the negative class, we have $S_{-1,i}^{(1)} = [m]$ using the same argument. $\qquad \square$

Now, we analyze the dynamics of the coefficients in the training process, where we first give the following weight-decomposition lemma.

**Lemma E.2** (Weight Decomposition). *During the training dynamics, there exists the following coefficient sequences $\lambda_{s,r,j}^{(t)}$ and $\sigma_{s,r,i}^{(t)}$ for each $s \in \{-1, +1\}, r \in [m], j \in J, i \in I$ such that*

$$\boldsymbol{w}_{s,r}^{(t)} = \boldsymbol{w}_{s,r}^{(0)} + \sum_{j \in J} \lambda_{s,r,j}^{(t)} \boldsymbol{\mu}_j \|\boldsymbol{\mu}_j\|^{-2} + \sum_{i \in I} \sigma_{s,r,i}^{(t)} \boldsymbol{\xi}_i \|\boldsymbol{\xi}_i\|^{-2}$$

*Proof of Lemma E.2.* First, we construct a set of $\{\hat{\lambda}_{s,r,j}^{(t)}\}$ and $\{\hat{\sigma}_{s,r,i}^{(t)}\}$ according to the following recursive formulas:

$$\hat{\lambda}_{s,r,j}^{(t+1)} = \hat{\lambda}_{s,r,j}^{(t)} - \frac{s\eta}{nm} \cdot \sum_{i \in I_j} \ell_i'^{(t)} \|\boldsymbol{\mu}_j\|^2 \mathbb{1} \left( \langle \boldsymbol{w}_{s,r}^{(t)}, \boldsymbol{x}_i \rangle + b_{s,r}^{(t)} \geq 0 \right).$$

$$\hat{\sigma}_{s,r,i}^{(t+1)} = \hat{\sigma}_{s,r,i}^{(t)} - \frac{s\eta}{nm} \cdot \ell_i'^{(t)} \|\boldsymbol{\xi}_i\|^2 \mathbb{1} \left( \langle \boldsymbol{w}_{s,r}^{(t)}, \boldsymbol{x}_i \rangle + b_{s,r}^{(t)} \geq 0 \right).$$

$$\hat{\lambda}_{s,r,j}^{(0)} = 0, \hat{\sigma}_{s,r,i}^{(0)} = 0.$$

Now, we prove by induction on $t$ that $\{\hat{\lambda}_{s,r,j}^{(t)}\}$ and $\{\hat{\sigma}_{s,r,i}^{(t)}\}$ constructed as above satisfy that

$$\boldsymbol{w}_{s,r}^{(t)} = \boldsymbol{w}_{s,r}^{(0)} + \sum_{j \in J} \hat{\lambda}_{s,r,j}^{(t)} \boldsymbol{\mu}_j \|\boldsymbol{\mu}_j\|^{-2} + \sum_{i \in I} \hat{\sigma}_{s,r,i}^{(t)} \boldsymbol{\xi}_i \|\boldsymbol{\xi}_i\|^{-2}$$

The base case when $t = 0$ the conclusion holds trivially. Assuming the inductive hypothesis holds at time step $t$, we now consider the case at time step $t + 1$. By the update equation in gradient descent, we know that

$$\boldsymbol{w}_{s,r}^{(t+1)} = \boldsymbol{w}_{s,r}^{(t)} - \frac{s\eta}{nm} \sum_{i \in I} \ell_i'^{(t)} \boldsymbol{x}_i \mathbb{1} \left( \langle \boldsymbol{w}_{s,r}^{(t)}, \boldsymbol{x}_i \rangle + b_{s,r}^{(t)} \geq 0 \right)$$

$$= \boldsymbol{w}_{s,r}^{(t)} - \frac{s\eta}{nm} \sum_{i \in I} \ell_i'^{(t)} \left( \boldsymbol{\mu}_{c(i)} + \boldsymbol{\xi}_i \right) \mathbb{1} \left( \langle \boldsymbol{w}_{s,r}^{(t)}, \boldsymbol{x}_i \rangle + b_{s,r}^{(t)} \geq 0 \right)$$

$$= \boldsymbol{w}_{s,r}^{(t)} - \frac{s\eta}{nm} \left( \sum_{j \in J} \boldsymbol{\mu}_j \sum_{i \in I_j} \ell_i'^{(t)} \mathbb{1} \left( \langle \boldsymbol{w}_{s,r}^{(t)}, \boldsymbol{x}_i \rangle + b_{s,r}^{(t)} \geq 0 \right) + \sum_{i \in I} \boldsymbol{\xi}_i \ell_i'^{(t)} \mathbb{1} \left( \langle \boldsymbol{w}_{s,r}^{(t)}, \boldsymbol{x}_i \rangle + b_{s,r}^{(t)} \geq 0 \right) \right)$$

$$= \boldsymbol{w}_{s,r}^{(0)} + \sum_{j \in J} \boldsymbol{\mu}_j \|\boldsymbol{\mu}_j\|^{-2} \left( \hat{\lambda}_{s,r,j}^{(t)} - \frac{s\eta}{nm} \cdot \sum_{i \in I_j} \ell_i'^{(t)} \|\boldsymbol{\mu}_j\|^2 \mathbb{1} \left( \langle \boldsymbol{w}_{s,r}^{(t)}, \boldsymbol{x}_i \rangle + b_{s,r}^{(t)} \geq 0 \right) \right)$$

$$+ \sum_{i \in I} \boldsymbol{\xi}_i \|\boldsymbol{\xi}_i\|^{-2} \left( \hat{\sigma}_{s,r,i}^{(t)} - \frac{s\eta}{nm} \cdot \ell_i'^{(t)} \|\boldsymbol{\xi}_i\|^2 \mathbb{1} \left( \langle \boldsymbol{w}_{s,r}^{(t)}, \boldsymbol{x}_i \rangle + b_{s,r}^{(t)} \geq 0 \right) \right)$$

$$= \boldsymbol{w}_{s,r}^{(0)} + \sum_{j \in J} \hat{\lambda}_{s,r,j}^{(t+1)} \boldsymbol{\mu}_j \|\boldsymbol{\mu}_j\|^{-2} + \sum_{i \in I} \hat{\sigma}_{s,r,i}^{(t+1)} \boldsymbol{\xi}_i \|\boldsymbol{\xi}_i\|^{-2}$$

This concludes the inductive step and the proof of the lemma. $\qquad \square$

Naturally, we have the following corollaries.

**Corollary E.3.** *The coefficients $\lambda_{s,r,j}^{(t)}, \sigma_{s,r,i}^{(t)}$ for $s \in \{-1, +1\}, r \in [m], j \in J, i \in I$ defined in Corollary (E.2) satisfy the following update equations:*

$$\lambda_{s,r,j}^{(t+1)} = \lambda_{s,r,j}^{(t)} - \frac{s\eta}{nm} \cdot \sum_{i \in I_j} \ell_i'^{(t)} \|\boldsymbol{\mu_j}\|^2 \mathbb{1}\left(\langle \boldsymbol{w}_{s,r}^{(t)}, \boldsymbol{x}_i \rangle + b_{s,r}^{(t)} \geq 0\right),$$

$$\sigma_{s,r,i}^{(t+1)} = \sigma_{s,r,i}^{(t)} - \frac{s\eta}{nm} \cdot \ell_i'^{(t)} \|\boldsymbol{\xi_i}\|^2 \mathbb{1}\left(\langle \boldsymbol{w}_{s,r}^{(t)}, \boldsymbol{x}_i \rangle + b_{s,r}^{(t)} \geq 0\right),$$

$$\lambda_{s,r,j}^{(0)} = 0, \sigma_{s,r,i}^{(0)} = 0.$$

Indeed, we can only focus on the dynamics of noise coefficients due to the following lemma.

**Corollary E.4.** *The coefficient sequences $\lambda_{s,r,j}^{(t)}$ and $\sigma_{s,r,i}^{(t)}$ for each pair $s \in \{-1, +1\}, r \in [m], j \in J, i \in I$ defined in Lemma F.6 satisfy:*

$$\lambda_{s,r,j}^{(t)} \|\boldsymbol{\mu}_j\|^{-2} = \sum_{i \in I_j} \sigma_{s,r,i}^{(t)} \|\boldsymbol{\xi}_i\|^{-2}.$$

*Proof of Corollary E.4.* Using the result in Corollary E.3, we know that

$$\left(\lambda_{s,r,j}^{(t'+1)} - \lambda_{s,r,j}^{(t')}\right) \|\boldsymbol{\mu}_j\|^{-2} = -\frac{s\eta}{nm} \cdot \sum_{i \in I_j} \ell_i'^{(t')} \mathbb{1}\left(\langle \boldsymbol{w}_{s,r}^{(t')}, \boldsymbol{x}_i \rangle + b_{s,r}^{(t')} \geq 0\right)$$

$$= \sum_{i \in I_j} \left(\sigma_{s,r,i}^{(t'+1)} - \sigma_{s,r,i}^{(t')}\right) \|\boldsymbol{\xi}_i\|^{-2}.$$

Then summing up the above equations from $t' = 0$ to $t' = t - 1$, we have

$$\lambda_{s,r,j}^{(t)} \|\boldsymbol{\mu}_j\|^{-2} = \sum_{i \in I_j} \sigma_{s,r,i}^{(t)} \|\boldsymbol{\xi}_i\|^{-2}.$$

$\square$

We show that the sign of each feature/noise coefficient remains unchanged during the full training process as in the following lemma.

**Corollary E.5.** *The coefficient sequences $\lambda_{s,r,j}^{(t)}$ and $\sigma_{s,r,i}^{(t)}$ for each pair $s \in \{-1, +1\}, r \in [m], j \in J, i \in I, t \geq 0$ defined in Lemma E.2 satisfy:*

$$\lambda_{s,r,j}^{(t)} \begin{cases} \geq 0 \text{ if } i \in J_s, \\ < 0 \text{ if } i \notin J_s. \end{cases}$$

$$\sigma_{s,r,i}^{(t)} \begin{cases} \geq 0 \text{ if } i \in I_s, \\ < 0 \text{ if } i \notin I_s. \end{cases}$$

*Proof of Corollary E.5.* Using the results in Corollary E.3 and Lemma D.15, we know that

$$\text{sgn}\left(\sigma_{s,r,i}^{(t+1)} - \sigma_{s,r,i}^{(t)}\right) = \text{sgn}\left(-\frac{s\eta}{nm} \cdot \ell_i'^{(t)} \|\boldsymbol{\xi_i}\|^2 \mathbb{1}\left(\langle \boldsymbol{w}_{s,r}^{(t)}, \boldsymbol{x}_i \rangle + b_{s,r}^{(t)} \geq 0\right)\right)$$

$$= \text{sgn}\left(-s\ell_i'^{(t)}\right)$$

$$= \text{sgn}\left(sy_i\right)$$

Noting that $\sigma_{s,r,i}^{(0)} = 0$, we know that

$$\text{sgn}\left(\sigma_{s,r,i}^{(t)}\right) = \text{sgn}\left(sy_i\right)$$

That is to say

$$\sigma_{s,r,i}^{(t)} \begin{cases} \geq 0 \text{ if } i \in I_s, \\ < 0 \text{ if } i \notin I_s. \end{cases}$$

Then using the result in Corollary E.4, we know

$$\lambda_{s,r,j}^{(t)} \begin{cases} \geq 0 \text{ if } j \in J_s, \\ < 0 \text{ if } j \notin J_s. \end{cases}$$

$\square$

In the following proof, we need the concept of *margin*. We denote the margin of training data point $\boldsymbol{x}_i$ at time step $t$ as $q_i^{(t)} = y_i f_{\theta^{(t)}}(\boldsymbol{x}_i)$ and the margin gap between training data points $\boldsymbol{x}_i$ and $\boldsymbol{x}_j$ at time step $t$ as $\Delta_q^{(t)}(i,j) = y_i f_{\theta^{(t)}}(\boldsymbol{x}_i) - y_j f_{\theta^{(t)}}(\boldsymbol{x}_j)$.

First, we analyze the relationship between the margin gap and the loss derivatives' ratio for the two training data points in the following lemma.

**Lemma E.6.** *For any time step $t$ and two training data points $\boldsymbol{x}_i, \boldsymbol{x}_j$, if $q_i \geq q_j$, we have*

$$e^{\Delta_q^{(t)}(i,j)/2} \leq \frac{y_j \ell_j'^{(t)}}{y_i \ell_i'^{(t)}} \leq e^{\Delta_q^{(t)}(i,j)}.$$

*Proof of Lemma E.6.* Recall $\ell'(x) = -\dfrac{e^{-x}}{1+e^{-x}} = \dfrac{1}{1+e^x}$. Then

$$\frac{\ell_j'^{(t)}}{\ell_i'^{(t)}} \cdot e^{-\Delta_q^{(t)}(i,j)} = \frac{1+e^{q_i}}{e^{\Delta_q^{(t)}(i,j)} + e^{q_i}} \leq 1.$$

Since the exponential function is convex, we know that

$$\frac{\ell_j'^{(t)}}{\ell_i'^{(t)}} \cdot e^{-\Delta_q^{(t)/2}(i,j)} = \frac{1+e^{q_i}}{e^{(q_i-q_j)/2} + e^{(q_i+q_j)/2}} \geq 1.$$

$\square$

Next, we establish the relationship between the coefficient $\lambda_{s,r,j}^{(t)}$'s and $\sigma_{s,r,i}^{(t)}$'s in Corollary E.2 and margin $q$ we defined before. We denote

$$\hat{q}_i^{(t)} = \frac{1}{m} \left( \sum_{r \in [m]} \lambda_{y_i,r,c(i)}^{(t)} + \sum_{r \in [m]} \sigma_{y_i,r,i}^{(t)} \right),$$

$$\hat{\Delta}_q^{(t)}(i,j) = \hat{q}_i - \hat{q}_j = \frac{1}{m} \sum_{r \in [m]} \left( \lambda_{y_i,r,c(i)}^{(t)} - \lambda_{y_j,r,c(j)}^{(t)} \right) + \frac{1}{m} \sum_{r \in [m]} \left( \sigma_{y_i,r,i}^{(t)} - \sigma_{y_j,r,j}^{(t)} \right).$$

Later, we will show that $\hat{q}_i^{(t)}$ is a good approximation for margin $q_i$ and $\hat{\Delta}_q^{(t)}(i,j)$ is a good approximation for margin gap $\Delta_q^{(t)}(i,j)$ in Lemma E.14 and Corollary E.15.

Next, we arrive at the main part of the proof. Inspired by Kou et al. (2023a), we can prove that the training data's margin tends to balance automatically.

Denote $\epsilon = \max \left\{ \dfrac{2\ln(\frac{n}{k})}{\sqrt{\frac{n}{k}} - \ln(\frac{n}{k})}, \dfrac{k^2\Delta}{d}, \dfrac{k^2}{n} \right\}$. We know that $\epsilon = o(k^{-2.5})$ according to our hyper-parameter Assumption D.4.

**Lemma E.7.** *For $t \leq T_0 = \exp(\tilde{O}(k^{0.5})), i,j \in I, s \in \{-1,1\}$, the following statements hold:*

1. $\dfrac{k}{2n}\ln(t\eta) \leq \sigma^{(t)}_{y_i,r,i} \leq \dfrac{2k}{n}\ln(t+1)$,

2. $\hat{\Delta}^{(t)}_q(i,j) \leq 5\epsilon$ *when* $c(i) = c(j)$,

3. $\hat{\Delta}^{(t)}_q(i,j) \leq 63\epsilon$,

4. $\Delta^{(t)}_q(i,j) \leq 65\epsilon$,

5. $y_i\ell'^{(t)}_i / y_j\ell'^{(t)}_j \leq 1 + 14\epsilon$ *when* $c(i) = c(j)$,

6. $y_i\ell'^{(t)}_i / y_j\ell'^{(t)}_j \leq 1 + 130\epsilon$,

7. $S^{(t)}_{s,i} = [m]$ *for* $i \in I_s$,

8. $|\lambda^{(t)}_{-s,r,c(i)}| \leq \epsilon, |\sigma^{(t)}_{-s,r,i}| \leq 2\epsilon$ *for* $i \in I_s$.

**Remark E.8.** *Property 1 of Lemma E.7 shows that the growth rate of noise coefficient is the logarithm of time, i.e., $\sigma^{(t)}_{s,r,i} \approx \log t$. Property 2 and Property 3 show that the approximate margin gap is small, and the gap between two training data points with the same cluster index is smaller. Property 4 suggests that the exact margin gap is also small. Property 5 and Property 6 provide upper bounds for the loss derivative ratio. Property 7 manifests that the positive training data points can activate all positive neurons and the negative training data points can activate all negative neurons, respectively.*

*Proof of Lemma E.7.* Without loss of generality, we assume that $i \in I_+$.

We use induction to prove this lemma.

**Step 1:** We first consider the base case when $t = 1$ for the induction.

**Property 7:** Property 7 is exactly the conclusion of Lemma E.1.

Indeed, other properties can be easily verified because we adopted a small initialization.

**Property 1 and 8:** By Lemma E.3, $\eta \leq d^{-2}$ and noting that $\sigma^{(0)}_{s,r,i} = 0$, we have

$$|\sigma^{(1)}_{s,r,i}| = |\sigma^{(0)}_{s,r,i} - \dfrac{\eta}{nm} \cdot \ell'^{(0)}_i\|\boldsymbol{\xi_i}\|^2 \mathbb{1}\left(\langle \boldsymbol{w}^{(0)}_{s,r}, \boldsymbol{x}_i\rangle + b^{(0)}_{s,r} \geq 0\right)| \leq \dfrac{2\eta d}{nm} \leq \dfrac{2k}{n}\ln 2 \leq 2\epsilon.$$

and by Corollary E.5, for $i \in I_s$ we have

$$\sigma^{(1)}_{s,r,i} \geq 0 \geq \dfrac{k}{2n}\ln(\eta).$$

**Property 2, 3 and 8:** By applying Lemma E.3, $\eta \leq d^{-2}$ and noting that $\lambda^{(0)}_{s,r,j} = 0$, we have

$$|\lambda^{(1)}_{s,r,j}| = |\lambda^{(0)}_{s,r,j} - \dfrac{s\eta}{nm} \cdot \sum_{i \in I_j}\ell'^{(0)}_i\|\boldsymbol{\mu_j}\|^2 \mathbb{1}\left(\langle \boldsymbol{w}^{(0)}_{s,r}, \boldsymbol{x}_i\rangle + b^{(0)}_{s,r} \geq 0\right)| \leq \dfrac{2\eta d}{m} \leq \dfrac{\epsilon}{2m} \text{(Property 8)}$$

Then, by the definition of $\hat{\Delta}^{(t)}_q(i,j)$, we know that

$$\begin{aligned}
\hat{\Delta}^{(1)}_q(i,j) &= \sum_{r \in [m]}\left(\lambda^{(1)}_{1,r,c(i)} - \lambda^{(1)}_{1,r,c(j)}\right) + \sum_{r \in [m]}\left(\sigma^{(1)}_{1,r,i} - \sigma^{(1)}_{1,r,j}\right)\\
&\leq 2m\dfrac{\epsilon}{2m} + 2m\dfrac{2\eta d}{nm}\\
&\leq 5\epsilon \qquad \text{(Property 2)}\\
&\leq 63\epsilon \qquad \text{(Property 3)}
\end{aligned}$$

**Property 4:** By applying Lemma D.15, we know that

$$
\begin{aligned}
\left\|\boldsymbol{w}_{s,j}^{(1)}\right\| &= \left\|\boldsymbol{w}_{s,j}^{(0)} - \frac{\eta}{nm}\sum_{i\in I}\ell_i'^{(0)}\mathbb{1}\left(\langle\boldsymbol{w}_{s,r}^{(t)},\boldsymbol{x}_i\rangle + b_{s,r}^{(t)} \geq 0\right)\boldsymbol{x}_i\right\| \\
&\leq \left\|\boldsymbol{w}_{s,j}^{(0)}\right\| + \frac{\eta}{nm}\sum_{i\in I}\|\boldsymbol{x}_i\| \\
&\leq \left\|\boldsymbol{w}_{s,j}^{(0)}\right\| + \frac{3\sqrt{d}\eta}{m} \\
&\leq \frac{\epsilon}{4\sqrt{d}} \\
\left\|b_{s,j}^{(1)}\right\| &= \left\|b_{s,j}^{(0)} - \frac{\eta}{nm}\sum_{i\in I}\ell_i'^{(0)}\mathbb{1}\left(\langle\boldsymbol{w}_{s,r}^{(t)},\boldsymbol{x}_i\rangle + b_{s,r}^{(t)} \geq 0\right)\right\| \\
&\leq \left\|b_{s,j}^{(0)}\right\| + \frac{\eta}{m} \\
&\leq \frac{\epsilon}{10}
\end{aligned}
\tag{16}
$$

Then, we have

$$
\begin{aligned}
|q_i^{(1)}| &= |f_{\theta^{(1)}}(\boldsymbol{x}_i)| \\
&\leq \frac{1}{m}\left(\sum_{s\in\{-1,+1\}}\sum_{r\in[m]}|\langle\boldsymbol{w}_{s,j}^{(0)},\boldsymbol{x}_i\rangle + b_{s,j}^{(0)}|\right) \\
&\leq \frac{1}{m}\left(2m\left(\frac{\epsilon}{4} + \frac{\epsilon}{10}\right)\right) \\
&\leq \epsilon
\end{aligned}
$$

By triangle inequality, we have

$$
\Delta_q^{(1)}(i,j) \leq |q_i^{(1)}| + |q_j^{(1)}| \leq 2\epsilon
$$

**Property 5 and 6:** By using the inequality above, Lemma E.6 and noting that $e^x \leq 1 + 2x$ for small $x$, we have

$$
\frac{\ell_j'^{(1)}}{\ell_i'^{(1)}} \leq e^{\Delta_q^{(1)}(i,j)} \leq 1 + 2\Delta_q^{(1)}(i,j) \leq \underbrace{1 + 14\epsilon}_{\text{Property 5}} \leq \underbrace{1 + 130\epsilon}_{\text{Property 6}}.
$$

Now we complete the proof of the base case when $t = 1$ for induction.

**Step 2:** Assuming that the inductive hypothesis at time step $t$ holds, we consider time step $t + 1$. We first give some useful lemmas based on the inductive hypotheses, and then go on to inductive proofs based on these lemmas.

**Lemma E.9.** *Assuming the inductive hypotheses hold before time step $t$ and $i \in I_s$, the update equations in Corollary E.3 can be simplified as follows:*

$$
\lambda_{s,r,c(i)}^{(t+1)} = \lambda_{s,r,c(i)}^{(t)} - \frac{s\eta}{nm}\cdot\sum_{p\in I_{c(i)}}\ell_p'^{(t)}\|\boldsymbol{\mu}_{c(i)}\|^2.
$$

$$
\sigma_{s,r,i}^{(t+1)} = \sigma_{s,r,i}^{(t)} - \frac{s\eta}{nm}\cdot\ell_i'^{(t)}\|\boldsymbol{\xi_i}\|^2.
$$

*Proof of Lemma E.9.* By Property 7 in Lemma E.7 (inductive hypothesis) and Corollary E.3, the conclusion is straightforward. ∎

Then, we demonstrate that the bias term remains consistently small as the following lemma.

**Lemma E.10.** *For every* $s \in \{-1, +1\}, r \in [m], i \in I$, *we have*

$$|b_{s,r}^{(t)}| \leq \frac{\epsilon}{6}.$$

*Proof of Lemma E.10.* Without loss of generality, we assume that $s = 1$. Then, for any $t' \leq t$, by using Lemma E.9, we know that

$$\sigma_{1,r,i}^{(t'+1)} = \sigma_{1,r,i}^{(t')} - \frac{\eta}{nm} \cdot \ell_i^{'(t)} \|\boldsymbol{\xi_i}\|^2.$$

$$b_{1,r}^{(t'+1)} = b_{1,r}^{(t')} - \frac{\eta}{nm} \sum_{p \in I} \ell_p^{'(t)} \mathbb{1} \left( \langle \boldsymbol{w}_{1,r}^{(t)}, \boldsymbol{x}_p \rangle + b_{1,r}^{(t)} \geq 0 \right)$$

$$\leq b_{1,r}^{(t')} - \frac{\eta}{nm} \sum_{p \in I^+} \ell_p^{'(t)}$$

Thus, we derive

$$\sigma_{1,r,i}^{(t'+1)} - \sigma_{1,r,i}^{(t')} \geq \frac{\ell_i^{'(t)} \|\boldsymbol{\xi_i}\|^2}{\sum\limits_{p \in I^+} \ell_p^{'(t)}} \left( b_{1,r}^{(t'+1)} - b_{1,r}^{(t')} \right) \geq \frac{d}{2n} \left( b_{1,r}^{(t'+1)} - b_{1,r}^{(t')} \right).$$

Summing up the above inequality from $t' = 1$ to $t' = t - 1$, we have

$$b_{1,r}^{(t)} - b_{1,r}^{(1)} \leq \frac{2n}{d} \left( \sigma_{1,r,i}^{(t)} - \sigma_{1,r,i}^{(1)} \right).$$

Then by inequality (16) and Property (1) in the inductive hypotheses, we know that

$$b_{1,r}^{(t)} \leq \frac{2n}{d} \sigma_{1,r,i}^{(t)} - \frac{2n}{d} \sigma_{1,r,i}^{(1)} + b_{1,r}^{(1)} \leq \frac{4k \ln(t+1)}{d} + \frac{\epsilon}{10} \leq \frac{\epsilon}{6}.$$

Reusing the same argument as in the previous proof, we know that $b_{1,r}^{(t)} \geq -\frac{\epsilon}{6}$. ∎

Next, we prove that the true value of the margin $q_i^{(t)}$ is close to its estimated value $\hat{q}_i^{(t)}$. To estimate the margin, we first estimate the value of each neuron in the following Lemma E.11, Lemma E.12 and Lemma E.13.

**Lemma E.11.** *Assuming the inductive hypotheses hold before time step* $t$, *for all* $s \in \{-1, +1\}, r \in [m], j \in J$, *we have*

$$\left| \langle \boldsymbol{w}_{s,r}^{(t)}, \boldsymbol{\mu}_j \rangle - \lambda_{s,r,j}^{(t)} \right| \leq \frac{\epsilon}{6}.$$

*Proof of Lemma E.11.* We bound the gap between the inner product $\langle \boldsymbol{w}_{s,r}^{(t)}, \boldsymbol{\mu}_j \rangle$ and feature coefficient $\lambda_{s,r,j}^{(t)}$ as follows:

$$\left| \langle \boldsymbol{w}_{s,r}^{(t)}, \boldsymbol{\mu}_j \rangle - \lambda_{s,r,j}^{(t)} \right|$$

$$= \left| \left\langle \boldsymbol{w}_{s,r}^{(0)} + \sum_{p \in J} \lambda_{s,r,p}^{(t)} \boldsymbol{\mu}_p \|\boldsymbol{\mu}_p\|^{-2} + \sum_{q \in I} \sigma_{s,r,q}^{(t)} \boldsymbol{\xi}_q \|\boldsymbol{\xi}_q\|^{-2}, \boldsymbol{\mu}_j \right\rangle - \lambda_{s,r,j}^{(t)} \right|$$

$$\leq \left| \langle \boldsymbol{w}_{s,r}^{(0)}, \boldsymbol{\mu}_j \rangle \right| + \left| \langle \lambda_{s,r,j}^{(t)} \boldsymbol{\mu}_j \|\boldsymbol{\mu}_j\|^{-2}, \boldsymbol{\mu}_j \rangle - \lambda_{s,r,j}^{(t)} \right| + \sum_{p \neq j} \lambda_{s,r,p}^{(t)} \|\boldsymbol{\mu}_p\|^{-2} |\langle \boldsymbol{\mu_p}, \boldsymbol{\mu}_j \rangle| + \sum_{q \in I} \sigma_{s,r,q}^{(t)} \|\boldsymbol{\xi}_q\|^{-2} |\langle \boldsymbol{\xi}_q, \boldsymbol{\mu}_j \rangle|$$

$$\leq \left| \langle \boldsymbol{w}_{s,r}^{(0)}, \boldsymbol{\mu}_j \rangle \right| + \sum_{p \neq j} \lambda_{s,r,p}^{(t)} \frac{2\Delta}{d} + \sum_{q \in I} \sigma_{s,r,q}^{(t)} \frac{2\Delta}{d}$$

$$= \left| \langle \boldsymbol{w}_{s,r}^{(0)}, \boldsymbol{\mu}_j \rangle \right| + \frac{2\Delta}{d} \left( \sum_{p \neq j} \sum_{q \in I_p} \sigma_{s,r,q}^{(t)} \frac{\|\boldsymbol{\xi}_q\|^2}{\|\boldsymbol{\mu}_p\|^2} + \sum_{q \in I} \sigma_{s,r,q}^{(t)} \right)$$

$$\leq \sqrt{d} \|\boldsymbol{w}_{s,r}^{(0)}\| + \frac{12\Delta \ln(t+1)k}{d} \leq \frac{\epsilon}{6}.$$

The first equation employs the weight decomposition in Lemma E.2; the second inequality expands the inner product and applies the triangle inequality; the third inequality utilizes the properties from Proposition D.6; the fourth equation utilizes Corollary E.4; the fifth inequality utilizes Property 1 and Property 8 in the inductive hypotheses. ∎

**Lemma E.12.** *Assuming the inductive hypotheses hold before time step $t$, for all $s \in \{-1, +1\}, r \in [m], i \in I$, we have*

$$\left| \langle \boldsymbol{w}_{s,r}^{(t)}, \boldsymbol{\xi}_i \rangle - \sigma_{s,r,i}^{(t)} \right| \leq \frac{\epsilon}{6}.$$

*Proof of Lemma E.12.* We bound the gap between the inner product $\langle \boldsymbol{w}_{s,r}^{(t)}, \boldsymbol{\xi}_i \rangle$ and noise coefficient $\sigma_{s,r,i}^{(t)}$ as follows:

$$\left| \langle \boldsymbol{w}_{s,r}^{(t)}, \boldsymbol{\xi}_i \rangle - \sigma_{s,r,i}^{(t)} \right|$$

$$= \left| \left\langle \boldsymbol{w}_{s,r}^{(0)} + \sum_{p \in J} \lambda_{s,r,p}^{(t)} \boldsymbol{\mu}_p \|\boldsymbol{\mu}_p\|^{-2} + \sum_{q \in I} \sigma_{s,r,q}^{(t)} \boldsymbol{\xi}_q \|\boldsymbol{\xi}_q\|^{-2}, \boldsymbol{\xi}_i \right\rangle - \sigma_{s,r,i}^{(t)} \right|$$

$$\leq \left| \langle \boldsymbol{w}_{s,r}^{(0)}, \boldsymbol{\xi}_i \rangle \right| + \left| \langle \sigma_{s,r,i}^{(t)} \boldsymbol{\xi}_i \|\boldsymbol{\xi}_i\|^{-2}, \boldsymbol{\xi}_i \rangle - \sigma_{s,r,i}^{(t)} \right| + \sum_{p \in J} \lambda_{s,r,p}^{(t)} \|\boldsymbol{\mu}_p\|^{-2} \left| \langle \boldsymbol{\mu_p}, \boldsymbol{\xi}_i \rangle \right| + \sum_{q \neq i} \sigma_{s,r,q}^{(t)} \|\boldsymbol{\xi}_q\|^{-2} \left| \langle \boldsymbol{\xi}_q, \boldsymbol{\xi}_i \rangle \right|$$

$$\leq \left| \langle \boldsymbol{w}_{s,r}^{(0)}, \boldsymbol{\xi}_i \rangle \right| + \sum_{p \in J} \lambda_{s,r,p}^{(t)} \frac{2\Delta}{d} + \sum_{q \neq i} \sigma_{s,r,q}^{(t)} \frac{2\Delta}{d}$$

$$= \left| \langle \boldsymbol{w}_{s,r}^{(0)}, \boldsymbol{\mu}_j \rangle \right| + \frac{2\Delta}{d} \left( \sum_{p \neq j} \sum_{q \in I_p} \sigma_{s,r,q}^{(t)} \frac{\|\boldsymbol{\xi}_q\|^2}{\|\boldsymbol{\mu}_p\|^2} + \sum_{q \in I} \sigma_{s,r,q}^{(t)} \right)$$

$$\leq \sqrt{d} \|\boldsymbol{w}_{s,r}^{(0)}\| + \frac{12\Delta \ln(t+1)k}{d} \leq \frac{\epsilon}{6}.$$

The first equation employs the weight decomposition in Lemma E.2; the second inequality expands the inner product and applies the triangle inequality; the third inequality utilizes the properties from Proposition D.6; the fourth equation utilizes Corollary E.4; the fifth inequality utilizes Property 1 and Property 8 in the inductive hypotheses. ∎

**Lemma E.13.** *Assuming the inductive hypotheses hold before time step $t$, for all $s \in \{-1, +1\}, r \in [m], i \in I$, we have*

$$\left| \langle \boldsymbol{w}_{s,r}^{(t)}, \boldsymbol{x}_i \rangle - \lambda_{s,r,c(i)}^{(t)} - \sigma_{s,r,i}^{(t)} \right| \leq \frac{\epsilon}{3}.$$

*Proof of Lemma E.13.* Using the conclusion in Lemma E.11 and Lemma E.12 and triangle inequality, we can directly obtain the conclusion in this lemma.

$$\left| \langle \boldsymbol{w}_{s,r}^{(t)}, \boldsymbol{x}_i \rangle - \lambda_{s,r,c(i)}^{(t)} - \sigma_{s,r,i}^{(t)} \right| \leq \left| \langle \boldsymbol{w}_{s,r}^{(t)}, \boldsymbol{\mu}_{c(i)} \rangle - \lambda_{s,r,c(i)}^{(t)} \right| + \left| \langle \boldsymbol{w}_{s,r}^{(t)}, \boldsymbol{\xi}_i \rangle - \sigma_{s,r,i}^{(t)} \right| \leq \frac{\epsilon}{3}.$$

∎

**Lemma E.14.** *Assuming the inductive hypotheses hold before time step $t$, for all $i \in I$, we have*

$$|q_i^{(t)} - \hat{q}_i^{(t)}| \leq \epsilon$$

*Proof of Lemma E.14.* Without loss of generality, we assume that $i \in I_+$.

Using Property 7 in the inductive hypotheses, we know that

$$
\begin{aligned}
|q_i - \hat{q}_i| = \frac{1}{m} \left| y_i f_{\theta^{(t)}}(\boldsymbol{x}_i) - \left( \sum_{r \in [m]} \lambda_{1,r,c(i)}^{(t)} + \sum_{r \in [m]} \sigma_{1,r,i}^{(t)} \right) \right| \\
\leq \frac{1}{m} \sum_{r \in [m]} \left| \langle \boldsymbol{w}_{1,r}^{(t)}, \boldsymbol{x}_i \rangle + b_{1,r}^{(t)} - \lambda_{1,r,c(i)}^{(t)} - \sigma_{1,r,i}^{(t)} \right| + \frac{1}{m} \sum_{r \in [m]} \mathrm{ReLU}\left( \langle \boldsymbol{w}_{-1,r}^{(t)}, \boldsymbol{x}_i \rangle + b_{-1,r}^{(t)} \right) \\
\leq \underbrace{\frac{1}{m} \sum_{r \in [m]} \left| \langle \boldsymbol{w}_{1,r}^{(t)}, \boldsymbol{x}_i \rangle - \lambda_{1,r,c(i)}^{(t)} - \sigma_{1,r,i}^{(t)} \right|}_{\mathcal{L}_1} + \underbrace{\frac{1}{m} \sum_{r \in [m]} \mathrm{ReLU}\left( \langle \boldsymbol{w}_{-1,r}^{(t)}, \boldsymbol{x}_i \rangle \right)}_{\mathcal{L}_2} + \underbrace{\frac{1}{m} \sum_{r \in [m]} \left( |b_{-1,r}^{(t)}| + |b_{1,r}^{(t)}| \right)}_{\mathcal{L}_3}.
\end{aligned}
$$

For $\mathcal{L}_1$ term, using the conclusion in Lemma E.13, we know that $\mathcal{L}_1 \leq \dfrac{\epsilon}{3}$.

For $\mathcal{L}_2$ term, we consider each term in the summation by distinguishing between two scenarios..

Case(I): $\langle \boldsymbol{w}_{-1,r}^{(t)}, \boldsymbol{x}_i \rangle \leq 0$.

Then we know that $\mathrm{ReLU}\left( \langle \boldsymbol{w}_{-1,r}^{(t)}, \boldsymbol{x}_i \rangle \right) = 0 < \dfrac{\epsilon}{3}$.

Case(II): $\langle \boldsymbol{w}_{-1,r}^{(t)}, \boldsymbol{x}_i \rangle \geq 0$.

Using the conclusion in Lemma E.13 and Corollary E.5, we know that

$$
\mathrm{ReLU}\left( \left\langle \boldsymbol{w}_{-1,r}^{(t)}, \boldsymbol{x}_i \right\rangle \right) = \langle \boldsymbol{w}_{-1,r}^{(t)}, \boldsymbol{x}_i \rangle \leq \langle \boldsymbol{w}_{-1,r}^{(t)}, \boldsymbol{x}_i \rangle - \lambda_{-1,r,c(i)}^{(t)} - \sigma_{-1,r,i}^{(t)} \leq \frac{\epsilon}{3}.
$$

Combining Case (I) and (II) together, we know that $\mathcal{L}_2 \leq \dfrac{\epsilon}{3}$.

For $\mathcal{L}_3$ term, using the conclusion in Lemma E.10, we know that $\mathcal{L}_3 \leq \dfrac{\epsilon}{3}$.

Combining the above together, we know that

$$
|q_i^{(t)} - \hat{q}_i^{(t)}| \leq \epsilon.
$$

■

Then, we can estimate the margin gap between two training data points using a simple triangle inequality.

**Corollary E.15.** *Assuming the inductive hypotheses hold before time step $t$, for all $i, j \in I$, we have*

$$
\left| \Delta_q^{(t)}(i,j) - \hat{\Delta}_q^{(t)}(i,j) \right| \leq 2\epsilon.
$$

*Proof of Corollary E.15.* By Lemma E.14 and triangle inequality, we have

$$
\left| \Delta_q^{(t)}(i,j) - \hat{\Delta}_q^{(t)}(i,j) \right| \leq |q_i^{(t)} - \hat{q}_i^{(t)}| + |q_j^{(t)} - \hat{q}_j^{(t)}| \leq 2\epsilon.
$$

■

Then we will analyze update equations for $\hat{\Delta}_q^{(t)}(i,j)$.

By Lemma E.9, we know that

$$
\sum_{r \in [m]} \left( \lambda_{y_i,r,c(i)}^{(t+1)} - \lambda_{y_j,r,c(j)}^{(t+1)} \right) = \sum_{r \in [m]} \left( \lambda_{y_i,r,c(i)}^{(t)} - \lambda_{y_j,r,c(j)}^{(t)} \right) - \frac{\eta}{2n} \left( \sum_{p \in I_{c(i)}} \ell_p'^{(t)} \|\boldsymbol{\mu}_{c(i)}\|^2 - \sum_{p \in I_{c(j)}} \ell_p'^{(t)} \|\boldsymbol{\mu}_{c(j)}\|^2 \right)
$$

$$
\sum_{r \in [m]} \left( \sigma_{y_i,r,i}^{(t+1)} - \sigma_{y_j,r,j}^{(t+1)} \right) = \sum_{r \in [m]} \left( \sigma_{y_i,r,i}^{(t)} - \sigma_{y_j,r,j}^{(t)} \right) - \frac{\eta}{2n} \left( \ell_i'^{(t)} \|\boldsymbol{\xi}_i\|^2 - \ell_j'^{(t)} \|\boldsymbol{\xi}_j\|^2 \right)
$$

Combining the above two equations together, we get the update equation for $\hat{\Delta}_q^{(t)}(i,j)$.

$$\hat{\Delta}_q^{(t+1)}(i,j) = \hat{\Delta}_q^{(t)}(i,j) - \frac{\eta}{2n}\left(\sum_{p \in I_{c(i)}} \ell_p'^{(t)}\|\boldsymbol{\mu}_{c(i)}\|^2 - \sum_{p \in I_{c(j)}} \ell_p'^{(t)}\|\boldsymbol{\mu}_{c(j)}\|^2 + \ell_i'^{(t)}\|\boldsymbol{\xi}_i\|^2 - \ell_j'^{(t)}\|\boldsymbol{\xi}_j\|^2\right)$$

(17)

**Lemma E.16** (Property 8). *For $s \in \{-1, +1\}, i \in I_s$, we have $|\lambda_{-s,r,c(i)}^{(t)}| \leq \epsilon, |\sigma_{-s,r,i}| \leq 2\epsilon$.*

*Proof of Lemma E.16.* We prove that for $j \in J^+, |\lambda_{-1,r,j}^{(t)}| \leq \epsilon$ for all $r \in [m]$ and the proof of the other part is similar.

We distinguish between two scenarios.

Case(I): For all $i \in I_j, \langle \boldsymbol{w}_{-1,r}^{(t)}, \boldsymbol{x}_i \rangle + b_{-1,r}^{(t)} < 0$.

Then by Corollary E.3 and inductive hypothesis, we know that

$$|\lambda_{-1,r,j}^{(t+1)}| = |\lambda_{-1,r,j}^{(t)}| \leq \epsilon.$$

Case(II): There exists $i \in I_j$ such that $\langle \boldsymbol{w}_{-1,r}^{(t)}, \boldsymbol{x}_i \rangle + b_{-1,r}^{(t)} \geq 0$.

By Lemma E.13, we know that

$$\langle \boldsymbol{w}_{-1,r}^{(t)}, \boldsymbol{x}_i \rangle - \lambda_{-1,r,j}^{(t)} - \sigma_{-1,r,i}^{(t)} \leq \frac{\epsilon}{3}.$$

Then by Lemma E.10 and noting that $\langle \boldsymbol{w}_{-1,r}^{(t)}, \boldsymbol{x}_i \rangle + b_{-1,r}^{(t)} \geq 0$ and $\sigma_{-1,r,i}^{(t)} \leq 0$, we have

$$\lambda_{-1,r,j}^{(t)} \geq \langle \boldsymbol{w}_{-1,r}^{(t)}, \boldsymbol{x}_i \rangle - \sigma_{-1,r,i}^{(t)} - \frac{\epsilon}{3} \geq -b_{-1,r}^{(t)} - \frac{\epsilon}{3} \geq -\frac{2\epsilon}{3}.$$

Then using the conclusion in Lemma E.3, we know that

$$|\lambda_{-1,r,j}^{(t+1)} - \lambda_{-1,r,j}^{(t)}| \leq \frac{\eta d}{nm}\sum_{i \in I_j}|\ell_i'^{(t)}| \leq \frac{\eta d}{m} \leq \frac{\epsilon}{3}.$$

Thus we have $\lambda_{-1,r,j}^{(t+1)} \geq -\epsilon$. Noting that $\lambda_{-1,r,j}^{(t+1)} \leq 0$, we have $|\lambda_{-1,r,j}^{(t+1)}| \leq \epsilon$. Then by Corollary E.4, we know that $|\sigma_{s,r,i}^{(t)}| \leq 2|\lambda_{s,r,j}^{(t+1)}| \leq 2\epsilon$. ∎

The lemmas we used for the inductive proof have all been proved, and now we can begin the main part of our proof.

**Property 2:** We first prove that Property 2 as the following lemma.

**Lemma E.17** (Property 2 of Lemma E.7). *Assuming the inductive hypotheses hold before time step $t$ and $c(i) = c(j)$, we have*

$$\left|\hat{\Delta}_q^{(t+1)}(i,j)\right| \leq 5\epsilon.$$

*Proof of Lemma E.17.* Without loss of generality, we assume that $\hat{q}_i^{(t+1)} \geq \hat{q}_j^{(t+1)}$. We distinguish between two scenarios., one is when $\left|\hat{\Delta}_q^{(t)}(i,j)\right|$ is relatively small and the other is when $\left|\hat{\Delta}_q^{(t)}(i,j)\right|$ is relatively large.

Case(I): $\hat{\Delta}_q^{(t)}(i,j) \leq 4\epsilon$.

By equation (17), Lemma D.15 and $\eta \leq d^{-2}$, we know that

$$\left|\hat{\Delta}_q^{(t+1)}(i,j) - \hat{\Delta}_q^{(t)}(i,j)\right|$$
$$= \frac{\eta}{2n}\left|\ell_i'^{(t)}\|\boldsymbol{\xi}_i\|^2 - \ell_j'^{(t)}\|\boldsymbol{\xi}_j\|^2\right| \leq \frac{\eta}{2n}\left(\left|\ell_i'^{(t)}\|\boldsymbol{\xi}_i\|^2\right| + \left|\ell_j'^{(t)}\|\boldsymbol{\xi}_j\|^2\right|\right) \leq \frac{2\eta d}{n} \leq \epsilon$$

So we have

$$\hat{\Delta}_q^{(t+1)}(i,j) \le 5\epsilon.$$

Case(II) : $\hat{\Delta}_q^{(t)}(i,j) \ge 4\epsilon$.

By Corollary E.15, we know that

$$\Delta_q^{(t)}(i,j) \ge \hat{\Delta}_q^{(t)}(i,j) - 2\epsilon \ge 2\epsilon.$$

By Lemma E.6, we know that

$$\frac{\ell_j'^{(t)}}{\ell_i'^{(t)}} \ge e^{\Delta_q^{(t)}(i,j)/2} \ge 1 + \Delta_q^{(t)}(i,j)/2 \ge 1 + \epsilon. \tag{18}$$

By Equation (17) and $c(i) = c(j)$, we know that

$$
\begin{aligned}
&\hat{\Delta}_q^{(t+1)}(i,j) - \hat{\Delta}_q^{(t)}(i,j) \\
&= -\frac{\eta}{2n}\left( \sum_{p \in I_{c(i)}} \ell_p'^{(t)}\|\boldsymbol{\mu}_{c(i)}\|^2 - \sum_{p \in I_{c(j)}} \ell_p'^{(t)}\|\boldsymbol{\mu}_{c(j)}\|^2 + \ell_i'^{(t)}\|\boldsymbol{\xi}_i\|^2 - \ell_j'^{(t)}\|\boldsymbol{\xi}_j\|^2 \right) \\
&= -\frac{\eta}{2n}\left( \ell_i'^{(t)}\|\boldsymbol{\xi}_i\|^2 - \ell_j'^{(t)}\|\boldsymbol{\xi}_j\|^2 \right) \\
&= -\frac{\eta}{2n}\|\boldsymbol{\xi}_j\|^2\ell_i'^{(t)}\left( \underbrace{(1+\epsilon) - \frac{\ell_j'^{(t)}}{\ell_i'^{(t)}}}_{<0,\text{ by inequality (18)}} \right) + \frac{\eta}{2n}\ell_i'^{(t)}\left( (1+\epsilon)\|\boldsymbol{\xi}_j\|^2 - \|\boldsymbol{\xi}_i\|^2 \right) \\
&\le\ 0
\end{aligned} \tag{19}
$$

Furthermore, due to the inductive hypothesis,

$$\hat{\Delta}_q^{(t+1)}(i,j) \le \hat{\Delta}_q^{(t)}(i,j) \le 5\epsilon.$$

$\blacksquare$

**Property 5:** Using the result in this lemma and Corollary E.15, we know that

$$\Delta_q^{(t+1)}(i,j) \le 7\epsilon.$$

Using the above inequality and Lemma E.6 and noting that $e^x \le 1 + 2x$ for small $x$ we know that

$$\frac{\ell_j'^{(t+1)}}{\ell_i'^{(t+1)}} \le e^{\Delta_q^{(t+1)}(i,j)} \le 1 + 2\Delta_q^{(t+1)}(i,j) \le 1 + 14\epsilon.$$

At this point, we have completed the inductive proofs for Property 2 and 5 in Lemma E.7.

**Property 3, 4 and 6:** Next, we consider the general case where the two training data points $\boldsymbol{x}_i, \boldsymbol{x}_j$ are not necessarily in the same cluster to prove Property 3 and 6 in Lemma E.7. This part of the proof overlaps significantly with the previous one, with the main difference being the addition of an extra term in the update equation of $\hat{\Delta}_q^{(t)}(i,j)$.

**Lemma E.18** (Property 3). *Assuming the inductive hypotheses hold before time step t, we have*

$$\left| \hat{\Delta}_q^{(t+1)}(i,j) \right| \le 63\epsilon.$$

*Proof of Lemma E.18.* We distinguish between two scenarios, one is when $\left| \hat{\Delta}_q^{(t)}(i,j) \right|$ is relative small and the other is when $\left| \hat{\Delta}_q^{(t)}(i,j) \right|$ is relative large.

Case(I): $\hat{\Delta}_q^{(t)}(i,j) \le 62\epsilon$.

By equation (17), Lemma D.15 and $\eta \le d^{-2}$, we know that

$$\left| \hat{\Delta}_q^{(t+1)}(i,j) - \hat{\Delta}_q^{(t)}(i,j) \right|$$

$$= \frac{\eta}{2n} \left| \sum_{p \in I_{c(i)}} \ell_p'^{(t)} \|\boldsymbol{\mu}_{c(i)}\|^2 - \sum_{p \in I_{c(j)}} \ell_p'^{(t)} \|\boldsymbol{\mu}_{c(j)}\|^2 + \ell_i'^{(t)} \|\boldsymbol{\xi}_i\|^2 - \ell_j'^{(t)} \|\boldsymbol{\xi}_j\|^2 \right|$$

$$\le \frac{\eta}{2n} \left( \sum_{p \in I_{c(i)}} \ell_p'^{(t)} \|\boldsymbol{\mu}_{c(i)}\|^2 + \sum_{p \in I_{c(j)}} \ell_p'^{(t)} \|\boldsymbol{\mu}_{c(j)}\|^2 + \left| \ell_i'^{(t)} \|\boldsymbol{\xi}_i\|^2 \right| + \left| \ell_j'^{(t)} \|\boldsymbol{\xi}_j\|^2 \right| \right)$$

$$\le \frac{\eta}{2n} \left( \frac{2nd}{k} + \frac{2nd}{k} + 2d + 2d \right) \le \frac{5\eta d}{2k} \le \epsilon$$

So we have

$$\hat{\Delta}_q^{(t+1)}(i,j) \le 63\epsilon.$$

Case(II): $\hat{\Delta}_q^{(t)}(i,j) \ge 62\epsilon.$

By Lemma E.14, we know that

$$\Delta_q^{(t)}(i,j) \ge \hat{\Delta}_q^{(t)}(i,j) - 2\epsilon \ge 60\epsilon.$$

By Lemma E.6, we know that

$$\frac{\ell_j'^{(t)}}{\ell_i'^{(t)}} \ge e^{\Delta_q^{(t)}(i,j)/2} \ge 1 + \Delta_q^{(t)}(i,j)/2 \ge 1 + 30\epsilon. \tag{20}$$

Furthermore, due to the inductive hypothesis, for any $p \in I_{c(i)}, q \in I_{c(j)}$, we know that

$$\frac{\ell_p'^{(t)}}{\ell_q'^{(t)}} \le \frac{(1+14\epsilon)\ell_i'^{(t)}}{\ell_j'^{(t)}/(1+14\epsilon)} \le \frac{(1+14\epsilon)^2}{1+30\epsilon} \le \frac{1}{1+\epsilon}. \tag{21}$$

By Equation 17 and $c(i) = c(j)$, we know that

$$\hat{\Delta}_q^{(t+1)}(i,j) - \hat{\Delta}_q^{(t)}(i,j)$$

$$= -\frac{\eta}{2n} \left( \sum_{p \in I_{c(i)}} \ell_p'^{(t)} \|\boldsymbol{\mu}_{c(i)}\|^2 - \sum_{p \in I_{c(j)}} \ell_p'^{(t)} \|\boldsymbol{\mu}_{c(j)}\|^2 + \ell_i'^{(t)} \|\boldsymbol{\xi}_i\|^2 - \ell_j'^{(t)} \|\boldsymbol{\xi}_j\|^2 \right)$$

$$= -\frac{\eta}{2n} \left( \ell_i'^{(t)} \|\boldsymbol{\xi}_i\|^2 - \ell_j'^{(t)} \|\boldsymbol{\xi}_j\|^2 \right) - \frac{\eta}{2n} \left( \sum_{p \in I_{c(i)}} \ell_p'^{(t)} \|\boldsymbol{\mu}_{c(i)}\|^2 - \sum_{p \in I_{c(j)}} \ell_p'^{(t)} \|\boldsymbol{\mu}_{c(j)}\|^2 \right)$$

We analyze each of these two terms separately.

Similar to the proof of inequality (19),

$$-\frac{\eta}{2n} \left( \ell_i'^{(t)} \|\boldsymbol{\xi}_i\|^2 - \ell_j'^{(t)} \|\boldsymbol{\xi}_j\|^2 \right)$$

$$= -\frac{\eta}{2n} \|\boldsymbol{\xi}_j\|^2 \ell_i'^{(t)} \left( \underbrace{(1+30\epsilon) - \frac{\ell_j'^{(t)}}{\ell_i'^{(t)}}}_{<0, \text{ by inequality } (20)} \right) + \frac{\eta}{2n} \ell_i'^{(t)} \left( (1+30\epsilon) \|\boldsymbol{\xi}_j\|^2 - \|\boldsymbol{\xi}_i\|^2 \right)$$

$$\le 0 \tag{22}$$

By inequality (21) and Property 7 in Proposition D.6, we have

$$\frac{\sum\limits_{p \in I_{c(i)}} \ell_p'^{(t)} \|\boldsymbol{\mu}_{c(i)}\|^2}{\sum\limits_{q \in I_{c(j)}} \ell_q'^{(t)} \|\boldsymbol{\mu}_{c(i)}\|^2} = \frac{\sum\limits_{p \in I_{c(i)}} \ell_p'^{(t)}}{\sum\limits_{q \in I_{c(j)}} \ell_q'^{(t)}} \le \frac{|I_i|}{(1+\epsilon)|I_j|} \le 1.$$

Thus we know that

$$-\frac{\eta}{2n}\left(\sum_{p\in I_{c(i)}}\ell_p'^{(t)}\|\boldsymbol{\mu}_{c(i)}\|^2 - \sum_{p\in I_{c(j)}}\ell_p'^{(t)}\|\boldsymbol{\mu}_{c(j)}\|^2\right)\leq 0. \tag{23}$$

By combining (22) and (23) with the inductive hypothesis, we know that

$$\hat{\Delta}_q^{(t+1)}(i,j)\leq \hat{\Delta}_q^{(t)}(i,j)\leq 63\epsilon.$$

∎

Using the result in this lemma and Corollary E.15, we know that

$$\Delta_q^{(t+1)}(i,j)\leq 65\epsilon.$$

Using the above inequality and Lemma E.6 and noting that $e^x\leq 1+2x$ for small $x$ we know that

$$\frac{y_j\ell_j'^{(t+1)}}{y_i\ell_i'^{(t+1)}}\leq e^{\Delta_q^{(t+1)}(i,j)}\leq 1+2\Delta_q^{(t+1)}(i,j)\leq 1+130\epsilon.$$

Now, we have completed the inductive proofs for Property 2, 3, 4, 5 and 6 in Lemma E.7.

**Property 1:** To prove Property 1, we need to analyze the update equation for $\sigma_{s,r,i}^{(t)}$. We first prove that $\sigma_{s,r,i}^{(t)}$'s are balanced as follows.

**Lemma E.19.** *For all $i_1,i_2\in I, r_1,r_2\in[m]$, we have*

$$(1-200\epsilon)\,\sigma_{y_{i_2},r_2,i_2}^{(t)} - \frac{1}{nd}\leq \sigma_{y_{i_1},r_1,i_1}^{(t)}\leq (1+200\epsilon)\,\sigma_{y_{i_2},r_2,i_2}^{(t)} + \frac{1}{nd}.$$

*Proof of Lemma E.19.* We first prove the right-hand side of the inequality and the proof for the left-hand side is similar. By Lemma E.9, for any $t'<t$, we know that

$$\frac{\sigma_{y_{i_1},r_1,i_1}^{(t'+1)} - \sigma_{y_{i_1},r_1,i_1}^{(t')}}{\sigma_{y_{i_2},r_2,i_2}^{(t'+1)} - \sigma_{y_{i_2},r_2,i_2}^{(t')}} = \frac{y_{i_1}\ell_{i_1}'^{(t')}\|\boldsymbol{\xi_{i_1}}\|^2}{y_{i_2}\ell_{i_2}'^{(t')}\|\boldsymbol{\xi_{i_2}}\|^2}$$

$$\leq (1+130\epsilon)\left(\frac{\sqrt{d}+\ln(d)}{\sqrt{d}-\ln(d)}\right)^2 \quad \text{(Applying Property 1 in Proposition D.6)}$$

$$\leq 1+200\epsilon$$

That is to say

$$\sigma_{y_{i_1},r_1,i_1}^{(t'+1)} - \sigma_{y_{i_1},r_1,i_1}^{(t')}\leq (1+200\epsilon)\left(\sigma_{y_{i_2},r_2,i_2}^{(t'+1)} - \sigma_{y_{i_2},r_2,i_2}^{(t')}\right).$$

Summing the above inequality from $t'=1$ to $t'=t-1$, we have

$$\sigma_{y_{i_1},r_1,i_1}^{(t)} - \sigma_{y_{i_1},r_1,i_1}^{(1)}\leq (1+200\epsilon)\left(\sigma_{y_{i_2},r_2,i_2}^{(t)} - \sigma_{y_{i_2},r_2,i_2}^{(1)}\right).$$

Then, we can derive that

$$\sigma_{y_{i_1},r_1,i_1}^{(t)}\leq (1+200\epsilon)\,\sigma_{y_{i_2},r_2,i_2}^{(t)} + \sigma_{y_{i_1},r_1,i_1}^{(1)}$$

$$= (1+200\epsilon)\,\sigma_{y_{i_2},r_2,i_2}^{(t)} - \frac{\eta}{nm}\cdot\ell_i'^{(t)}\|\boldsymbol{\xi_i}\|^2\mathbb{1}\left(\langle\boldsymbol{w}_{s,r}^{(t)},\boldsymbol{x}_i\rangle + b_{s,r}^{(t)}\geq 0\right)$$

$$\leq (1+200\epsilon)\,\sigma_{y_{i_2},r_2,i_2}^{(t)} + \frac{2d\eta}{nm}$$

$$\leq (1+200\epsilon)\,\sigma_{y_{i_2},r_2,i_2}^{(t)} + \frac{1}{nd}.$$

Reusing the logic of the above proof, we know that

$$(1-200\epsilon)\,\sigma_{y_{i_2},r_2,i_2}^{(t)} - \frac{1}{nd}\leq \sigma_{y_{i_1},r_1,i_1}^{(t)}.$$

∎

Then, we estimate the margin $q_i^{(t)}$ only using $\sigma_{s,r,i}^{(t)}$, which is presented as the following lemma.

**Lemma E.20.** *For every $i \in I, r_0 \in [m]$, we have $\frac{n}{2k}\sigma_{y_i,r_0,i}^{(t)} - 2\epsilon \le q_i^{(t)} \le \frac{2n}{k}\sigma_{y_i,r_0,i}^{(t)} + 2\epsilon$.*

*Proof of Lemma E.20.* Without loss of generality, we assume that $i \in I_+$.

We first prove the right-hand side of the inequality.

$$
\begin{aligned}
m\hat{q}_i^{(t)} &= \sum_{r \in [m]} \lambda_{1,r,c(i)}^{(t)} + \sum_{r \in [m]} \sigma_{1,r,i}^{(t)} \\
&= \sum_{r \in [m]} \sum_{p \in I_{c(i)}} \frac{\|\boldsymbol{\xi}_i\|^{-2}}{\|\boldsymbol{\mu}_{c(i)}\|^{-2}} \sigma_{1,r,p}^{(t)} + \sum_{r \in [m]} \sigma_{1,r,i}^{(t)} \\
&\le \sum_{r \in [m]} \sum_{p \in I_{c(i)}} \frac{\|\boldsymbol{\xi}_i\|^{-2}}{\|\boldsymbol{\mu}_{c(i)}\|^{-2}} \left( (1 + 200\epsilon)\sigma_{1,r_0,i}^{(t)} + \frac{1}{nd} \right) + \sum_{r \in [m]} \left( (1 + 200\epsilon)\sigma_{1,r_0,i}^{(t)} + \frac{1}{nd} \right) \\
&\le \frac{2mn}{k}\sigma_{1,r_0,i}^{(t)} + \frac{2m}{kd} \\
&\le \frac{2mn}{k}\sigma_{1,r_0,i}^{(t)} + m\epsilon.
\end{aligned}
$$

Then, by Lemma E.14, we know that

$$
q_i^{(t)} \le \hat{q}_i^{(t)} + \epsilon \le \frac{2n}{k}\sigma_{1,r_0,i}^{(t)} + 2\epsilon.
$$

Reusing the argument of the above proof, we prove the left-hand side of the inequality.

$$
\begin{aligned}
m\hat{q}_i^{(t)} &= \sum_{r \in [m]} \lambda_{1,r,c(i)}^{(t)} + \sum_{r \in [m]} \sigma_{1,r,i}^{(t)} \\
&= \sum_{r \in [m]} \sum_{p \in I_{c(i)}} \frac{\|\boldsymbol{\xi}_i\|^{-2}}{\|\boldsymbol{\mu}_{c(i)}\|^{-2}} \sigma_{1,r,p}^{(t)} + \sum_{r \in [m]} \sigma_{1,r,i}^{(t)} \\
&\ge \sum_{r \in [m]} \sum_{p \in I_{c(i)}} \frac{\|\boldsymbol{\xi}_i\|^{-2}}{\|\boldsymbol{\mu}_{c(i)}\|^{-2}} \left( (1 - 200\epsilon)\sigma_{1,r_0,i}^{(t)} - \frac{1}{nd} \right) + \sum_{r \in [m]} \left( (1 - 200\epsilon)\sigma_{1,r_0,i}^{(t)} - \frac{1}{nd} \right) \\
&\ge \frac{mn}{2k}\sigma_{1,r_0,i}^{(t)} - \frac{m}{2kd} \\
&\ge \frac{mn}{2k}\sigma_{1,r_0,i}^{(t)} - m\epsilon.
\end{aligned}
$$

Then by Lemma E.14, we know that

$$
q_i^{(t)} \ge \hat{q}_i^{(t)} - \epsilon \ge \frac{n}{2k}\sigma_{1,r_0,i}^{(t)} - 2\epsilon.
$$

∎

Furthermore, we also need to estimate $\ell_i'^{(t)}$ using $\sigma_{s,r,i}^{(t)}$ as the following lemma.

**Lemma E.21.** *For every $i \in I, r \in [m]$, we have*

$$
\frac{1}{3}\exp\left( -\frac{2n}{k}\sigma_{y_i,r,i}^{(t)} \right) \le -y_i\ell_i'^{(t)} \le 2\exp\left( -\frac{n}{2k}\sigma_{y_i,r,i}^{(t)} \right).
$$

*Proof of Lemma E.21.* Without loss of generality, we assume that $i \in I_+$.

By Lemma E.20, we know that

$$
-y_i\ell_i'^{(t)} = \frac{1}{1 + \exp\left( q_i^{(t)} \right)} \ge \frac{1}{2\exp\left( q_i^{(t)} \right)} \ge \frac{1}{2}\exp\left( -\frac{2n}{k}\sigma_{1,r,i}^{(t)} - 2\epsilon \right) \ge \frac{1}{3}\exp\left( -\frac{2n}{k}\sigma_{1,r,i}^{(t)} \right).
$$

$$-y_i \ell_i'^{(t)} = \frac{1}{1 + \exp\left(q_i^{(t)}\right)} \leq \frac{1}{\exp\left(q_i^{(t)}\right)} \leq \exp\left(-\frac{n}{2k}\sigma_{1,r,i}^{(t)} + 2\epsilon\right) \leq 2\exp\left(-\frac{2n}{k}\sigma_{1,r,i}^{(t)}\right).$$

∎

Then, we can prove Property 1 based on the inductive hypothesis.

Without loss of generality, we assume that $i \in I_+$.

We first prove the left-hand side of the inequality.

$$\begin{aligned}
\sigma_{1,r,i}^{(t+1)} &= \sigma_{1,r,i}^{(t)} - \frac{\eta}{nm} \cdot \ell_i'^{(t)} \|\boldsymbol{\xi_i}\|^2 \\
&\leq \sigma_{1,r,i}^{(t)} + \frac{2\eta d}{nm} \exp\left(-\frac{n}{2k}\sigma_{1,r,i}^{(t)}\right) && \text{(Applying Lemma E.21)} \\
&\leq \frac{2k}{n}\ln(t+1) + \frac{2\eta d}{nm}\frac{1}{t+1} && \text{(Monotone with respect to } \sigma_{1,r,i}^{(t)}) \\
&\leq \frac{2k}{n}\ln(t+2),
\end{aligned}$$

$$\sigma_{-1,r,i}^{(t+1)} \leq 0 \leq \frac{2k}{n}\ln(t+2) \text{(Corollary E.5)}.$$

Then we prove the right-hand side of the inequality.

$$\begin{aligned}
\sigma_{1,r,i}^{(t+1)} &= \sigma_{1,r,i}^{(t)} - \frac{\eta}{nm} \cdot \ell_i'^{(t)} \|\boldsymbol{\xi_i}\|^2 \\
&\geq \sigma_{1,r,i}^{(t)} + \frac{\eta d}{3nm} \exp\left(-\frac{2n}{k}\sigma_{1,j,i}^{(t)}\right) && \text{(Applying Lemma E.21)} \\
&\geq \frac{k}{2n}\left(\ln(t) + \ln(\eta)\right) + \frac{d}{3nm}\frac{1}{t} && \text{(Monotonic with respect to } \sigma_{1,r,i}^{(t)}) \\
&\geq \frac{k}{2n}\left(\ln\left((t+1)\right) + \ln(\eta)\right) \\
&= \frac{k}{2n}\ln((t+1)\eta).
\end{aligned}$$

Finally, we prove Property 7. The proof is very similar to the proof of Lemma E.1. We show that $\langle \boldsymbol{w}_{1,r}^{(t+1)}, \boldsymbol{x}_i \rangle + b_{1,r}^{(t+1)} \geq 0$ for all $i \in I_+$. By the inductive hypothesis, we know that $\langle \boldsymbol{w}_{1,r}^{(t)}, \boldsymbol{x}_i \rangle + b_{1,r}^{(t)} \geq 0$.

$$\begin{aligned}
\langle \boldsymbol{w}_{1,r}^{(t+1)}, \boldsymbol{x}_i \rangle + b_{1,r}^{(t+1)} &= \langle \boldsymbol{w}_{1,r}^{(t)}, \boldsymbol{x}_i \rangle + b_{1,r}^{(t)} - \eta\left(\langle \nabla_{\boldsymbol{w}_{1,r}}\mathcal{L}(\boldsymbol{\theta}^{(t)}), \boldsymbol{x}_i \rangle + \nabla_{b_{1,r}}\mathcal{L}(\boldsymbol{\theta}^{(t)})\right) \\
&\geq -\eta\left(\left\langle \nabla_{\boldsymbol{w}_{1,r}}\mathcal{L}(\boldsymbol{\theta}^{(t)}), \boldsymbol{x}_i \right\rangle + \nabla_{b_{1,r}}\mathcal{L}(\boldsymbol{\theta}^{(t)})\right).
\end{aligned}$$

Denote $\ell'^{(t)} := \ell_1'^{(t)}$. By Property (6) in inductive hypotheses, we know that for all $i \in I$

$$\frac{1}{1+130\epsilon}\ell_i'^{(t)} \leq \ell'^{(t)} \leq (1+130\epsilon)\ell_i'^{(t)}$$

We examine the update of linear term first.

$$\begin{aligned}
-\langle \nabla_{\boldsymbol{w}_{1,r}}\mathcal{L}(\boldsymbol{\theta}^{(t)}), \boldsymbol{x}_i \rangle &= -\langle \sum_{p\in\mathcal{I}_+} \ell_p'^{(t)}\boldsymbol{x}_p, \boldsymbol{x}_i \rangle \\
&\geq -\sum_{p\in I_{c(i)}} \ell_p'^{(t)}\langle \boldsymbol{x}_p, \boldsymbol{x_i} \rangle + \sum_{p\notin I_{c(i)}} \ell_p'^{(t)}|\langle \boldsymbol{x}_p, \boldsymbol{x_i} \rangle| \\
&\geq -\frac{d}{2}\sum_{p\in I_{c(i)}} \ell_p'^{(t)} + \Delta\sum_{p\notin I_{c(i)}} \ell_p'^{(t)} \\
&\geq -\frac{dn\ell'^{(t)}}{4k} + 2n\Delta\ell'^{(t)} \\
&\geq -n\Delta\ell'^{(t)}.
\end{aligned}$$
(24)

Then we examine the update of bias term.

$$-\nabla_{b_{1,r}} \mathcal{L}(\boldsymbol{\theta}^{(t)}) = -\sum_{p\in\mathcal{I}} \ell_p^{\prime(t)} \mathbb{1}\left(\langle \boldsymbol{w}_{1,r}^{(t)}, \boldsymbol{x}_i\rangle + b_{1,r}^{(t)} \geq 0\right) \geq n\ell^{\prime(t)}. \tag{25}$$

Combining (24) and (25) together, we know that

$$\langle \boldsymbol{w}_{1,r}^{(t+1)}, \boldsymbol{x}_i\rangle + b_{1,r}^{(t+1)} \geq 0$$

Thus we know that $S_{1,i}^{(t+1)} = [m]$.

For the case when $\boldsymbol{x}_i$ belongs to the negative class, we can obtain $S_{-1,i}^{(t+1)} = [m]$ using the same argument. Now, we have completed the proof of Lemma E.7. $\qquad\square$

### E.2 PROOF OF THEOREM 4.5

Now, we start to prove the main result Theorem 4.5.

**Theorem E.22** (Restatement of **Theorem** 4.5). *In the setting of training a two-layer ReLU network on the binary classification problem $\mathcal{D}(\{\boldsymbol{\mu}_j\}_{j=1}^k, J_\pm)$ as described in Section 3, under Assumptions 3.2, 3.3 and 4.3, for some $\gamma = o(1)$, after $\Omega(\eta^{-1}) \leq T \leq \exp(\tilde{O}(k^{1/2}))$ iterations, with probability at least $1 - \gamma$, the neural network satisfies the following properties:*

1. *The clean accuracy is nearly perfect:* $\mathrm{Acc}_{\mathrm{clean}}^{\mathcal{D}}(f_{\boldsymbol{\theta}^{(T)}}) \geq 1 - \exp(-\Omega(\log^2 d))$.

2. *Gradient descent leads the network to the feature-averaging regime: there exists a time-variant coefficient $\lambda^{(T)} \in [\Omega(1), +\infty)$ such that for all $s \in \{\pm 1\}$, $r \in [m]$, the weight vector $\boldsymbol{w}_{s,r}^{(T)}$ can be approximated as*

$$\left\|\boldsymbol{w}_{s,r}^{(T)} - \lambda^{(T)} \sum_{j\in J_s} \|\boldsymbol{\mu}_j\|^{-2}\boldsymbol{\mu}_j\right\| \leq o(d^{-1/2}),$$

   *and the bias term keeps sufficiently small, i.e., $\left|b_{s,r}^{(T)}\right| \leq o(1)$.*

3. *Consequently, the network is non-robust: for perturbation radius $\delta = \Omega(\sqrt{d/k})$, the $\delta$-robust accuracy is nearly zero, i.e., $\mathrm{Acc}_{\mathrm{robust}}^{\mathcal{D}}(f_{\boldsymbol{\theta}^{(T)}}; \delta) \leq \exp(-\Omega(\log^2 d))$.*

*Proof of Theorem E.22.* We first prove that gradient descent leads the network to the feature-averaging regime (Property 2).

**Lemma E.23.** *For all $s \in \{-1, +1\}, j \in J_s, r \in [m]$, we have $\dfrac{\ln(T\eta)}{4} \leq \lambda_{s,r,j}^{(T)} \leq 4\ln(T+1)$.*

*Proof of Lemma E.23.* Without loss of generality, we assume that $s = 1$.

Using Property 1 in Lemma E.7 and Corollary E.4, we know that

$$\lambda_{1,r,j}^{(T)} = \sum_{p\in I_j} \frac{\|\boldsymbol{\xi}_p\|^2}{\|\boldsymbol{\mu}_j\|^2}\sigma_{1,r,p}^{(T)} \geq \frac{n}{2k}\sigma_{1,r,1}^{(T)} \geq \frac{\ln(T\eta)}{4}$$

$$\lambda_{1,r,j}^{(T)} = \sum_{p\in I_j} \frac{\|\boldsymbol{\xi}_p\|^2}{\|\boldsymbol{\mu}_j\|^2}\sigma_{1,r,p}^{(T)} \leq \frac{2n}{k}\sigma_{1,r,1}^{(T)} \leq 4\ln(T+1)$$

$\blacksquare$

**Lemma E.24.** *For $r_1, r_2 \in [m], s_1, s_2 \in \{-1, +1\}, j_1 \in J_{s_1}, j_2 \in J_{s_2}$, we have*

$$\frac{\lambda_{s_1,r_1,j_1}^{(T)}}{\lambda_{s_2,r_2,j_2}^{(T)}} \leq 1 + 204\epsilon.$$

*Proof.* By Lemma E.19, we know that for any $i_1, i_2 \in I$

$$\sigma^{(t)}_{y_{i_1}, r_1, i_1} \le (1 + 200\epsilon)\, \sigma^{(t)}_{y_{i_2}, r_2, i_2} + (nd)^{-1} \le (1 + 201\epsilon)\sigma^{(t)}_{y_{i_1}, r_2, i_2}.$$

$$
\begin{aligned}
\frac{\lambda^{(T)}_{s_1, r_1, j_1}}{\lambda^{(T)}_{s_2, r_2, j_2}} &= \frac{\sum_{p \in I_{j_1}} \|\boldsymbol{\xi}_p\|^2 \sigma^{(t)}_{s_1, r_1, p}}{\sum_{p \in I_{j_2}} \|\boldsymbol{\xi}_p\|^2 \sigma^{(t)}_{s_2, r_2, p}} \\
&\le (1 + 201\epsilon) \frac{\|\sqrt{d} + \ln(d)\|^2}{\|\sqrt{d} - \ln(d)\|^2} \frac{|I_{j_1}|}{|I_{j_2}|} \\
&\le (1 + 201\epsilon)(1 + \epsilon)(1 + \epsilon) \\
&\le 1 + 204\epsilon.
\end{aligned}
$$

∎

We denote $\lambda^{(T)} = \lambda^{(T)}_{1,1,j_0}$ for some $j_0 \in J_+$ as the representative of $\{\lambda_{s,r,j} : s \in \{-1, +1\}, r \in [m], j \in J_s\}$.

By Lemma E.23 and Lemma E.24 ,for all $s \in \{-1, +1\}, r \in [m], j \in J_s$, we have

$$|\lambda^{(T)} - \lambda^{(T)}_{s,r,j}| \le 204\epsilon\lambda^{(T)},$$

$$\lambda^{(T)} \le 4\ln(T + 1),$$

$$\lambda^{(T)} \ge \frac{\ln(T\eta)}{4} = \Omega(1).$$

**Lemma E.25.** *For all $s \in \{-1, +1\}, r \in [m]$, We have*

$$\sqrt{d} \left\| \boldsymbol{w}^{(T)}_{s,r} - \lambda^{(T)} \sum_{j \in J_s} \boldsymbol{\mu}_j \|\boldsymbol{\mu}_j\|^{-2} \right\| = o(1).$$

*Proof of Lemma E.25.* Recall the weight decomposition in Lemma E.2.

$$
\sqrt{d} \left( \boldsymbol{w}^{(T)}_{s,r} - \lambda^{(T)} \sum_{j \in J_s} \boldsymbol{\mu}_j \|\boldsymbol{\mu}_j\|^{-2} \right) = \underbrace{\sqrt{d} \boldsymbol{w}^{(0)}_{s,r}}_{\mathcal{L}_1} + \underbrace{\sqrt{d} \sum_{j \in J_s} \left( \lambda^{(T)}_{s,r,j} - \lambda^{(T)} \right) \boldsymbol{\mu}_j \|\boldsymbol{\mu}_j\|^{-2}}_{\mathcal{L}_2}
$$
$$
+ \underbrace{\sqrt{d} \sum_{j \in J_{-s}} \lambda^{(T)}_{s,r,j} \boldsymbol{\mu}_j \|\boldsymbol{\mu}_j\|^{-2}}_{\mathcal{L}_3} + \underbrace{\sqrt{d} \sum_{i \in I} \sigma^{(T)}_{s,r,i} \boldsymbol{\xi}_i \|\boldsymbol{\xi}_i\|^{-2}}_{\mathcal{L}_4}
$$

For $\mathcal{L}_1$ term, using the conclusion in Lemma D.10, we know that

$$\|\sqrt{d} \boldsymbol{w}^{(0)}_{s,r}\| \le 2d\sigma_w \le \epsilon = o(1)$$

For $\mathcal{L}_2$ term, using the conclusion in Lemma E.23, Lemma E.24 and noting that $\mu_j$ are pairwise orthogonal, we know that

$$
\begin{aligned}
\left\| \sqrt{d} \sum_{j \in J} \left( \lambda^{(T)}_{s,r,j} - \lambda^{(T)} \right) \boldsymbol{\mu}_j \|\boldsymbol{\mu}_j\|^{-2} \right\| &= \sqrt{\sum_{j \in J_s} \left( \lambda^{(T)}_{s,r,j} - \lambda^{(T)} \right)^2} \\
&\le \sqrt{\sum_{j \in J_s} (204\epsilon)^2 \left( \lambda^{(T)} \right)^2} \\
&\le 204\epsilon\sqrt{k}\lambda^{(T)} \\
&\le 816\epsilon\sqrt{k}\ln(T + 1) \\
&\le 900k\epsilon = o(1).
\end{aligned}
$$

For $\mathcal{L}_3$ term, by Lemma E.16 and triangle inequality, we know that

$$\left\| \sqrt{d} \sum_{j \in J_{-s}} \lambda_{s,r,j}^{(T)} \boldsymbol{\mu}_j \|\boldsymbol{\mu}_j\|^{-2} \right\| \le k\epsilon = o(1).$$

For $\mathcal{L}_4$ term, by Property (1) in Lemma E.7, we have

$$
\begin{aligned}
\left\| \sqrt{d} \sum_{i \in I} \sigma_{s,r,i}^{(T)} \boldsymbol{\xi}_i \|\boldsymbol{\xi}_i\|^{-2} \right\|^2 &= d \sum_{i \in I} \left( \sigma_{s,r,i}^{(T)} \right)^2 \|\boldsymbol{\xi}_i\|^{-2} + d \sum_{i_1 \neq i_2} \sigma_{s,r,i_1}^{(T)} \sigma_{s,r,i_2}^{(T)} \langle \boldsymbol{\xi}_{i_1}, \boldsymbol{\xi}_{i_2} \rangle \|\boldsymbol{\xi}_i\|^{-4} \\
&\le 2 \sum_{i \in I} \left( \sigma_{s,r,i_1}^{(T)} \right)^2 + \frac{2\Delta}{d} \sum_{i_1 \neq i_2} \sigma_{s,r,i_1}^{(T)} \sigma_{s,r,i_2}^{(T)} \\
&\le \frac{8k^2 \ln^2(T+1)}{n} + \frac{8k^2 \ln^2(T+1)\Delta}{d} \\
&\le \frac{8k^3}{n} + \frac{8k^3 \Delta}{d} \le 16k\epsilon = o(1).
\end{aligned}
$$

Combining the above together, we know that

$$\sqrt{d} \left\| \boldsymbol{w}_{s,r}^{(T)} - \lambda^{(T)} \sum_{j \in J_s} \boldsymbol{\mu}_j \|\boldsymbol{\mu}_j\|^{-2} \right\| = o(1).$$

$\blacksquare$

By Lemma E.10, we know that $|b_{s,r}^{(T)}| \le \epsilon$.

Then, we prove that the clean accuracy is nearly perfect (Property 1). We first need to prove the following lemma, which shows that the correlation between network weight and random noise is small.

**Lemma E.26.** *Let $\boldsymbol{\xi} \sim \mathcal{N}(0, I_d)$. Then, with probability at least $1 - 2nd^{-\ln(d)/2}$, for all $s \in \{-1, +1\}, r \in [m]$ we have*

$$|\langle \boldsymbol{w}_{s,r}^{(t)}, \boldsymbol{\xi} \rangle| \le \frac{\epsilon}{6}.$$

*Proof of Lemma E.26.* Reusing the argument of proof of Property (3) and (4) in Proposition D.6. We know that with probability at least $1 - 2nd^{-\ln(d)/2}$, for all $i \in I, j \in J$,

$$|\langle \boldsymbol{\mu}_j, \boldsymbol{\xi} \rangle| \le \Delta, |\langle \boldsymbol{\xi}_i, \boldsymbol{\xi} \rangle| \le \Delta.$$

The remaining part of the proof is similar to the proof of Lemma E.11 and Lemma E.12.

$$
\begin{aligned}
\left| \langle \boldsymbol{w}_{s,r}^{(t)}, \boldsymbol{\xi} \rangle \right| &= \left| \left\langle \boldsymbol{w}_{s,r}^{(0)} + \sum_{p \in J} \lambda_{s,r,p}^{(t)} \boldsymbol{\mu}_p \|\boldsymbol{\mu}_p\|^{-2} + \sum_{q \in I} \sigma_{s,r,q}^{(t)} \boldsymbol{\xi}_q \|\boldsymbol{\xi}_q\|^{-2}, \boldsymbol{\xi} \right\rangle \right| \\
&\le \left| \langle \boldsymbol{w}_{s,r}^{(0)}, \boldsymbol{\mu}_j \rangle \right| + \sum_{p \in J} \lambda_{s,r,p}^{(t)} \|\boldsymbol{\mu}_p\|^{-2} |\langle \boldsymbol{\mu}_p, \boldsymbol{\mu}_j \rangle| + \sum_{q \in I} \sigma_{s,r,q}^{(t)} \|\boldsymbol{\xi}_q\|^{-2} |\langle \boldsymbol{\xi}_q, \boldsymbol{\mu}_j \rangle| \\
&\le \left| \langle \boldsymbol{w}_{s,r}^{(0)}, \boldsymbol{\mu}_j \rangle \right| + \frac{2\Delta}{d} \left( \sum_{p \in J} \lambda_{s,r,p}^{(t)} + \sum_{q \in I} \sigma_{s,r,q}^{(t)} \right) \le \frac{\epsilon}{6}.
\end{aligned}
$$

$\blacksquare$

Assume $(\boldsymbol{x}, y)$ is randomly sampled from the data distribution $\mathcal{D}$. Without loss of generality, we assume that $\boldsymbol{x} = \boldsymbol{\mu}_j + \boldsymbol{\xi}, y = 1$. Using the conclusion in Lemma E.11, Lemma E.26 and Lemma

E.10, we know that

$$
\begin{aligned}
\langle \boldsymbol{w}_{1,r}^{(T)}, \boldsymbol{x} \rangle + b_{1,r}^{(T)} &= \langle \boldsymbol{w}_{1,r}^{(T)}, \boldsymbol{\mu}_j \rangle + \langle \boldsymbol{w}_{1,r}^{(T)}, \boldsymbol{\xi} \rangle + b_{1,r}^{(T)} \\
&\geq \lambda_{1,r,j}^{(T)} - \frac{\epsilon}{6} - \frac{\epsilon}{6} - \frac{\epsilon}{3} \\
&\geq \lambda_{1,r,j}^{(T)} - \epsilon.
\end{aligned}
$$

$$
\begin{aligned}
\langle \boldsymbol{w}_{-1,r}^{(T)}, \boldsymbol{x} \rangle + b_{1,r}^{(T)} &= \langle \boldsymbol{w}_{-1,r}^{(T)}, \boldsymbol{\mu}_j \rangle + \langle \boldsymbol{w}_{-1,r}^{(T)}, \boldsymbol{\xi} \rangle + b_{-1,r}^{(T)} \\
&\leq \lambda_{-1,r,j}^{(T)} + \frac{\epsilon}{6} + \frac{\epsilon}{6} + \frac{\epsilon}{3} \\
&\leq \lambda_{-1,r,j}^{(T)} + \epsilon \\
&\leq \epsilon.
\end{aligned}
$$

Then, we have

$$
\begin{aligned}
f_{\boldsymbol{\theta}^{(T)}}(\boldsymbol{x}) &= \frac{1}{m} \sum_{r \in [m]} \mathrm{ReLU}\left( \langle \boldsymbol{w}_{1,r}^{(T)}, \boldsymbol{x} \rangle + b_{1,r}^{(T)} \right) - \frac{1}{m} \sum_{r \in [m]} \mathrm{ReLU}\left( \langle \boldsymbol{w}_{-1,r}^{(T)}, \boldsymbol{x} \rangle + b_{-1,r}^{(T)} \right) \\
&\geq \frac{1}{m} \sum_{r \in [m]} \left( \lambda_{1,r,j}^{(T)} - \epsilon \right) - \frac{1}{m} \sum_{r \in [m]} \epsilon \\
&= \frac{1}{m} \sum_{r \in [m]} \lambda_{1,r,j}^{(T)} - 2\epsilon \geq 0.
\end{aligned}
$$

Thus $f_{\boldsymbol{\theta}^{(T)}}$ has perfect standard accuracy.

Finally, we prove that the network is non-robust (Property 3).

We consider the following perturbation

$$
\boldsymbol{\rho} = -\frac{2(1+c)}{k} \left( \sum_{j \in J_+} \boldsymbol{\mu}_j - \sum_{j \in J_-} \boldsymbol{\mu}_j \right),
$$

where $c$ is a constant such that $c^{-1} \leq |J_+|/|J_-| \leq c$. This is to say $|J_+|, |J_-| \geq \dfrac{k}{1+c}$.

Then, we have

$$
\begin{aligned}
&\langle \boldsymbol{w}_{1,r}^{(T)}, \boldsymbol{x} + \boldsymbol{\rho} \rangle + b_{1,r}^{(T)} \\
\leq &\langle \boldsymbol{w}_{1,r}^{(T)}, \boldsymbol{\mu}_{j_0} \rangle + \langle \boldsymbol{w}_{1,r}^{(T)}, \boldsymbol{\xi} \rangle - \frac{2(1+c)}{k} \sum_{j \in J_+} \langle \boldsymbol{w}_{1,r}^{(T)}, \boldsymbol{\mu}_j \rangle + \frac{2(1+c)}{k} \sum_{j \in J_-} \langle \boldsymbol{w}_{1,r}^{(T)}, \boldsymbol{\mu}_j \rangle + \frac{\epsilon}{3} \\
\leq &\lambda_{1,r,j_0}^{(T)} + \frac{\epsilon}{6} + \frac{\epsilon}{6} - \frac{2(1+c)}{k} \sum_{j \in J_+} \left( \lambda_{1,r,j}^{(T)} - \frac{\epsilon}{6} \right) + \frac{2(1+c)}{k} \sum_{j \in J_-} \left( \lambda_{1,r,j}^{(T)} + \frac{\epsilon}{6} \right) + \frac{\epsilon}{3} \\
\leq &\lambda_{1,r,j_0}^{(T)} - \frac{2(1+c)}{k} \sum_{j \in J_+} \lambda_{1,r,j}^{(T)} + \frac{(3+c)\epsilon}{3} \\
\leq &\lambda_{1,r,j_0}^{(T)} - \frac{2(1+c)}{k} \sum_{j \in J_+} \frac{3}{4} \lambda_{1,r,j_0}^{(T)} + \frac{(3+c)\epsilon}{3} \\
\leq &\lambda_{1,r,j_0}^{(T)} \left( 1 - \frac{3(1+c)|J_+|}{2k} \right) + \frac{(3+c)\epsilon}{3} \\
\leq &-\frac{1}{2} \lambda_{1,r,j_0}^{(T)} + \frac{(3+c)\epsilon}{3} < 0.
\end{aligned}
$$

The first equation expands $\boldsymbol{x}$ and $\boldsymbol{\rho}$ and uses the conclusion in Lemma E.10; the second inequality uses the conclusion in Lemma E.11 and Lemma E.26; the third inequality rearranges the terms and uses the conclusion in Corollary E.5; the fourth inequality uses conclusion in Theorem 4.5.

Reusing the logic of the above inequality, we have

$$
\langle \boldsymbol{w}_{-1,r}^{(T)}, \boldsymbol{x} + \boldsymbol{\rho} \rangle + b_{-1,r}^{(T)}
$$

$$
\geq \langle \boldsymbol{w}_{-1,r}^{(T)}, \boldsymbol{\mu}_{j_0} \rangle + \langle \boldsymbol{w}_{-1,r}^{(T)}, \boldsymbol{\xi} \rangle - \frac{2(1+c)}{k} \sum_{j \in J_+} \langle \boldsymbol{w}_{-1,r}^{(T)}, \boldsymbol{\mu}_j \rangle + \frac{2(1+c)}{k} \sum_{j \in J_-} \langle \boldsymbol{w}_{-1,r}^{(T)}, \boldsymbol{\mu}_j \rangle - \frac{\epsilon}{3}
$$

$$
\geq \lambda_{-1,r,j_0}^{(T)} - \frac{\epsilon}{6} - \frac{\epsilon}{6} - \frac{2(1+c)}{k} \sum_{j \in J_+} \left( \lambda_{-1,r,j}^{(T)} + \frac{\epsilon}{6} \right) + \frac{2(1+c)}{k} \sum_{j \in J_-} \left( \lambda_{-1,r,j}^{(T)} - \frac{\epsilon}{6} \right) - \frac{\epsilon}{3}
$$

$$
\geq \lambda_{-1,r,j_0}^{(T)} + \frac{2(1+c)}{k} \sum_{j \in J_-} \lambda_{-1,r,j}^{(T)} - \frac{(3+c)\epsilon}{3}
$$

$$
\geq \frac{2(1+c)}{k} \sum_{j \in J_-} \lambda_{-1,r,j}^{(T)} - \frac{(6+c)\epsilon}{3}
$$

$$
\geq \frac{2(1+c)|J_-|}{k} - \frac{(6+c)\epsilon}{3} \geq 0.
$$

By combining the two inequalities above, we can obtain that

$$
f_{\boldsymbol{\theta}^{(T)}}(\boldsymbol{x}+\boldsymbol{\rho}) = \frac{1}{m} \sum_{r \in [m]} \mathrm{ReLU}\left( \langle \boldsymbol{w}_{1,r}^{(T)}, \boldsymbol{x} + \boldsymbol{\rho} \rangle + b_{1,r}^{(T)} \right) - \frac{1}{m} \sum_{r \in [m]} \mathrm{ReLU}\left( \langle \boldsymbol{w}_{-1,r}^{(T)}, \boldsymbol{x} + \boldsymbol{\rho} \rangle + b_{-1,r}^{(T)} \right) < 0.
$$

This is to say $\mathrm{sgn}(f_{\boldsymbol{\theta}^{(T)}}(\boldsymbol{x}+\boldsymbol{\rho})) \neq \mathrm{sgn}(f_{\boldsymbol{\theta}^{(T)}}(\boldsymbol{x}))$, which means $\mathrm{Acc}_{\mathrm{robust}}^{\mathcal{D}}(f_{\boldsymbol{\theta}^{(T)}}; 2(1+c)\sqrt{d/k}) = o(1)$. □

### E.3 Proof of Theorem 4.6

**Theorem E.27** (Restatement of Theorem 4.6). *In the setting of Theorem 4.5,*

$$
\inf_{C>0} \sup_{\boldsymbol{x} \in \mathbb{R}^d : \|\boldsymbol{x}\|_2 = \sqrt{d}} |C f_{\mathrm{FA}}(\boldsymbol{x}) - f_{\boldsymbol{\theta}^{(T)}}(\boldsymbol{x})| = o(1),
$$

*where $f_{\mathrm{FA}}(\boldsymbol{x})$ is the feature-averaging network (Definition 4.1).*

*Proof of Theorem E.27.* By Lemma E.25, we have

$$
\left| \mathrm{ReLU}\left( \langle \boldsymbol{w}_{s,r}^{(T)}, \boldsymbol{x} \rangle \right) - \mathrm{ReLU}\left( \left\langle \lambda^{(T)} \sum_{j \in J_s} \boldsymbol{\mu}_j \|\boldsymbol{\mu}_j\|^{-2}, \boldsymbol{x} \right\rangle \right) \right|
$$

$$
\leq \left| \langle \boldsymbol{w}_{s,r}^{(T)}, \boldsymbol{x} \rangle - \langle \lambda^{(T)} \sum_{j \in J_s} \boldsymbol{\mu}_j \|\boldsymbol{\mu}_j\|^{-2}, \boldsymbol{x} \rangle \right|
$$

$$
\leq \|\boldsymbol{x}\| \left\| \boldsymbol{w}_{s,r}^{(T)} - \lambda^{(T)} \sum_{j \in J_s} \boldsymbol{\mu}_j \|\boldsymbol{\mu}_j\|^{-2} \right\| = o(1).
$$

Thus we have

$$
\left| \frac{1}{m} \sum_{r \in [m]} \mathrm{ReLU}\left( \langle \boldsymbol{w}_{1,r}^{(T)}, \boldsymbol{x} \rangle + b_{1,r}^{(T)} \right) - \mathrm{ReLU}\left( \left\langle \lambda^{(T)} \sum_{j \in J_+} \boldsymbol{\mu}_j \|\boldsymbol{\mu}_j\|^{-2}, \boldsymbol{x} \right\rangle \right) \right|
$$

$$
= \left| \frac{1}{m} \sum_{r \in [m]} \mathrm{ReLU}\left( \langle \boldsymbol{w}_{1,r}^{(T)}, \boldsymbol{x} \rangle \right) - \frac{1}{m} \sum_{r \in [m]} \mathrm{ReLU}\left( \left\langle \lambda^{(T)} \sum_{j \in J_+} \boldsymbol{\mu}_j \|\boldsymbol{\mu}_j\|^{-2}, \boldsymbol{x} \right\rangle \right) \right| + \frac{1}{m} \sum_{r \in [m]} |b_{1,r}^{(T)}|
$$

$$
\leq \frac{1}{m} \sum_{r \in [m]} \left| \langle \boldsymbol{w}_{1,r}^{(T)}, \boldsymbol{x} \rangle - \langle \lambda^{(T)} \sum_{j \in J_+} \boldsymbol{\mu}_j \|\boldsymbol{\mu}_j\|^{-2}, \boldsymbol{x} \rangle \right| + \epsilon
$$

$$
\leq \frac{1}{m} \sum_{r \in [m]} \|\boldsymbol{x}\| \left\| \boldsymbol{w}_{1,r}^{(T)} - \lambda^{(T)} \sum_{j \in J_+} \boldsymbol{\mu}_j \|\boldsymbol{\mu}_j\|^{-2} \right\| + \epsilon = o(1).
$$

Similarly, we have

$$
\left| \frac{1}{m} \sum_{r \in [m]} \mathrm{ReLU}\left( \langle \boldsymbol{w}_{-1,r}^{(T)}, \boldsymbol{x} \rangle + b_{-1,r}^{(T)} \right) - \mathrm{ReLU}\left( \left\langle \lambda^{(T)} \sum_{j \in J_-} \boldsymbol{\mu}_j \|\boldsymbol{\mu}_j\|^{-2}, \boldsymbol{x} \right\rangle \right) \right| = o(1)
$$

Combining these two inequalities together, we have

$$
\sup_{\boldsymbol{x} \in \mathbb{R}^d : \|\boldsymbol{x}\|_2 = \sqrt{d}} \left| \frac{\lambda^{(T)}}{d} f_{\mathrm{FA}}(\boldsymbol{x}) - f_{\boldsymbol{\theta}^{(T)}}(\boldsymbol{x}) \right| = o(1).
$$

$\square$

# F    PROOF FOR SECTION 4: FEATURE-DECOUPLING REGIME

First, we recall the fine-Grained supervision, multi-Class network classifier and training algorithm.

**Fine-Grained Supervision.**    Following the setting in Section 3, we consider the binary classification task with data distribution $\mathcal{D}(\{\boldsymbol{\mu}_j\}_{j=1}^k, J_\pm)$. But instead of training the model directly to predict the binary labels, we assume that we are able to label each data point with the cluster $\hat{y} \in [k]$ it belongs to, and then we train a $k$-class classifier to predict the cluster labels. More specifically, we first sample a training set $\mathcal{S} := \{(\boldsymbol{x}_i, y_i)\}_{i=1}^n \subseteq \mathbb{R}^d \times \{\pm 1\}$ from $\mathcal{D}$, along with the cluster labels $\{\tilde{y}_i\}_{i=1}^n$ for all data points. Then a $k$-class neural network classifier is trained on $\tilde{\mathcal{S}} := \{(\boldsymbol{x}_i, \tilde{y}_i)\}_{i=1}^n \subseteq \mathbb{R}^d \times [k]$.

**Multi-Class Network Classifier.** We train the following two-layer neural network for the $k$-class classification mentioned above: $\boldsymbol{F_\theta}(\boldsymbol{x}) := (f_1(\boldsymbol{x}), f_2(\boldsymbol{x}), \dots, f_k(\boldsymbol{x})) \in \mathbb{R}^k$, where $f_j(\boldsymbol{x}) := \frac{1}{m} \sum_{r=1}^h \mathrm{ReLU}(\langle \boldsymbol{w}_{j,r}, \boldsymbol{x} \rangle)$, and $\boldsymbol{\theta} := (\boldsymbol{w}_{1,1}, \boldsymbol{w}_{1,2}, \dots, \boldsymbol{w}_{k,h}) \in \mathbb{R}^{khd}$ are trainable weights, and $h = \Theta(1)$ is the width of each sub-network. The outputs $\boldsymbol{F_\theta}(\boldsymbol{x})$ are then converted to probabilities using the softmax function, namely $p_j(\boldsymbol{x}) := \frac{\exp(f_j(\boldsymbol{x}))}{\sum_{i=1}^k \exp(f_i(\boldsymbol{x}))}$ for $j \in [k]$. For predicting the binary label for the original binary classification task on $\mathcal{D}$, we take the difference of the probabilities of the positive and negative classes, i.e., $F_{\boldsymbol{\theta}}^{\mathrm{binary}}(\boldsymbol{x}) := \sum_{j \in J_+} p_j(\boldsymbol{x}) - \sum_{j \in J_-} p_j(\boldsymbol{x})$. The clean accuracy $\mathrm{Acc}_{\mathrm{clean}}^{\mathcal{D}}(F_{\boldsymbol{\theta}}^{\mathrm{binary}})$ and $\delta$-robust accuracy $\mathrm{Acc}_{\mathrm{robust}}^{\mathcal{D}}(F_{\boldsymbol{\theta}}^{\mathrm{binary}}; \delta)$ are then defined similarly as before.

**Training Objective and Gradient Descent.** We train the multi-class network $\boldsymbol{F_\theta}(\boldsymbol{x})$ to minimize the cross-entropy loss $\mathcal{L}_{\mathrm{CE}}(\boldsymbol{\theta}) := -\frac{1}{n} \sum_{i=1}^n \log p_{\tilde{y}_i}(\boldsymbol{x}_i)$. Similar to Section 3, we use gradient descent to minimize the loss function $\mathcal{L}_{\mathrm{CE}}(\boldsymbol{\theta})$ with learning rate $\eta$, i.e., $\boldsymbol{\theta}^{(t+1)} = \boldsymbol{\theta}^{(t)} - \eta \nabla_{\boldsymbol{\theta}} \mathcal{L}_{CE}(\boldsymbol{F}_{\boldsymbol{\theta}^{(t)}})$. At initialization, we set $\boldsymbol{w}_{j,r}^{(0)} \sim \mathcal{N}(0, \sigma_{\mathrm{w}}^2 \boldsymbol{I}_d)$ for some $\sigma_{\mathrm{w}} > 0$.

Denote $\ell_{i,j}^{\prime(t)} := \nabla_{f_j(\boldsymbol{x}_i)} \mathcal{L}_{CE}(\boldsymbol{F}^{(t)}) = -\mathbb{1}(\boldsymbol{x}_i \in I_j) + \dfrac{\exp(f_j^{(t)}(\boldsymbol{x}_i))}{\displaystyle\sum_{p \in J} \exp(f_p^{(t)}(\boldsymbol{x}_i))}.$

Since many of the proofs in this section are very similar to those in Appendix E, we reuse the logic of the proofs and present the key steps.

We also assume that the properties of the training dataset(Proposition D.6) in Appendix D hold.

## F.1    PROPOSITIONS OF NETWORK INITIALIZATION

**Proposition F.1.** *With probability at least $1 - 4hmd^{-\ln(d)} - 2h^{-3}m^{-3}$ , we have the following properties for our network initialization:*

- *For any $r \in [m]$, we have $\sigma_w \left( \sqrt{d} - 2\ln(d) \right) \leq \|\boldsymbol{w}_{s,r}^{(0)}\| \leq \sigma_w \left( \sqrt{d} + 2\ln(d) \right)$.*

The proof of Proposition F.1 is the same as the proof of D.10.

**Definition F.2** (Activation Region over Data Input). *Let $T_{s,r,j} := \{i \in I_j : \langle \boldsymbol{w}_{s,r}^{(0)}, \boldsymbol{x}_i \rangle + b_{s,r}^{(0)} \geq 0\}$ be the set of indices of training data points in the $j$-th cluster which can activate the neuron with weight $\boldsymbol{w}_{s,r}$ at time step 0.*

Then, we give the following result about the activation region $T_{s,r,j}$.

**Proposition F.3.** *Assuming Proposition D.6 and Proposition D.10 holds. Then with probability at least $1 - (hk)^{-0.01} - 2hk^2 \exp\left(-\frac{n}{9k^3h^2}\right)$, for all $r \in [h], s, j \in J$, we have*

$$|T_{s,r,j}| \geq \frac{n}{3k^3h^2}.$$

The proof of this lemma is the same as the proof of Proposition D.12.

**Lemma F.4.** *Assuming Proposition F.1 holds, for all $i \in I, s \in J$, we have*

$$-\mathbb{1}(\boldsymbol{x}_i \in I_s) + \frac{1}{2k} \leq \ell_{i,s}^{\prime(0)} \leq -\mathbb{1}(\boldsymbol{x}_i \in I_s) + \frac{2}{k}.$$

*Proof of Lemma F.4.* By Proposition F.1 and Property 2 in D.6, for every $s \in J$, we have

$$\left| f_s^{(0)}(\boldsymbol{x}_i) \right| \leq \frac{1}{h} \sum_{r \in [h]} \left| \langle \boldsymbol{w}_{s,r}^{(0)}, \boldsymbol{x}_i \rangle \right| \leq \frac{1}{h} \sum_{r \in [h]} |\boldsymbol{w}_{s,r}^{(0)}| \, |\boldsymbol{x}_i| \leq 4\sigma_w d \leq \ln(2).$$

Thus $1 \leq \exp(f_s(\boldsymbol{x}_i)) \leq 2$. Then we have

$$\frac{1}{2k} \leq \frac{\exp(f_s^{(0)}(\boldsymbol{x}_i))}{\sum\limits_{p \in J} \exp(f_p^{(0)}(\boldsymbol{x}_i))} \leq \frac{2}{k}.$$

$$-\mathbb{1}(\boldsymbol{x}_i \in I_s) + \frac{1}{2k} \leq \ell_{i,s}^{\prime(0)} = -\mathbb{1}(\boldsymbol{x}_i \in I_s) + \frac{\exp(f_s^{(0)}(\boldsymbol{x}_i))}{\sum\limits_{p \in J} \exp(f_p^{(0)}(\boldsymbol{x}_i))} \leq -\mathbb{1}(\boldsymbol{x}_i \in I_s) + \frac{2}{k}.$$

$\square$

### F.2    ANALYSIS OF TRAINING DYNAMICS

Denote $S_{i,s}^{(t)} := \{r \in [h] : \langle \boldsymbol{w}_{s,r}^{(t)}, \boldsymbol{x}_i \rangle \geq 0\}$ for $i \in I, s \in J$.

**Lemma F.5.** *For every* $i \in I$, *we have* $S_{i,c(i)}^{(1)} = [h]$.

*Proof of Lemma F.5.* This proof is similar to the proof of Lemma E.1.

For every $i \in I, r \in [h]$, we have

$$\langle \boldsymbol{w}_{c(i),r}^{(1)}, \boldsymbol{x}_i \rangle$$
$$= \langle \boldsymbol{w}_{c(i),r}^{(0)}, \boldsymbol{x}_i \rangle - \eta \langle \nabla_{\boldsymbol{w}_{c(i),r}} \mathcal{L}(\boldsymbol{\theta}^{(0)}), \boldsymbol{x}_i \rangle$$

We examine the update term $\langle \nabla_{\boldsymbol{w}_{c(i),r}} \mathcal{L}(\boldsymbol{\theta}^{(0)}), \boldsymbol{x}_i \rangle$.

$$- hn \langle \nabla_{\boldsymbol{w}_{c(i),r}} \mathcal{L}(\boldsymbol{\theta}^{(0)}), \boldsymbol{x}_i \rangle$$
$$= - hn \langle \sum_{j \in I_{c(i)}} \ell_{j,c(i)}^{\prime(0)} \mathbb{1}\left( \langle \boldsymbol{w}_{c(i),r}^{(0)}, \boldsymbol{x}_j \rangle \geq 0 \right) \boldsymbol{x}_j + \sum_{j \notin I_{c(i)}} \ell_{j,c(i)}^{\prime(0)} \mathbb{1}\left( \langle \boldsymbol{w}_{c(i),r}^{(0)}, \boldsymbol{x}_j \rangle \geq 0 \right) \langle \boldsymbol{x}_j, \boldsymbol{x}_i \rangle$$
$$\geq - hn \sum_{j \in I_{c(i)}} \ell_{j,c(i)}^{\prime(0)} \mathbb{1}\left( \langle \boldsymbol{w}_{c(i),r}^{(0)}, \boldsymbol{x}_j \rangle \geq 0 \right) \langle \boldsymbol{x}_j, \boldsymbol{x}_i \rangle - \sum_{j \notin I_{c(i)}} \ell_{j,c(i)}^{\prime(0)} \mathbb{1}\left( \langle \boldsymbol{w}_{c(i),r}^{(0)}, \boldsymbol{x}_j \rangle \geq 0 \right) |\langle \boldsymbol{x}_j, \boldsymbol{x_i} \rangle|$$

By Lemma F.4 and Lemma F.3, we know that

$$- \sum_{j \in I_{c(i)}} \ell_{j,c(i)}^{\prime(0)} \mathbb{1}\left( \langle \boldsymbol{w}_{c(i),r}^{(0)}, \boldsymbol{x}_j \rangle \geq 0 \right) \langle \boldsymbol{x}_j, \boldsymbol{x}_i \rangle \geq (1 - \frac{2}{k}) \frac{d}{2} |T_{c(i),r,c(i)}| \geq \frac{dn}{12 k^3 h^2}.$$

$$- \sum_{j \notin I_{c(i)}} \ell_{j,c(i)}^{\prime(0)} \mathbb{1}\left( \langle \boldsymbol{w}_{c(i),r}^{(0)}, \boldsymbol{x}_j \rangle \geq 0 \right) |\langle \boldsymbol{x}_j, \boldsymbol{x_i} \rangle| \geq -\frac{2n\Delta}{k}.$$

Combining the two inequalities above, we have

$$\langle \boldsymbol{w}_{c(i),r}^{(1)}, \boldsymbol{x}_i \rangle \geq \langle \boldsymbol{w}_{c(i),r}^{(0)}, \boldsymbol{x}_i \rangle + \frac{\eta}{h} \left( \frac{d}{12 k^3 h^2} - \frac{2\Delta}{k} \right)$$
$$\geq -2\sigma_w d + \frac{\eta \Delta}{h} \geq 0.$$

Therefore, we have $S_{i,c(i)}^{(1)} = [h]$ for every $i \in I$.

$\square$

Recall the definition of weight decomposition we will use in multi-classification tasks.

**Lemma F.6** (Weight Decomposition). *During the training dynamics, there exists the following coefficient sequences $\lambda_{s,r,j}^{(t)}$ and $\sigma_{s,r,i}^{(t)}$ for each neuron $s, j \in J, r \in [h]$ such that*

$$\boldsymbol{w}_{s,r}^{(t)} = \boldsymbol{w}_{s,r}^{(0)} + \sum_{j \in J} \lambda_{s,r,j}^{(t)} \boldsymbol{\mu}_j \|\boldsymbol{\mu}_j\|^{-2} + \sum_{i \in I} \sigma_{s,r,i}^{(t)} \boldsymbol{\xi}_i \|\boldsymbol{\xi}_i\|^{-2}.$$

**Corollary F.7.** *The coefficient sequences $\lambda_{r,j}^{(t)}$ and $\sigma_{i,j}^{(t)}$ for each pair $i \in I, r \in J, j \in J$ defined in Lemma F.6 satisfy:*

$$\lambda_{s,r,j}^{(t)} \|\boldsymbol{\mu}_j\|^{-2} = \sum_{i \in I_j} \sigma_{s.r,i}^{(t)} \|\boldsymbol{\xi}_i\|^{-2}.$$

**Corollary F.8.** *For all $i \in I, r, j \in J$, we have the following update equation for $\lambda_{s,r,j}$ and $\sigma_{s,r,i}$.*

$$\lambda_{s,r,j}^{(t+1)} = \lambda_{s,r,j}^{(t)} - \frac{\eta}{nh} \sum_{p \in I_j} \ell_{p,s}'^{(t)} \|\boldsymbol{\mu}_j\|^2 \mathbb{1}\left(\langle \boldsymbol{w}_{s,r}^{(t)}, \boldsymbol{x}_p \rangle \geq 0\right),$$

$$\sigma_{s,r,i}^{(t+1)} = \sigma_{s,r,i}^{(t)} - \frac{\eta}{nh} \ell_{i,s}'^{(t)} \|\boldsymbol{\xi}_i\|^2 \mathbb{1}\left(\langle \boldsymbol{w}_{s,r}^{(t)}, \boldsymbol{x}_i \rangle \geq 0\right),$$

$$\lambda_{s,r,j}^{(0)} = 0, \sigma_{s,r,i}^{(0)} = 0.$$

**Corollary F.9.** *The coefficient sequences $\lambda_{s,r,j}^{(t)}$ and $\sigma_{s,r,i}^{(t)}$ for each pair $s, j \in J, i \in I, r \in [h]$ defined in Lemma F.6 satisfy:*

$$\lambda_{s,r,j}^{(t)} \geq 0 \ \text{ iff } s = j,$$

$$\sigma_{s,r,i}^{(t)} \geq 0 \ \text{ iff } s = c(i).$$

Then we reuse the logic of the proof of Lemma E.7 to prove the main result in our multi-classification setting.

Denote $q_i^{(t)} = f_{c(i)}^{(t)}(\boldsymbol{x}_i), \hat{q}_i^{(t)} = \frac{1}{h} \sum_{r \in [h]} \left(\lambda_{c(i),r,c(i)}^{(t)} + \sigma_{c(i),r,i}^{(t)}\right).$

$\Delta_q^{(t)}(i, j) = q_i^{(t)} - q_j^{(t)},$

$\hat{\Delta}_q^{(t)}(i, j) = \hat{q}_i^{(t)} - \hat{q}_j^{(t)} = \frac{1}{h} \sum_{r \in [h]} \left(\lambda_{c(i),r,c(i)}^{(t)} - \lambda_{c(j),r,c(j)}^{(t)} + \sigma_{c(i),r,i}^{(t)} - \sigma_{c(j),r,j}^{(t)}\right).$

Denote $\epsilon = \max\left\{\frac{2\ln(\frac{n}{k})}{\sqrt{\frac{n}{k}} - \ln(\frac{n}{k})}, \frac{k^2\Delta}{d}, \frac{k^2}{n}\right\}$. We know that $\epsilon = o(k^{-2.5})$ according to our hyper-parameter Assumption D.4.

**Lemma F.10.** *For $t \leq T_0 = \exp(\tilde{O}(k^{0.5})), i, j \in I, r \in [h]$, we have*

1. $\frac{k}{2n} \ln(t\eta) \leq \sigma_{c(i),r,i} \leq \frac{2k}{n} \ln(t+1)$

2. $\hat{\Delta}_q^{(t)}(i, j) \leq 5k\epsilon$ when $c(i) = c(j)$,

3. $\hat{\Delta}_q^{(t)}(i, j) \leq 32k^2\epsilon,$

4. $\Delta_q^{(t)}(i, j) \leq 33k^2\epsilon,$

5. $\ell_{i,c(i)}'^{(t)}/\ell_{j,c(j)}'^{(t)} \leq 1 + 14k\epsilon$ when $c(i) = c(j),$

6. $\ell_{i,c(i)}'^{(t)}/\ell_{j,c(j)}'^{(t)} \leq 1 + 67k^2\epsilon,$

7. $S_{i,c(i)}^{(1)} = [h],$

8. $\left|\lambda^{(t)}_{s,r,c(i)}\right| \leq \epsilon, \left|\sigma^{(t)}_{s,r,c(i)}\right| \leq 2\epsilon$ *for* $s \in J, s \neq c(i)$.

*Proof of Lemma F.10.* Since the proof of this lemma follows exactly the same logic as Lemma E.7, we omit some details and only outlined the necessary lemmas and the key steps of the proof.

First, the base case of the induction is simple, so we only consider the inductive step.

**Lemma F.11.** *Assuming the inductive hypotheses hold before time step* $t$, *for all* $r \in [h], s, j \in J$, *we have*

$$\left|\langle \boldsymbol{w}^{(t)}_{s,r}, \boldsymbol{\mu}_j \rangle - \lambda^{(t)}_{s,r,j}\right| \leq \frac{\epsilon}{6}.$$

**Lemma F.12.** *Assuming the inductive hypotheses hold before time step* $t$, *for all* $r \in [h], s, j \in J$, *we have*

$$\left|\langle \boldsymbol{w}^{(t)}_{s,r}, \boldsymbol{\xi}_i \rangle - \sigma^{(t)}_{s,r,i}\right| \leq \frac{\epsilon}{6}.$$

**Lemma F.13.** *Assuming the inductive hypotheses hold before time step* $t$, *for all* $r \in [h], s, j \in J$, *we have*

$$\left|\langle \boldsymbol{w}^{(t)}_{s,r}, \boldsymbol{x}_i \rangle - \lambda^{(t)}_{s,r,c(i)} - \sigma^{(t)}_{s,r,i}\right| \leq \frac{\epsilon}{3}.$$

**Lemma F.14.** *Assuming the inductive hypotheses hold before time step* $t$, *for all* $i \in I$, *we have*

$$|q^{(t)}_i - \hat{q}^{(t)}_i| \leq \epsilon$$

The proofs of these three lemmas are identical to the proofs of Lemma E.11, Lemma E.12, Lemma E.13 and Lemma E.14 in Appendix Appendix E, except that in the previous proof, there was an additional subscript $s$ used to indicate 2-classification label, whereas here it is used to represent a fine-grained $k$-classification label.

**Corollary F.15.** *Assuming the inductive hypotheses hold before time step* $t$, *for all* $i, j \in I$, *we have*

$$\left|\Delta^{(t)}_q(i,j) - \hat{\Delta}^{(t)}_q(i,j)\right| \leq 2\epsilon.$$

Next, we present the key steps of the auto-balance process for $\hat{\Delta}_q(i,j)$.

**Lemma F.16** (Property 8). *Assuming the inductive hypotheses hold before time step* $t$, *for* $i \in I, s \in J, s \neq c(i), r \in [m]$, *we have* $|\lambda^{(t)}_{s,r,c(i)}| \leq \epsilon, |\sigma^{(t)}_{s,r,i}| \leq 2\epsilon$.

*Proof of Lemma F.16.* We distinguish between two scenarios.

Case(I): For all $i \in I_j$, $\langle \boldsymbol{w}^{(t)}_{s,r}, \boldsymbol{x}_i \rangle < 0$.

Then by Corollary F.8 and inductive hypothesis, we know that

$$|\lambda^{(t+1)}_{s,r,j}| = |\lambda^{(t)}_{s,r,j}| \leq \epsilon.$$

Case(II): There exists $i \in I_j$ such that $\langle \boldsymbol{w}^{(t)}_{s,r}, \boldsymbol{x}_i \rangle \geq 0$.

By Lemma F.13, we know that

$$\langle \boldsymbol{w}^{(t)}_{s,r}, \boldsymbol{x}_i \rangle - \lambda^{(t)}_{s,r,j} - \sigma^{(t)}_{s,r,i} \leq \frac{\epsilon}{3}.$$

Then noting that $\langle \boldsymbol{w}^{(t)}_{s,r}, \boldsymbol{x}_i \rangle \geq 0$ and $\sigma^{(t)}_{s,r,i} \leq 0$, we have

$$\lambda^{(t)}_{s,r,j} \geq \langle \boldsymbol{w}^{(t)}_{s,r}, \boldsymbol{x}_i \rangle - \sigma^{(t)}_{s,r,i} - \frac{\epsilon}{3} \geq -\frac{\epsilon}{3}.$$

Then using the conclusion in Corollary F.8, we know that

$$|\lambda_{s,r,j}^{(t+1)} - \lambda_{s,r,j}^{(t)}| \le \frac{\eta d}{nm} \sum_{i \in I_j} -\ell_{i,s}'^{(t)} \le \frac{\eta d}{m} \le \frac{\epsilon}{3}.$$

Thus we have $\lambda_{r,j}^{(t+1)} \ge -\epsilon$. Noting that $\lambda_{s,r,j}^{(t+1)} \le 0$, we have $|\lambda_{s,r,j}^{(t+1)}| \le \epsilon$. Then by Corollary F.7, we know that $|\sigma_{s,r,i}^{(t)}| \le 2|\lambda_{s,r,j}^{(t+1)}| \le 2\epsilon$. ∎

**Lemma F.17.** *Assuming the inductive hypotheses hold before time step $t$, for all $s \in J, i \in I, s \ne c(i)$, we have*

$$f_s(\boldsymbol{x_i}) \le \epsilon.$$

*Proof.* By Lemma F.13 and Corollary F.9, we know that

$$\begin{aligned}
f_s(\boldsymbol{x_i}) &= \frac{1}{h} \sum_{r \in [h]} \mathrm{ReLU}(\langle \boldsymbol{w}_{s,r}^{(t)}, \boldsymbol{x_i} \rangle) \\
&\le \frac{1}{h} \sum_{r \in [h]} |\langle \boldsymbol{w}_{s,r}^{(t)}, \boldsymbol{x_i} \rangle| \\
&\le \frac{1}{h} \sum_{r \in [h]} (\frac{\epsilon}{3} + \lambda_{s,r,c(i)}^{(t)} + \sigma_{s,r,i}^{(t)}) \\
&\le \frac{1}{h} \sum_{r \in [h]} \frac{\epsilon}{3} \\
&\le \epsilon.
\end{aligned}$$

∎

**Lemma F.18.** *Assuming the inductive hypotheses hold before time step $t$, for any two training data points $\boldsymbol{x}_i, \boldsymbol{x}_j$, if $q_i \ge q_j$, we have*

$$\frac{1}{1+2\epsilon} \frac{e^{\Delta_q^{(t)}(i,j)} + k - 1}{k} \le \frac{\ell_{j,c(j)}'^{(t)}}{\ell_{i,c(i)}'^{(t)}} \le e^{\Delta_q^{(t)}(i,j)}(1+2\epsilon).$$

*Proof of Lemma F.18.* By Lemma F.17 and noting that $\exp(\epsilon) \le 1 + 2\epsilon$, we know that

$$\frac{k-1}{k-1+\exp(q_i^{(t)})} \le -\ell_{i,c(i)}'^{(t)} = \frac{\sum\limits_{p \ne c(i)} \exp(f_p^{(t)}(\boldsymbol{x}_i))}{\sum\limits_{p \in J} \exp(f_p^{(t)}(\boldsymbol{x}_i))} \le \frac{(k-1)\exp(\epsilon)}{(k-1)\exp(\epsilon) + \exp(q_i^{(t)})} \le \frac{(k-1)(1+2\epsilon)}{k-1+\exp(q_i^{(t)})}$$

Thus we know that

$$\frac{1}{(1+2\epsilon)} \frac{k-1+\exp(q_i^{(t)})}{k-1+\exp(q_j^{(t)})} \le \frac{\ell_{j,c(j)}'^{(t)}}{\ell_{i,c(i)}'^{(t)}} \le (1+2\epsilon) \frac{k-1+\exp(q_i^{(t)})}{k-1+\exp(q_j^{(t)})}$$

$$\begin{aligned}
\frac{k-1+\exp(q_i^{(t)})}{k-1+\exp(q_j^{(t)})} &= 1 + \frac{\exp(q_i^{(t)}) - \exp(q_j^{(t)})}{k-1+\exp(q_j^{(t)})} \\
&= 1 + \frac{\exp(q_j^{(t)})}{k-1+\exp(q_j^{(t)})} \left( \exp\left( \Delta_q^{(t)}(i,j) \right) - 1 \right)
\end{aligned}$$

For the second term,

$$\frac{\exp\left(\Delta_q^{(t)}(i,j)\right)-1}{k} \leq \frac{\exp(q_j^{(t)})}{k-1+\exp(q_j^{(t)})}\left(\exp\left(\Delta_q^{(t)}(i,j)\right)-1\right) \leq \exp\left(\Delta_q^{(t)}(i,j)\right)-1.$$

Thus we know

$$\frac{1}{1+2\epsilon}\frac{e^{\Delta_q^{(t)}(i,j)}+k-1}{k} \leq \frac{\ell_{j,c(j)}'^{(t)}}{\ell_{i,c(i)}'^{(t)}} \leq e^{\Delta_q^{(t)}(i,j)}(1+2\epsilon).$$

∎

We first consider the case when $c(i) = c(j)$. We distinguish between two scenarios, one is when $\left|\hat{\Delta}_q^{(t)}(i,j)\right|$ is relatively small and the other is when $\left|\hat{\Delta}_q^{(t)}(i,j)\right|$ is relatively large.

Case(I): $\hat{\Delta}_q^{(t)}(i,j) \leq 4k\epsilon$.

In this case, we have $\hat{\Delta}_q^{(t+1)}(i,j) \leq 5k\epsilon$ due to small learning rate $\eta$.

Case(II): $\hat{\Delta}_q^{(t)}(i,j) \geq 4k\epsilon$.

By Lemma F.14, we know that

$$\Delta_q^{(t)}(i,j) \geq \hat{\Delta}_q^{(t)}(i,j) - 2\epsilon \geq 3k\epsilon.$$

By Lemma F.18, we know that

$$\frac{\ell_{j,c(j)}'^{(t)}}{\ell_{i,c(i)}'^{(t)}} \geq \frac{1}{1+2\epsilon}\frac{e^{\Delta_q^{(t)}(i,j)}+k-1}{k} \geq \frac{1}{1+2\epsilon}(1+\Delta_q^{(t)}(i,j)/k) \geq 1+\epsilon/2.$$

Noting that $c(i) = c(j)$, we know that

$$\hat{\Delta}_q^{(t+1)}(i,j) - \hat{\Delta}_q^{(t)}(i,j)$$
$$= -\frac{\eta}{nh}\left(\ell_{i,c(i)}'^{(t)}\|\boldsymbol{\xi}_i\|^2 - \ell_{j,c(j)}'^{(t)}\|\boldsymbol{\xi}_j\|^2\right)$$
$$\leq 0$$

Then due to the inductive hypothesis,

$$\hat{\Delta}_q^{(t+1)}(i,j) \leq \hat{\Delta}_q^{(t)}(i,j) \leq 5k\epsilon.$$

By Corollary F.15, we can get Property 4

$$\Delta_q^{(t)}(i,j) \leq \Delta_q^{(t)}(i,j) + 2\epsilon \leq 6k\epsilon.$$

By Lemma F.18 and noting that $e^x \leq 1+2x$ for small $x$ we know that

$$\frac{\ell_{j,c(j)}'^{(t+1)}}{\ell_{i,c(i)}'^{(t+1)}} \leq (1+2\epsilon)e^{\Delta_q^{(t+1)}(i,j)} \leq (1+2\epsilon)(1+2\Delta_q^{(t+1)}(i,j)) \leq 1+14k\epsilon.$$

Next, we consider the case when $c(i) \neq c(j)$. We also distinguish between the two scenarios.

Case(I): $\hat{\Delta}_q^{(t)}(i,j) \leq 31k^2\epsilon$. In this case, we have $\hat{\Delta}_q^{(t+1)}(i,j) \leq 32k^2\epsilon$ due to small learning rate $\eta$.

Case(II): $\hat{\Delta}_q^{(t)}(i,j) \geq 31k^2\epsilon$.

By Lemma F.14, we know that

$$\Delta_q^{(t)}(i,j) \geq \hat{\Delta}_q^{(t)}(i,j) - 2\epsilon \geq 30k^2\epsilon.$$

By Lemma F.18, we know that

$$\frac{\ell_{j,c(j)}^{\prime(t)}}{\ell_{i,c(i)}^{\prime(t)}} \geq \frac{1}{1+2\epsilon} \frac{e^{\Delta_q^{(t)}(i,j)} + k - 1}{k} \geq \frac{1}{1+2\epsilon}(1 + \Delta_q^{(t)}(i,j)/k) \geq 1 + 29k\epsilon.$$

Furthermore, due to the inductive hypothesis, for any $p \in I_{c(i)}, q \in I_{c(j)}$, we know that

$$\frac{\ell_{p,c(p)}^{\prime(t)}}{\ell_{q,c(q)}^{\prime(t)}} \leq \frac{(1+14\epsilon)\ell_{i,c(i)}^{\prime(t)}}{\ell_{j,c(j)}^{\prime(t)}/(1+14k\epsilon)} \leq \frac{(1+14k\epsilon)^2}{1+29k\epsilon} \leq \frac{1}{1+\epsilon}.$$

We know that

$$\hat{\Delta}_q^{(t+1)}(i,j) - \hat{\Delta}_q^{(t)}(i,j)$$

$$= -\frac{\eta}{nh}\left(\sum_{p \in I_{c(i)}} \ell_{p,c(i)}^{\prime(t)}\|\boldsymbol{\mu}_{c(i)}\|^2 - \sum_{p \in I_{c(j)}} \ell_{p,c(j)}^{\prime(t)}\|\boldsymbol{\mu}_{c(j)}\|^2 + \ell_{i,c(i)}^{\prime(t)}\|\boldsymbol{\xi}_i\|^2 - \ell_{j,c(j)}^{\prime(t)}\|\boldsymbol{\xi}_j\|^2\right)$$

$$= -\frac{\eta}{nh}\left(\ell_{i,c(i)}^{\prime(t)}\|\boldsymbol{\xi}_i\|^2 - \ell_{j,c(j)}^{\prime(t)}\|\boldsymbol{\xi}_j\|^2\right) - \frac{\eta}{nh}\left(\sum_{p \in I_{c(i)}} \ell_{p,c(i)}^{\prime(t)}\|\boldsymbol{\mu}_{c(i)}\|^2 - \sum_{p \in I_{c(j)}} \ell_{p,c(j)}^{\prime(t)}\|\boldsymbol{\mu}_{c(j)}\|^2\right)$$

$$\leq 0.$$

By inductive hypothesis, we know that

$$\hat{\Delta}_q^{(t+1)}(i,j) \leq \hat{\Delta}_q^{(t)}(i,j) \leq 32k^2\epsilon.$$

Now, we have completed the main part of the proof, the inductive proofs of Properties 2 and 3. Subsequently, Properties 4, 5, and 6 can be directly derived from Lemma F.18.

By Corollary F.15, we can get Property 4

$$\Delta_q^{(t)}(i,j) \leq \Delta_q^{(t)}(i,j) + 2\epsilon \leq 33k^2\epsilon$$

By Lemma F.18 and noting that $e^x \leq 1 + 2x$ for small $x$ we know that

$$\frac{\ell_{j,c(j)}^{\prime(t+1)}}{\ell_{i,c(i)}^{\prime(t+1)}} \leq (1+2\epsilon)e^{\Delta_q^{(t+1)}(i,j)} \leq (1+2\epsilon)(1+2\Delta_q^{(t+1)}(i,j)) \leq 1 + 67k^2\epsilon.$$

**Lemma F.19.** *For all $i_1, i_2 \in I, r_1, r_2 \in [h]$, we have*

$$\left(1 - 200k^2\epsilon\right)\sigma_{c(i_2),r_2,i_2}^{(t)} - \frac{1}{nd} \leq \sigma_{c(i_1),r_1,i_1}^{(t)} \leq \left(1 + 200k^2\epsilon\right)\sigma_{c(i_2),r_2,i_2}^{(t)} + \frac{1}{nd}.$$

**Lemma F.20.** *For every $i \in I, r_0 \in [h]$, we have $\frac{n}{2k}\sigma_{c(i),r_0,i}^{(t)} - 2\epsilon \leq q_i^{(t)} \leq \frac{2n}{k}\sigma_{c(i),r_0,i}^{(t)} + 2\epsilon$.*

The proof of these lemmas are the same as the proof of Lemma E.19 and Lemma E.20.

**Lemma F.21.** *For every $i \in I, r \in [h]$, we have*

$$\frac{1}{2}\exp\left(-\frac{2n}{k}\sigma_{1,r,i}^{(t)}\right) \leq -\ell_{i,c(i)}^{\prime(t)} \leq 2k\exp\left(-\frac{n}{2k}\sigma_{1,r,i}^{(t)}\right).$$

*Proof of Lemma F.21.* By Lemma F.20, we know that

$$-\ell_{i,c(i)}'^{(t)} \geq \frac{k-1}{k-1+\exp\left(q_i^{(t)}\right)}$$

$$\geq \frac{k-1}{k-1+\exp\left(\frac{2n}{k}\sigma_{1,r,i}^{(t)}+2\epsilon\right)}$$

$$\geq \frac{k-1}{k-1+2\exp\left(\frac{2n}{k}\sigma_{1,r,i}^{(t)}\right)}$$

$$\geq \frac{1}{2}\exp\left(-\frac{2n}{k}\sigma_{1,r,i}^{(t)}\right).$$

By Lemma F.17 and Lemma F.20, we have

$$-\ell_{i,c(i)}'^{(t)} \leq \frac{(k-1)\exp(\epsilon)}{(k-1)\exp(\epsilon)+\exp\left(q_i^{(t)}\right)}$$

$$\leq (1+2\epsilon)\frac{k-1}{k-1+\exp\left(\frac{n}{2k}\sigma_{1,r,i}^{(t)}-2\epsilon\right)}$$

$$\leq \frac{2(k-1)}{k-1+\exp\left(\frac{n}{2k}\sigma_{1,r,i}^{(t)}\right)}$$

$$\leq 2k\exp\left(-\frac{n}{2k}\sigma_{1,r,i}^{(t)}\right)$$

■

Next, we prove Property 1.

$$\sigma_{c(i),r,i}^{(t+1)} = \sigma_{c(i),r,i}^{(t)} - \frac{\eta}{nh}\cdot\ell_{i,c(i)}'^{(t)}\|\boldsymbol{\xi_i}\|^2$$

$$\leq \sigma_{c(i),r,i}^{(t)} + \frac{2k\eta d}{nh}\exp(-\frac{n}{2k}\sigma_{c(i),r,i}^{(t)})$$

$$\leq \frac{2k}{n}\ln(t+1) + \frac{2k}{n}\frac{1}{2t}$$

$$\leq \frac{2k}{n}\ln(t+2)$$

$$\sigma_{c(i),r,i}^{(t+1)} = \sigma_{c(i),r,i}^{(t)} - \frac{\eta}{nh}\cdot\ell_{i,c(i)}'^{(t)}\|\boldsymbol{\xi_i}\|^2$$

$$\geq \sigma_{c(i),r,i}^{(t)} + \frac{\eta d}{2nh}\exp(-\frac{k}{2n}\sigma_{c(i),r,i}^{(t)})$$

$$\geq \frac{k}{2n}\ln(t\eta) + \frac{k}{2n}\frac{2}{t}$$

$$\geq \frac{k}{2n}\ln((t+1)\eta)$$

Finally, we prove Property 7.

By Property 6 in the inductive hypotheses, we know that for all $i \in I$,

$$-\frac{1}{1+67k^2\epsilon}\ell_{1,c(1)}'^{(t)} \leq |\ell_{i,c(i)}'^{(t)}| \leq -(1+67k^2\epsilon)\ell_{1,c(1)}'^{(t)}.$$

Then we know that

$$
\begin{aligned}
-\langle \nabla_{\boldsymbol{w}_p} \mathcal{L}(\boldsymbol{\theta}^{(t)}), \boldsymbol{x}_i \rangle &\geq - \sum_{p \in I_{c(i)}} \ell'^{(t)}_{p,c(i)} \langle \boldsymbol{x}_p, \boldsymbol{x}_i \rangle - \sum_{p \notin I_{c(i)}} |\ell'^{(t)}_{p,c(i)}| |\langle \boldsymbol{x}_p, \boldsymbol{x}_i \rangle| \\
&\geq - \sum_{p \in I_{c(i)}} \ell'^{(t)}_{p,c(p)} \langle \boldsymbol{x}_p, \boldsymbol{x}_i \rangle - \sum_{p \notin I_{c(i)}} |\ell'^{(t)}_{p,c(p)}| |\langle \boldsymbol{x}_p, \boldsymbol{x}_i \rangle| \\
&\geq - \ell^{(t)}_{1,c(1)} \left( \sum_{p \in I_{c(i)}} \frac{d}{2(1 + 67k^2\epsilon)} - \sum_{p \notin I_{c(i)}} (1 + 67k^2\epsilon)\Delta \right) \\
&\geq - \ell^{(t)}_{1,c(1)} \left( \frac{dn}{4k(1 + 67k^2\epsilon)} - (1 + 67k^2\epsilon)n\Delta \right) \geq 0.
\end{aligned}
$$

By Property 7 in the inductive hypotheses, we know that $\langle \boldsymbol{w}^{(t+1)}_{c(i)}, \boldsymbol{x}_i \rangle \geq \langle \boldsymbol{w}^{(t)}_{c(i)}, \boldsymbol{x}_i \rangle \geq 0$.

We complete the proof of Lemma F.10.

$\square$

### F.3  PROOF OF THEOREM 4.7

**Theorem F.22** (Restatement of Theorem 4.7). *In the setting of training a multi-class network on the multiple classification problem $\tilde{\mathcal{S}} := \{(\boldsymbol{x}_i, \tilde{y}_i)\}_{i=1}^n \subseteq \mathbb{R}^d \times [k]$ as described in the above, under Assumptions 3.2, 3.3 and 4.3, for some $\gamma = o(1)$, after $\Omega(\eta^{-1}k^8) \leq T \leq \exp(\tilde{O}(k^{1/2}))$ iterations, with probability at least $1 - \gamma$, the neural network satisfies the following properties:*

1. *The clean accuracy is nearly perfect:* $\mathrm{Acc}^{\mathcal{D}}_{\mathrm{clean}}(F^{\mathrm{binary}}_{\boldsymbol{\theta}^{(T)}}) \geq 1 - \exp(-\Omega(\log^2 d))$.

2. *The network converges to the feature-decoupling regime: there exists a time-variant coefficient $\lambda^{(T)} \in [\Omega(\log k), +\infty)$ such that for all $j \in [k]$, $r \in [h]$, the weight vector $\boldsymbol{w}^{(T)}_{j,r}$ can be approximated as*

$$
\left\| \boldsymbol{w}^{(T)}_{j,r} - \lambda^{(T)} \|\boldsymbol{\mu}_j\|^{-2} \boldsymbol{\mu}_j \right\| \leq o(d^{-1/2}).
$$

3. *Consequently, the corresponding binary classifier achieves optimal robustness: for perturbation radius $\delta = O(\sqrt{d})$, the $\delta$-robust accuracy is also nearly perfect, i.e.,* $\mathrm{Acc}^{\mathcal{D}}_{\mathrm{robust}}(F^{\mathrm{binary}}_{\boldsymbol{\theta}^{(T)}}; \delta) \geq 1 - \exp(-\Omega(\log^2 d))$.

*Proof of Theorem Theorem F.22.* We first prove that the network converges to the feature-decoupling regime(Property 2).

**Lemma F.23.** *For all $j \in J, r \in [h]$, we have $\dfrac{\ln(T\eta)}{4} \leq \lambda^{(T)}_{j,r,j} \leq 4\ln(T + 1)$.*

*Proof of Lemma F.23.* Using Property 1 in Lemma F.10 and Corollary F.7, we know that

$$
\begin{aligned}
\lambda^{(T)}_{j,r,j} &= \sum_{p \in I_j} \frac{\|\boldsymbol{\xi}_p\|^2}{\|\boldsymbol{\mu}_j\|^2} \sigma^{(T)}_{j,r,p} \geq \frac{\ln(T\eta)}{4} \\
\lambda^{(T)}_{j,r,j} &= \sum_{p \in I_j} \frac{\|\boldsymbol{\xi}_p\|^2}{\|\boldsymbol{\mu}_j\|^2} \sigma^{(T)}_{j,r,p} \leq 4\ln(T + 1)
\end{aligned}
$$

$\blacksquare$

**Lemma F.24.** *For $r_1, r_2 \in [h], j_1, j_2 \in J$, we have*

$$
\frac{\lambda^{(T)}_{j_1,r_1,j_1}}{\lambda^{(T)}_{j_2,r_2,j_2}} \leq 1 + 204k^2\epsilon.
$$

*Proof.* By Lemma F.19, we know that for any $i_1 \in I_{j_1}, i_2 \in I_{j_2}$

$$\sigma_{j_1,r_1,i_1}^{(t)} \leq \left(1 + 200k^2\epsilon\right) \sigma_{j_2,r_2,i_2}^{(t)} + (nd)^{-1} \leq (1 + 201k^2\epsilon)\sigma_{j_1,r_2,i_2}^{(t)}.$$

$$
\begin{aligned}
\frac{\lambda_{j_1,r_1,j_1}^{(T)}}{\lambda_{j_2,r_2,j_2}^{(T)}} &= \frac{\sum\limits_{p \in I_{j_1}} \|\boldsymbol{\xi}_p\|^2 \sigma_{j_1,r_1,p}^{(t)}}{\sum\limits_{p \in I_{j_2}} \|\boldsymbol{\xi}_p\|^2 \sigma_{j_2,r_2,p}^{(t)}} \\
&\leq (1 + 201k^2\epsilon) \frac{\|\sqrt{d} + \ln(d)\|^2}{\|\sqrt{d} - \ln(d)\|^2} \frac{|I_{j_1}|}{|I_{j_2}|} \\
&\leq (1 + 201k^2\epsilon)(1 + \epsilon)(1 + \epsilon) \\
&\leq 1 + 204k^2\epsilon.
\end{aligned}
$$

∎

We denote $\lambda^{(T)} = \lambda_{1,1,1}^{(T)}$ as the representative of $\{\lambda_{j,r,j} : r \in [m], j \in J\}$.

By Lemma F.23 and Lemma F.24 ,for all $r \in [h], j \in J$, we have

$$|\lambda^{(T)} - \lambda_{j,r,j}^{(T)}| \leq 204k^2\epsilon\lambda^{(T)},$$
$$\lambda^{(T)} \leq 4\ln(T+1),$$
$$\lambda^{(T)} \geq \frac{\ln(T\eta)}{4} \geq 2\ln(k) = \Omega(\log(k)).$$

**Lemma F.25.** *For all $s \in J, r \in [h]$, We have*

$$\sqrt{d} \left\| \boldsymbol{w}_{s,r}^{(T)} - \lambda^{(T)} \boldsymbol{\mu}_s \|\boldsymbol{\mu}_s\|^{-2} \right\| = o(1).$$

*Proof of Lemma E.25.* Recall weight decomposition in Lemma E.2.

$$
\sqrt{d}\left(\boldsymbol{w}_{s,r}^{(T)} - \lambda^{(T)}\boldsymbol{\mu}_s\|\boldsymbol{\mu}_s\|^{-2}\right) = \underbrace{\sqrt{d}\boldsymbol{w}_{s,r}^{(0)}}_{\mathcal{L}_1} + \underbrace{\sqrt{d}\left(\lambda_{s,r,s}^{(T)} - \lambda^{(T)}\right)\boldsymbol{\mu}_j\|\boldsymbol{\mu}_j\|^{-2}}_{\mathcal{L}_2}
$$
$$
+ \underbrace{\sqrt{d}\sum_{j \neq s}\lambda_{s,r,j}^{(T)}\boldsymbol{\mu}_j\|\boldsymbol{\mu}_j\|^{-2}}_{\mathcal{L}_3} + \underbrace{\sqrt{d}\sum_{i \in I}\sigma_{s,r,i}^{(T)}\boldsymbol{\xi}_i\|\boldsymbol{\xi}_i\|^{-2}}_{\mathcal{L}_4}
$$

For $\mathcal{L}_1$ term, using the conclusion in Lemma D.10, we know that

$$\|\sqrt{d}\boldsymbol{w}_{s,r}^{(0)}\| \leq 2d\sigma_w \leq \epsilon = o(1).$$

For $\mathcal{L}_2$ term, using the conclusion in Lemma F.23, Lemma F.24, we know that

$$\left\|\sqrt{d}\left(\lambda_{s,r,s}^{(T)} - \lambda^{(T)}\right)\boldsymbol{\mu}_s\|\boldsymbol{\mu}_s\|^{-2}\right\| = |\lambda_{s,r,s}^{(T)} - \lambda^{(T)}| \leq 204k^2\epsilon\lambda^{(T)} \leq 816k^2\epsilon\ln(T+1) \leq 900k^{2.5}\epsilon = o(1).$$

For $\mathcal{L}_3$ term, by Lemma F.16 and triangle inequality, we know that

$$\left\|\sqrt{d}\sum_{j \neq s}\lambda_{s,r,j}^{(T)}\boldsymbol{\mu}_j\|\boldsymbol{\mu}_j\|^{-2}\right\| \leq k\epsilon = o(1).$$

For $\mathcal{L}_4$ term, by Property (1) in Lemma E.7, we have

$$\left\| \sqrt{d} \sum_{i \in I} \sigma_{s,r,i}^{(T)} \boldsymbol{\xi}_i \|\boldsymbol{\xi}_i\|^{-2} \right\|^2 = d \sum_{i \in I} \left( \sigma_{s,r,i}^{(T)} \right)^2 \|\boldsymbol{\xi}_i\|^{-2} + d \sum_{i_1 \neq i_2} \sigma_{s,r,i_1}^{(T)} \sigma_{s,r,i_2}^{(T)} \langle \boldsymbol{\xi}_{i_1}, \boldsymbol{\xi}_{i_2} \rangle \|\boldsymbol{\xi}_i\|^{-4}$$

$$\leq 2 \sum_{i \in I} \left( \sigma_{s,r,i_1}^{(T)} \right)^2 + \frac{2\Delta}{d} \sum_{i_1 \neq i_2} \sigma_{s,r,i_1}^{(T)} \sigma_{s,r,i_2}^{(T)}$$

$$\leq \frac{8k^2 \ln^2(T+1)}{n} + \frac{8k^2 \ln^2(T+1)\Delta}{d}$$

$$\leq \frac{8k^3}{n} + \frac{8k^3\Delta}{d} \leq 16k\epsilon = o(1).$$

Combining the above together, we know that

$$\sqrt{d} \left\| \boldsymbol{w}_{s,r}^{(T)} - \lambda^{(T)} \boldsymbol{\mu}_s \|\boldsymbol{\mu}_s\|^{-2} \right\| = o(1).$$

■

Then we prove that the the clean accuracy is nearly perfect(Property 1).

Assume $(\boldsymbol{x}, y)$ is randomly sampled from the data distribution $\mathcal{D}$. Without loss of generality, we assume that $\boldsymbol{x} = \boldsymbol{\mu}_j + \boldsymbol{\xi}, y = j$.

**Lemma F.26.** *Let $\boldsymbol{\xi} \sim \mathcal{N}(0, I_d)$. Then, with probability at least $1 - 2nd^{-\ln(d)/2}$, for all $s \in J, r \in [h]$ we have*

$$|\langle \boldsymbol{w}_{s,r}^{(T)}, \boldsymbol{\xi} \rangle| \leq \frac{\epsilon}{6}.$$

The proof of this lemma is the same as the proof of Lemma E.26.

Using the conclusion in Lemma F.13 and Lemma F.26, we know that for $s \in J, r \in [h], s \neq j$

$$\langle \boldsymbol{w}_{s,r}^{(T)}, \boldsymbol{x} \rangle = \langle \boldsymbol{w}_{s,r}^{(T)}, \boldsymbol{\mu}_j \rangle + \langle \boldsymbol{w}_{s,r}^{(T)}, \boldsymbol{\xi} \rangle \leq \lambda_{s,r,j}^{(T)} + \frac{\epsilon}{6} + \frac{\epsilon}{6} \leq \frac{\epsilon}{3},$$

$$\langle \boldsymbol{w}_{j,r}^{(T)}, \boldsymbol{x} \rangle = \langle \boldsymbol{w}_{j,r}^{(T)}, \boldsymbol{\mu}_j \rangle + \langle \boldsymbol{w}_{j,r}^{(T)}, \boldsymbol{\xi} \rangle \geq \lambda_{j,r,j}^{(T)} - \frac{\epsilon}{6} - \frac{\epsilon}{6} \geq \lambda_{j,r,j}^{(T)} - \frac{\epsilon}{3} \geq \frac{\epsilon}{3}.$$

Thus we know that $f_j(\boldsymbol{x}) = \max_{s \in J}\{f_s(\boldsymbol{x})\}$. So $\boldsymbol{F}_{\boldsymbol{\theta}^{(T)}}$ has standard perfect accuracy.

Finally, we prove that the corresponding binary classifier achieves optimal robustness(Property 3).

By Lemma F.25, we know that $\sqrt{d}\|\boldsymbol{w}_{s,r}^{(T)}\| \leq \lambda^{(T)} + o(1) \leq 2\lambda^{(T)}$.

Then for any perturbation $\boldsymbol{\rho}$ with $\boldsymbol{\rho} \leq \frac{\sqrt{d}}{10}$.

We know that

$$\langle \boldsymbol{w}_{j,r}^{(T)}, \boldsymbol{x} + \boldsymbol{\rho} \rangle = \langle \boldsymbol{w}_{j,r}^{(T)}, \boldsymbol{\mu}_i \rangle + \langle \boldsymbol{w}_{j,r}^{(T)}, \boldsymbol{\xi} \rangle + \langle \boldsymbol{w}_{j,r}^{(T)}, \boldsymbol{\rho} \rangle$$

$$\geq \lambda_{j,r,j}^{(T)} - \frac{\epsilon}{6} - \frac{\epsilon}{6} - \|\boldsymbol{w}_{j,r}^{(T)}\|\|\boldsymbol{\rho}\|$$

$$\geq \frac{3\lambda^{(T)}}{4}.$$

For $s \in J, s \neq j$, we know that

$$\langle \boldsymbol{w}_{s,r}^{(T)}, \boldsymbol{x} + \boldsymbol{\rho} \rangle = \langle \boldsymbol{w}_{s,r}^{(T)}, \boldsymbol{\mu}_i \rangle + \langle \boldsymbol{w}_{s,r}^{(T)}, \boldsymbol{\xi} \rangle + \langle \boldsymbol{w}_{s,r}^{(T)}, \boldsymbol{\rho} \rangle$$

$$\leq \lambda_{s,r,j}^{(T)} + \frac{\epsilon}{6} + \frac{\epsilon}{6} + \|\boldsymbol{w}_{s,r}^{(T)}\|\|\boldsymbol{\rho}\|$$

$$\leq \epsilon + \frac{\epsilon}{3} + \frac{\lambda^{(T)}}{5}$$

$$\leq \frac{3\lambda^{(T)}}{4} - \ln(k).$$

Thus we know that $f_j(\boldsymbol{x} + \boldsymbol{\rho}) \geq \dfrac{3\lambda^{(T)}}{4}$ and $f_s(\boldsymbol{x} + \boldsymbol{\rho}) \leq \dfrac{3\lambda^{(T)}}{4} - \ln(k)$.

let $G(\boldsymbol{x})$ denote the numerator of $F_{\boldsymbol{\theta}^{(T)}}^{binary}(\boldsymbol{x})$, where denominator is $\sum_{s \in J} e^{f_s(\boldsymbol{x})}$. We know

$$\mathrm{sgn}(F_{\boldsymbol{\theta}^{(T)}}^{binary}) = \mathrm{sgn}(G).$$

Thus we have

$$
\begin{aligned}
G(\boldsymbol{x} + \boldsymbol{\rho}) &= \sum_{j \in J_+} \exp\left(f_j(\boldsymbol{x} + \boldsymbol{\rho})\right) - \sum_{j \in J^-} \exp\left(f_j(\boldsymbol{x} + \boldsymbol{\rho})\right) \\
&\geq \exp\left(3\lambda^{(T)}/4\right) - \sum_{j \in J_-} \exp\left(3\lambda^{(T)}/4 - \ln(k)\right) \\
&\geq 0.
\end{aligned}
$$

That is to say $\mathrm{sgn}(G(\boldsymbol{x} + \boldsymbol{\rho})) = \mathrm{sgn}(G(\boldsymbol{x}))$, which means $F_{\boldsymbol{\theta}^{(T)}}^{binary}$ is robust under any perturbation with radius smaller than $\dfrac{\sqrt{d}}{10}$. $\qquad\square$

# G  TWO FEATURE LEARNING REGIMES: FEATURE AVERAGING AND FEATURE DECOUPLING

In this section, we present two distinct parameter regimes for our two-layer network learner: *feature averaging* and *feature decoupling*. The former means the weights associated with each neuron is a linear average of features, while the latter indicates that distinct features will be learned by separate neurons. Our construction is similar to that in Frei et al. (2024) and Min & Vidal (2024). We illustrate how a feature averaging solution leads to non-robustness, while a feature decoupling solution exists and is more robust (w.r.t. to a much larger robust radius).

## G.1  FEATURE-AVERAGING TWO-LAYER NEURAL NETWORK

Now, we begin by presenting the following example of a feature-averaging two-layer neural network, which is a more general version (including a bias term) than the one we mentioned in Definition 4.1.

**Feature-Averaging Two-Layer Neural Network.** Consider the following two-layer neural network with identical positive neurons and identical negative neurons, which can be simplified as (i.e., we merge identical neurons as one neuron):

$$f_{\boldsymbol{\theta}_{avg}}(\boldsymbol{x}) := \underbrace{\mathrm{ReLU}\left(\left\langle \sum_{j \in J_+} \boldsymbol{\mu}_j, \boldsymbol{x}\right\rangle + b_+\right)}_{\text{deals with all positive clusters}} - \underbrace{\mathrm{ReLU}\left(\left\langle \sum_{j \in J_-} \boldsymbol{\mu}_j, \boldsymbol{x}\right\rangle + b_-\right)}_{\text{deals with all negative clusters}},$$

where we choose weight $\boldsymbol{w}_{s,r} = \sum_{j \in J_s} \boldsymbol{\mu}_j$ for $s \in \{-1, +1\}, r \in [m]$ and bias $b_{s,r} = b_s$ for $s \in \{-1, +1\}, r \in [m]$.

Indeed, the feature-averaging network uses the first neuron to process all data within positive clusters, and it uses the second neuron to process all data within negative clusters. Thus, it can correctly classify clean data, which is shown as the following proposition.

**Theorem G.1.** *There exist values of $b_+$ and $b_-$ such that the feature-averaging network $f_{\boldsymbol{\theta}_{avg}}$ achieves $1 - o(1)$ standard accuracy over $\mathcal{D}$.*

*Proof of Theorem G.1.* Let $b_+ = b_- = 0$, and then we know, for data point $(\boldsymbol{x} = \alpha\boldsymbol{\mu}_i + \boldsymbol{\xi}, y) \sim \mathcal{D}$ within cluster $i$ (w.l.o.g. we assume cluster $i$ is a positive cluster), with high probability, it holds that

$$f_{\boldsymbol{\theta}_{avg}}(\boldsymbol{x}) \geq \langle \boldsymbol{\mu}_i, \alpha\boldsymbol{\mu}_i \rangle + \sum_{j \in J_+ \setminus \{i\}} \langle \boldsymbol{\mu}_j, \boldsymbol{\xi} \rangle - \sum_{j \in J_-} \langle \boldsymbol{\mu}_j, \boldsymbol{\xi} \rangle$$

$$\geq \Theta(d) - O(k\Delta) = \Theta(d) - O(k\sigma\sqrt{d}\ln(d)) \geq 0,$$

which implies that $f_{\boldsymbol{\theta}_{avg}}$ correctly classifies data $(\boldsymbol{x}, y)$ with high probability. $\square$

However, it fails to robustly classify perturbed data no matter what the bias term is, shown in the following theorem.

**Theorem G.2.** *For any values of $b_+$ and $b_-$ such that $f_{\boldsymbol{\theta}_{avg}}$ has $1 - o(1)$ standard accuracy, it holds that the feature-averaging network $f_{\boldsymbol{\theta}_{avg}}$ has zero $\delta-$robust accuracy for perturbation radius $\delta = \Omega(\sqrt{d/k})$.*

*Proof of Theorem G.2.* Indeed, we can choose the adversarial attack as $\boldsymbol{\rho} \propto -\sum_{j \in J_+} \boldsymbol{\mu}_j + \sum_{l \in J_-} \boldsymbol{\mu}_l$ and $\|\epsilon\| = \delta$. Then, for averaged features $\boldsymbol{w}_{s,r} = \sum_{j \in J_s} \boldsymbol{\mu}_j$, this perturbation can activate almost all of ReLU neurons, which w.h.p. leads a linearization over the perturbation $\boldsymbol{\rho}$

$$f_{\boldsymbol{\theta}_{avg}}(\boldsymbol{x} + \boldsymbol{\rho}) = f_{\boldsymbol{\theta}_{avg}}(\boldsymbol{x}) + \langle \nabla_{\boldsymbol{x}} f_{\boldsymbol{\theta}_{avg}}(\boldsymbol{x}), \boldsymbol{\rho} \rangle.$$

Since $f_{\boldsymbol{\theta}_{avg}}$ has $1 - o(1)$ standard accuracy, we know that the bias term satisfy that $b_+, b_- = O(d)$, which manifests that the classifier achieves a positive margin, i.e.

$$0 < y f_{\boldsymbol{\theta}_{avg}}(\boldsymbol{x}) \leq O(d),$$

w.h.p. over $(\boldsymbol{x}, y)$ sampled from $\mathcal{D}$.

Then, due to a large gradient norm over data input, i.e.

$$\|\nabla_{\boldsymbol{x}} f_{\boldsymbol{\theta}_{avg}}(\boldsymbol{x})\| = \|\sum_{j \in J_+} \boldsymbol{\mu}_j - \sum_{l \in J_-} \boldsymbol{\mu}_l\| = \Omega(\sqrt{kd}),$$

we derive that the feature-averaging network $f_{\boldsymbol{\theta}_{avg}}$ has zero $\delta$−robust accuracy for perturbation radius $\delta = \Omega(\sqrt{d/k})$. $\quad\square$

## G.2   ROBUST TWO-LAYER NEURAL NETWORK EXISTS

In this section, we show a robust two-layer network exists for $\mathcal{D}$, using a similar construction in Frei et al. (2024).

**Theorem G.3.** *There exists a two-layer network $f_{\boldsymbol{\theta}_{dec}}$ that is $\frac{\sqrt{d}}{3}$-robust for $\mathcal{D}$.*

*Proof of Theorem G.3.* The construction is similar to that in Frei et al. (2024). We define $f_{\boldsymbol{\theta}_{dec}}$ : $\mathbb{R}^d \to \mathbb{R}$ is a network that represents a positive constant times the following function:

$$f_{\boldsymbol{\theta}_{dec}}(\boldsymbol{x}) \propto \sum_{j \in J_+} \mathrm{ReLU}\left(\langle \boldsymbol{\mu}_j, \boldsymbol{x} \rangle - \frac{d}{2}\right) - \sum_{l \in J_-} \mathrm{ReLU}\left(\langle \boldsymbol{\mu}_l, \boldsymbol{x} \rangle - \frac{d}{2}\right),$$

In particular, we set a two-layer width-$k$ ReLU network with $\boldsymbol{w}_{s,j} = \mathbb{1}(j \in J_s)\boldsymbol{\mu}_j$, $b_{s,j} = -\mathbb{1}(j \in J_s)\frac{d}{2}$ for $s \in \{\pm 1\}$ and $j \in [k]$.

In this network, each neuron $\mathrm{ReLU}(\langle \boldsymbol{\mu}_j, \boldsymbol{x} \rangle - \frac{d}{2})$ (or $\mathrm{ReLU}(\langle \boldsymbol{\mu}_l, \boldsymbol{x} \rangle - \frac{d}{2})$) deals with one certain positive cluster $j$ (or negative cluster $l$), and we also apply the bias term to filter out intra/inter cluster noise. In this regime, for each data point $(\boldsymbol{x}, y)$ belonging to cluster $i$ (we assume cluster $i$ is a positive cluster and $y = 1$) and any perturbation $\boldsymbol{\rho}$ ($\|\boldsymbol{\rho}\| \leq \frac{\sqrt{d}}{3}$), we have the following linearization, w.h.p.

$$f_{\boldsymbol{\theta}_{dec}}(\boldsymbol{x} + \boldsymbol{\rho}) = \frac{1}{m}\langle \boldsymbol{\mu}_i, \boldsymbol{x} + \boldsymbol{\rho} \rangle.$$

Then, we know the network $f_{\boldsymbol{\theta}_{dec}}$ has $1 - o(1)$ $\delta$−robust accuracy for $\delta \leq \frac{\sqrt{d}}{3}$. $\quad\square$

Note that $f_{\boldsymbol{\theta}_{dec}}$ leverages *individual decoupled features*, which is a natural and robust solution to the binary classification on $\mathcal{D}$. In fact, one can easily verify that the robustness of $f_{\boldsymbol{\theta}_{dec}}$ is optimal up to a constant factor, as the distance between distinct cluster centers is $\Theta(\sqrt{d})$, i.e., $\|\boldsymbol{\mu}_i - \boldsymbol{\mu}_j\| = \Theta(\sqrt{d})$, for all $i \neq j$. However, as we show in our main result that gradient descent does not learn this feature-decoupled network directly from $\mathcal{D}$, and instead converges to a different solution that is $\Theta(\sqrt{k})$ times less robust.

## G.3   NON-ROBUST MULTI-CLASS NETWORK EXISTS

Similar to the feature-averaging binary-class network as that we mentioned in Definition 4.1, the non-robust multi-class network also exists, which is shown as the following proposition.

**Theorem G.4** (Restatement of Proposition 4.8). *Consider the following multi-class network $F_{\tilde{\boldsymbol{\theta}}}$: for all $j \in [k]$, the sub-network $f_j$ has only single neuron ($h = 1$) and is defined as $f_j(\boldsymbol{x}) = \mathrm{ReLU}\left(\langle \boldsymbol{\mu}_j + \sum_{l \in J_s} \boldsymbol{\mu}_l, \boldsymbol{x} \rangle\right)$, where cluster $j$ has binary label $s \in \{\pm 1\}$. With probability at least $1 - \exp(-\Omega(\log^2 d))$ over $\tilde{S}$, we have that $\mathcal{L}_{\mathrm{CE}}(\tilde{\boldsymbol{\theta}}) \leq \exp(-\Omega(d)) = o(1)$, where $\tilde{\boldsymbol{\theta}}$ denotes the weights of $F_{\tilde{\boldsymbol{\theta}}}$. Moreover, $\mathrm{Acc}_{\mathrm{clean}}^{\mathcal{D}}(F_{\tilde{\boldsymbol{\theta}}}^{\mathrm{binary}}) \geq 1 - \exp(-\Omega(\log^2 d))$, $\mathrm{Acc}_{\mathrm{robust}}^{\mathcal{D}}(F_{\tilde{\boldsymbol{\theta}}}^{\mathrm{binary}}; \Omega(\sqrt{d/k})) \leq \exp(-\Omega(\log^2 d))$.*

*Proof of Theorem G.4.* Consider data point $(\boldsymbol{x}, y)$ that is randomly sampled from the data distribution $\mathcal{D}$. Without loss of generality, we assume that $\boldsymbol{x} = \boldsymbol{\mu}_{j_0} + \boldsymbol{\xi}, j_0 \in J_+$. Reusing the argument of proof of Property (3) and (4) in Proposition D.6. We know that, with probability at least $1 - 2kd^{-\ln(d)/2}$, for all $j \in J$, we have

$$|\langle \boldsymbol{\mu}_j, \boldsymbol{\xi} \rangle| \leq \Delta.$$

First, we prove the network has perfect clean accuracy when the above properties hold. Indeed, we calculate the output value of each sub-network as follows.

$$f_{j_0}(\boldsymbol{x}) = \mathrm{ReLU}\left(\left\langle \boldsymbol{\mu}_{j_0} + \sum_{l \in J_+} \boldsymbol{\mu}_l, \boldsymbol{x} \right\rangle\right) \geq 2d - k\Delta.$$

For $j \in J_s, j \neq j_0$,

$$f_j(\boldsymbol{x}) = \mathrm{ReLU}\left(\left\langle \boldsymbol{\mu}_j + \sum_{l \in J_s} \boldsymbol{\mu}_l, \boldsymbol{x} \right\rangle\right) \leq d + k\Delta < 2d - k\Delta = f_{j_0}(\boldsymbol{x}).$$

Thus, we know that $\boldsymbol{F}_{\hat{\boldsymbol{\theta}}}(\boldsymbol{x}) = j_0$ with probability at least $1 - 2kd^{-\ln(d)/2}$.

Then, with probability at least $1 - \exp(-\Omega(\log^2 d))$ over $\tilde{S}$ sampled from $\mathcal{D}$, for all $i \in I$, we have

$$\begin{aligned}
-\log p_{\tilde{y}_i}(\boldsymbol{x}_i) = -\log \frac{\exp(f_{\tilde{y}_i}(\boldsymbol{x}_i))}{\sum_{j \in [k]} \exp(f_j(\boldsymbol{x}_i))} \\
\leq -\log(1 - \exp(-\Omega(d))) \\
\leq \exp(-\Omega(d)),
\end{aligned}$$

where the last inequality holds due to $\log(1 - z) \geq \Omega(z)$ for sufficiently small $z$. Therefore, we derive that

$$\mathcal{L}_{\mathrm{CE}}(\boldsymbol{\theta}_{\hat{\boldsymbol{\theta}}}) = \frac{1}{n} \sum_{i=1}^{n} \log p_{\tilde{y}_i}(\boldsymbol{x}_i) \leq \exp(-\Omega(d)) = o(1).$$

Finally, we prove that the network has at most $2kd^{-\ln(d)/2}$ robust test accuracy against perturbation radius $\delta = \Omega(\sqrt{d/k})$.

Consider perturbation $\boldsymbol{\rho} = \dfrac{3(1+c)}{k}\left(\sum_{l \in J_+} \boldsymbol{\mu}_l - \sum_{l \in J_-} \boldsymbol{\mu}_l\right)$.

For any $j \in J_+$, we know that

$$\begin{aligned}
\left\langle \boldsymbol{\mu}_j + \sum_{l \in J_+} \boldsymbol{\mu}_l, \boldsymbol{x} - \boldsymbol{\rho} \right\rangle = \left\langle \boldsymbol{\mu}_j + \sum_{l \in J_+} \boldsymbol{\mu}_l, \boldsymbol{\mu}_{j_0} + \boldsymbol{\xi} - \boldsymbol{\rho} \right\rangle \\
\leq 2d + (k+1)\Delta - \frac{k}{1+c}\frac{3(1+c)d}{k} \\
< 0.
\end{aligned}$$

For any $j \in J_-$, we know that

$$\begin{aligned}
\left\langle \boldsymbol{\mu}_j + \sum_{l \in J_-} \boldsymbol{\mu}_l, \boldsymbol{x} - \boldsymbol{\rho} \right\rangle = \left\langle \boldsymbol{\mu}_j + \sum_{l \in J_+} \boldsymbol{\mu}_l, \boldsymbol{\mu}_{j_0} + \boldsymbol{\xi} - \boldsymbol{\rho} \right\rangle \\
\geq d + (k+1)\Delta - \frac{k}{1+c} + \frac{3(1+c)d}{k} \\
> 0.
\end{aligned}$$

This is to say for any $j \in J_+$,

$$f_j(\boldsymbol{x} - \boldsymbol{\rho}) = \mathrm{ReLU}\left(\left\langle \boldsymbol{\mu}_j + \sum_{l \in J_+} \boldsymbol{\mu}_l, \boldsymbol{x} - \boldsymbol{\rho} \right\rangle\right) = 0.$$

For any $j \in J_-$,

$$f_j(\boldsymbol{x} - \boldsymbol{\rho}) = \mathrm{ReLU}\left(\left\langle \boldsymbol{\mu}_j + \sum_{l \in J_-} \boldsymbol{\mu}_l, \boldsymbol{x} - \boldsymbol{\rho} \right\rangle\right) > 0.$$

Thus, we obtain that $\mathrm{Acc}_{\mathrm{robust}}^{\mathcal{D}}(F_{\mathrm{FA}}^{\mathrm{binary}}; \delta) \leq \exp(-\Omega(\log^2 d))$

$$\square$$

## H ADDITIONAL REAL-WORLD EXPERIMENTS

### H.1 EMPIRICAL VERIFICATION OF FEATURE LEARNING PROCESS

Beyond verifying the alignment between our theoretical findings and the results of numerical simulation on the synthetic multi-cluster data setup, as described in Section 3, we also consider a more realistic setting where the multi-cluster structure of data naturally occurs.

**Transfer Learning Based on CLIP Model.** Here, we focus on a transfer learning setting, under which we utilize a pre-trained CLIP ViT-B-32 model (Radford et al., 2021) to obtain the image embedding for the CIFAR-10 dataset. We found that the embeddings of CIFAR-10 images approximately satisfy the multi-cluster structure, where the correlation between embeddings of images from the same class is significantly higher than that between embeddings of images from different classes. See experimental results in Figure 12, where we verify the orthogonality of the extracted features (i.e., image embeddings of CLIP model) by calculating the correlation between them.

**Binary Classification on CIFAR.** We create a 2-classification task from the CIFAR-10 dataset by merging the first 5 classes into one class and the other 5 classes into the other class. We apply two training strategies for training a two-layer neural network on this 2-classification task: one is to train directly on the image embedding labeled for 2-classification, and the other is to first train on the image embedding labeled for 10 classes and then convert it to 2-classification, where the two-layer network is as described in our theory (we fixe second layer as diagonal form, i.e., $f_j(\boldsymbol{z}) := \frac{1}{h} \sum_{r=1}^{h} \mathrm{ReLU}(\langle \boldsymbol{w}_{j,r}, \boldsymbol{z} \rangle), \forall j \in [2]$ or $[10]$, and $\boldsymbol{z}$ denotes the image embedding). We set the width of the first layer to be 1000 ($h = 10$) to ensure that the accuracy of the pre-trained model was not compromised. For 10-classification, we use $\boldsymbol{w}_j := \frac{1}{h} \sum_{r=1}^{h} \boldsymbol{w}_{j,r}$ as the equivalent weight of $f_j$. For the 500 positive weights and 500 negative weights in the binary classification network, we equally divide them into 5 positive classes and 5 negative classes to ensure a fair comparison, between the two figures which ensures that two models both have the same form $\boldsymbol{F} := (f_1, f_2, \ldots, f_{10}) \in \mathbb{R}^{10}$ and each sub-network $f_j$ corresponds to a weight vectors $\boldsymbol{w}_j$.

**Experiment Results.** See experiment results in Figure 13. Indeed, deep neural networks can be viewed as consisting of two parts: a feature extractor that is a mapping from the input space to the latent space, and a shallow classifier that predicts the classification results based on the extracted features (in the latent space). Feature averaging means that the shallow classifier mixes extracted features corresponding to different classes in the latent space. And our empirical results verify this.

### H.2 INFINITY-NORM CASE

Here, we aim to verify whether the model trained with fine-grained supervision information (i.e., 10-class labels) is more $\ell_\infty$-robust compared to the model trained with only binary (2-class) labels.

**Experiment Settings.** To ensure fairness in the comparison, we sum the logits corresponding to the 5 positive classes and subtracted the sum of the logits corresponding to the 5 negative classes from the 10-class model's output. This result is used as the binary classification output for the 10-class model. The robust accuracy is measured by using the standard PGD attacks (Madry et al., 2018) with different $\ell_\infty$-pertubation radius. We run experiments in the following datasets:

**Binary Classification on MNIST and CIFAR-10.** To further verify our theory in deep neural networks, on both MNIST and CIFAR-10 datasets, we train ResNet18 models from scratch with normal 10-classification labels and 2-classification labels (MNIST: parity-classification; CIFAR-10: binary-classification as that we mentioned in Section 5.1).

**Experiment Results.** The results are presented in Figure 14. With the perturbation radius increasing, we can see that the models trained with 10-class labels have higher robust test accuracy than those trained with 2-class labels in all datasets, which empirically shows that fine-grained supervision also improves $\ell_\infty$ robustness.

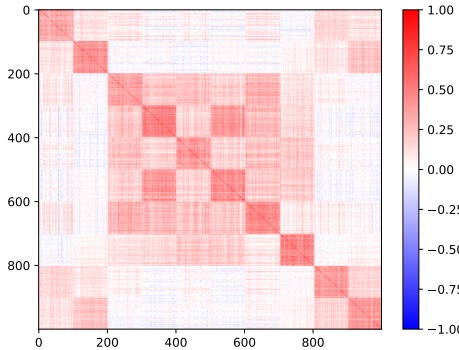

Figure 12: **Verifying Orthogonal Condition:** We plot the extracted feature correlation as a colormap, where each pixel represents some $\cos(\boldsymbol{z}_i, \boldsymbol{z}_j)$ between two extracted features $\boldsymbol{z}_i, \boldsymbol{z}_j$ of two data $\boldsymbol{x}_i, \boldsymbol{x}_j$ from CIFAR-10 training dataset (here, we sample 100 instances for each class).

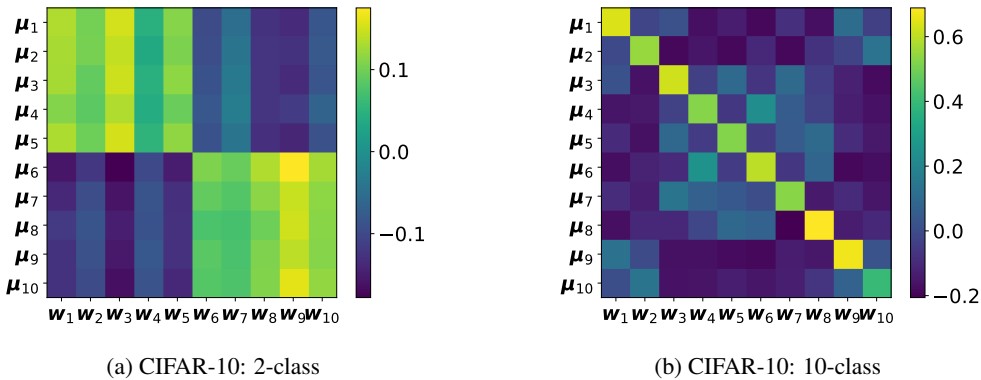

(a) CIFAR-10: 2-class  (b) CIFAR-10: 10-class

Figure 13: **Illustration of feature averaging and feature decoupling on CIFAR-10 dataset.** Figure (a) corresponds to models trained using 2-class labels, and Figure (b) corresponds to models trained using 10-class labels, respectively. Each element in the matrix, located at position $(i, j)$, represents the average cosine value of the angle between the feature vector $\boldsymbol{\mu_i}$ of the $i$-th feature and the equivalent weight vector $\boldsymbol{w}_j$ of the $f_j(\cdot)$.

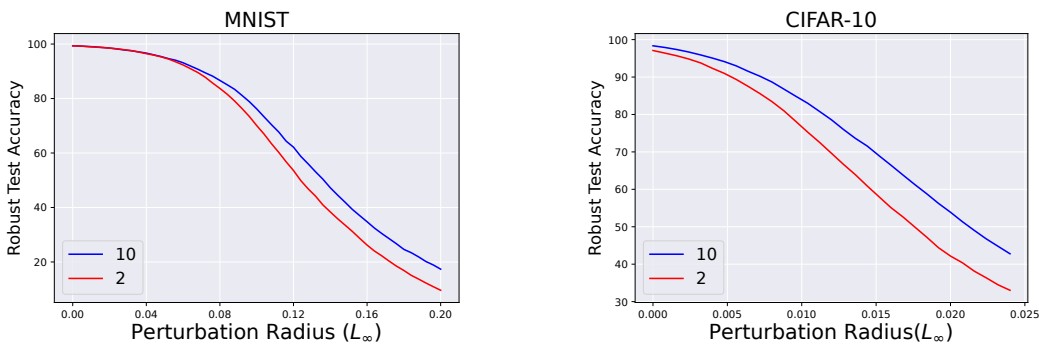

Figure 14: **Verifying robustness improvement in $\ell_\infty$ case:** We compare $\ell_\infty$ adversarial robustness between model trained by 2-class labels (red line) and model trained by 10-class labels (blue line) on MNIST (the left) and CIFAR-10 (the right).

