# OpenReview forum: "Feature Averaging: An Implicit Bias of Gradient Descent Leading to Non-Robustness in Neural Networks"
_ICLR.cc/2025/Conference — ICLR 2025 Poster_

### Official Review · Reviewer_MXLz · 2024-10-26

**Soundness:** 4
**Presentation:** 3
**Contribution:** 3
**Rating:** 8
**Confidence:** 5

**Summary:**

This paper theoretically studies the robustness of two layer neural networks under a specific data distribution that consists of $k$ clusters in $\mathbb{R}^d$ and the target consists of deciding on which group of clusters the input belongs to (binary classification). The authors show that these networks tend to learn an "averaging" solution (i.e., the first-layer weights encode the averages of the clusters), making the networks susceptible to $\Omega(\sqrt{d / k})$ perturbations of the input (despite the existence of networks capable of tolerating $O(\sqrt{d})$ perturbations). Motivated by this observation, they propose modifying the training process by using each example's cluster vector as the label, and they prove that this approach indeed yields optimally robust networks. Finally, the paper presents experiments with both synthetic data and image classification benchmarks that contextualize the theoretical findings.

**Strengths:**

This is a good paper, and it was a pleasure to read and review. I would like to highlight the following contributions:

- Important topic: The interplay between the implicit bias of optimization and robustness is, in my opinion, an important area of study.
- Finite time theoretical analysis of neural networks trained in the various settings: The paper rigorously analyzes the training trajectory of the networks. I enjoyed reading and learning about interesting techniques, such as the ones outlined in Section C.2. In general, the Appendix of the paper is exceptionally polished and the results are easy to follow and verify (to the extent that this was possible during the review process - the paper is 67pages long).
- Interesting suggestion on providing additional supervision during training for improved robustness: I found this observation interesting and the experimental results seem to suggest that these ideas may generalize beyond the toy distribution considered in the paper's analysis.

**Weaknesses:**

1) The main conceptual observation does not appear to be new. To my understanding, Frei et al. (2022) showed that neural networks trained with gradient descent on this distribution are not robust to $\ell_2$ perturbations of the input due to the implicit bias of optimization. In lines 153-171, the authors argue that their work differs from prior work because they do not explicitly analyze the properties of the KKT point of the max-margin problem and instead perform a finite-time analysis. Furthermore, they argue that convergence to the KKT points might be slow. However, we know from Liu & Li (2020) that these conditions are approximately satisfied as long as there is a positive margin. Thus, approximate results are conceptually consistent with the KKT analysis. Additionally, I believe that the fact that the network reaches an "averaging" solution can be deduced from the KKT equations in Frei et al. (2022)—see also (Q1).

2) The paper argues that "feature averaging is a principal factor contributing to non-robustness of neural networks" (line 014). However, it focuses on one type of perturbations. While the authors provide evidence in this direction for $\ell_2$ perturbations in a simple distribution (& experimental results in more complicated settings),  it is not clear that this holds more broadly. Note that the model’s solution is exceptionally contrived, as all neurons (within each group) converge to the same solution. See also (Q2, Q3).

2) While the paper is well-written, I found the discussion on prior work somewhat insufficient. I would appreciate a bit more clarity on what is novel about the first part of the contributions (i.e., the proof that networks reach feature-averaging solutions). Furthermore, in line 110, the authors mention the work of Min & Vidal (2024) and their conjecture for the first time. It would be more appropriate to introduce this work and its conjecture earlier in the introduction, making it easier for readers to appreciate the contribution.

4) A list of non-significant typos/grammatical errors I spotted:
- line 091: features-> feature
- line 107: exists-> exist
- line 167: analyze -> analysis
- line 261: neural -> neuron
- line 893: echos -> echoes
- line 1098: We -> we
- line 1100: margin -> margins
- lines 1307-1310 require rephrasing (they do not make sense),

**Questions:**

Q1: Can the fact that the network reaches an "averaging" solution be deduced from the KKT equations in Frei et al. (2022)?

Q2: The implicit bias of GD for this problem in 2-layer neural networks is undesirable for $\ell_2$ robustness. What about $\ell_\infty$ perturbations? It seems that this bias would help for $\ell_\infty$ perturbations, in contrast to the feature decoupling solution $NN_{FD}$. I would appreciate your thoughts on this.

Q3: Relatedly, would it be possible to measure robustness in the experiments on CIFAR10/MNIST against $\ell_\infty$ perturbations? If there is evidence for larger $\ell_\infty$ robustness when training with additional supervision, then this would make the authors' claims much stronger.

Q4: Figure 3: How do you measure perturbation radius? If, for example, you normalise MNIST images to be between 0 and 1, then a perturbation radius of 2.0 does not make any sense. Can you mention the ranges of the input for MNIST and CIFAR10?

Q5: In line 882, you mention that "In image classification tasks, Ilyas et al. (2019) visualized both robust and nonrobust features.". However, this is not true to the best of my knowledge. I believe there are 2 works that have provided such visualizations [1, 2].

[1]. G. Goh. A discussion of ’adversarial examples are not bugs, they are features’: Two examples of useful, non-robust features. Distill, 2019.
[2] Tsilivis, N. and Kempe, J. (2022). What can the neural tangent kernel tell us about adversarial robustness? Advances in Neural Information Processing Systems 35.

---

> ### Author Response · Authors · 2024-11-23
> **Response to Review MXLz (1/2)**
>
> We sincerely thank the reviewer for the positive support and valuable feedback! We greatly appreciate the insightful review, and the recognition of highlighting the significance of our contribution and the strength of our theory and experiments, as well as the clarity of our writing. We are very glad to address the questions and suggestions raised by the reviewer, which we believe will help further refine our work. Below are our responses to the questions and suggestions raised by the reviewer.
>
> >**[W]** Furthermore, in line 110, the authors mention the work of Min & Vidal (2024) and their conjecture for the first time. It would be more appropriate to introduce this work and its conjecture earlier in the introduction, making it easier for readers to appreciate the contribution.
>
> **[E]** We thank the reviewer for the suggestion. We have highlighted the work of [1] and their conjecture in line 70-77 of the revision.
>
> >**[Q1]** Can the fact that the network reaches an "averaging" solution be deduced from the KKT equations in Frei et al. (2022)?
>
> **[A1]** As we mentioned in the paper, the convergence of the network to KKT points is notably slow. Both Theorem A.9 in the appendix of [2] and Theorem 5 in [3] demonstrate that reaching KKT points requires exponentially many iterations, far exceeding the duration of practical training. In contrast, our analysis is applicable within polynomial time.
>
> In fact, we also attempted to derive our feature averaging conclusion solely from the KKT conditions presented by [4] but were unsuccessful. The primary challenges we encountered are as follows:
>  1. The difficulty in analyzing the specific magnitude of the dual variables in the KKT conditions.
>  1. We were unable to characterize the activation regions induced by the ReLU activations (like our Lemma C6) only from KKT conditions.
>  1. We suspect that there exists certain KKT solution that does not correspond to the uniform average of features (with nearly uniform weights). But this is hard to verify even empirically since reaching KKT points is quite slow.
>
> >**[Q2]** The implicit bias of GD for this problem in 2-layer neural networks is undesirable for  robustness. What about perturbations? It seems that this bias would help for perturbations, in contrast to the feature decoupling solution . I would appreciate your thoughts on this.
>
> **[A2]** If we consider the case where the features consist of k standard orthogonal basis ($e_1,e_2,\cdots,e_k$), the robustness under the $\ell_{\infty}$ norm also leads to similar conclusions. Specifically, we can simply prove that:
>
>  1. The network trained by 2-classification labels is non-robust to $\ell_{\infty}$-perturbation radius $\Omega(\sqrt(d)/k)$;
>
>  1. The network trained by fine-grained(k-classification) labels is robust to $\ell_{\infty}$-perturbation radius $O(\sqrt(d))$.
>
> For the general case, the theoretical analysis for $\ell_{\infty}$-​ perturbations may require additional assumptions, as the orthogonality condition holds less significance in this context. Therefore, we leave the study of the $\ell_{\infty}$ scenario for future work.

---

> ### Author Response · Authors · 2024-11-23
> **Response to Review MXLz (2/2)**
>
> >**[Q3]** Relatedly, would it be possible to measure robustness in the experiments on CIFAR10/MNIST against  perturbations? If there is evidence for larger robustness when training with additional supervision, then this would make the authors' claims much stronger.
>
> **[A3]** We conducted the experiments and presented the results in Appendix H in the revised version of our paper. The results show that fine-grained supervision also improves $\ell_{\infty}$ robustness.
>
> >**[Q4]** Figure 3: How do you measure perturbation radius? If, for example, you normalise MNIST images to be between 0 and 1, then a perturbation radius of 2.0 does not make any sense. Can you mention the ranges of the input for MNIST and CIFAR10?
>
> **[A4]** We follow the standard practice to normalize an image. (transforms.Normalize((0.4914, 0.4822, 0.4465), (0.2023, 0.1994, 0.2010)) for CIFAR-10 dataset and transforms.Normalize((0.5,), (0.5,)) for MNIST dataset)
> Given the high dimensionality of the data, the $\ell_2$ norm of the normalized input is significantly larger than 2. Therefore, an $l_2$ perturbation radius of 2.0 is reasonable.
>
> >**[Q5]** In line 882, you mention that "In image classification tasks, Ilyas et al. (2019) visualized both robust and nonrobust features.". However, this is not true to the best of my knowledge. I believe there are 2 works that have provided such visualizations [1, 2].
>
> **[A5]** The "visualization" referred to in this sentence corresponds to Figure 1 in [5].  We appreciate you pointing out these additional sources and have referenced them in the revised version of the paper.
>
> **Reference**
>
> [1] Min, H., & Vidal, R. (2024). Can Implicit Bias Imply Adversarial Robustness?. arXiv preprint arXiv:2405.15942.
>
> [2] Lyu, K., & Li, J. (2019). Gradient descent maximizes the margin of homogeneous neural networks. arXiv preprint arXiv:1906.05890.
>
> [3] Soudry, D., Hoffer, E., Nacson, M. S., Gunasekar, S., & Srebro, N. (2018). The implicit bias of gradient descent on separable data. Journal of Machine Learning Research, 19(70), 1-57.
>
> [4] Frei, S., Vardi, G., Bartlett, P., & Srebro, N. (2024). The double-edged sword of implicit bias: Generalization vs. robustness in relu networks. Advances in Neural Information Processing Systems, 36.
>
> [5] Ilyas, A., Santurkar, S., Tsipras, D., Engstrom, L., Tran, B., & Madry, A. (2019). Adversarial examples are not bugs, they are features. Advances in neural information processing systems, 32.

---

> > ### Comment · Reviewer_MXLz · 2024-11-23
> >
> > Thank you for the insightful answers. I appreciate the depth of your response and the changes made in the manuscript. I am certain that this paper is a valuable contribution. I updated my score accordingly.

---

> > > ### Author Response · Authors · 2024-11-23
> > > **Response to Review MXLz**
> > >
> > > Thank you very much for your kind review, and we are glad that you enjoyed our paper!

---

### Official Review · Reviewer_dDY4 · 2024-10-27

**Soundness:** 3
**Presentation:** 3
**Contribution:** 3
**Rating:** 8
**Confidence:** 3

**Summary:**

This paper studies the implicit bias of gradient descent, which the authors call 'Feature Averaging' by theoretically analyzing two layer ReLU network with only a learnable first layer, under a specific data distribution - binary data having multiple clusters with orthogonal cluster centers.
Under gradient descent, weights of neurons in the hidden layer converge to an average of the cluster centers, resulting in a non-robust network. Additionally, they also prove that with more fine-grained cluster information, the network can learn a robust solution for the binary classification problem.

**Strengths:**

* The theoretical analysis of the finite time gradient descent dynamics is thorough in the feature learning regime.
* This paper is an interesting contribution in understanding the robustness of neural networks from optimization perspective. The analysis also aided in proving a conjecture.
* The paper is well written and the presentation is clear.

**Weaknesses:**

* The assumptions on the data (orthonormal equinorm and multi-cluster) is restrictive.
* The proof is detailed and thorough for two layer ReLU with only one learnable hidden layer. Extending to deeper networks seems challenging.

**Questions:**

1. In the fine-grained supervision case of the multi-cluster data, I understand that the binary classification using the multi-class network results in a robust solution. But, I wonder if the learned network would still be non-robust for the multi-class classification.
2. It is not an ask to prove the following. I am only curious to understand and know the authors thought. The fine-grained supervision setting can also be seen as learning a more expressive/complex classifier (learn k classes to do binary classification). Then, can the authors comment on the adversarial min-max training? Would similar feature decoupling, ie the weight of each neuron is aligned with one cluster feature, happen naturally if one does adversarial training of the binary class data?

---

> ### Author Response · Authors · 2024-11-23
> **Response to Review dDY4**
>
> We sincerely thank the reviewer for the positive support and valuable feedback! We greatly appreciate the insightful review, and the recognition of highlighting the significance of our contribution and solidity of our theory, as well as the clarity of our writing. We are very glad to address the questions and suggestions raised by the reviewer, which we believe will help further refine our work. Below are our responses to the questions and suggestions raised by the reviewer.
>
>
> >**[W]** The assumptions on the data (orthonormal equinorm and multi-cluster) is restrictive.
>
> **[E]** To address the concern about the restrictive assumptions on the data, we have conducted several ablation studies to examine the effects of sample size, learning rate, initialization magnitude, signal-to-noise ratio, and orthogonal conditions. These studies demonstrate that our results can be extended to more relaxed settings.  In particular, regarding the equal-norm assumption, we have included an ablation study in Appendix B in the revised version of our paper.
>
> Besides, we believe that real-world data can exhibit a multi-cluster structure. To support this, we employed a CLIP model for feature extraction on the CIFAR-10 dataset. Our experiment result shows that the features extracted by CLIP exhibit a multi-cluster structure, which aligns with our data assumption.
> >**[Q1]** In the fine-grained supervision case of the multi-cluster data, I understand that the binary classification using the multi-class network results in a robust solution. But, I wonder if the learned network would still be non-robust for the multi-class classification.
>
> **[A1]**  The network remains robust for multi-class classification. Almost the same as the one for binary classification, we can prove that the network achieves O(\sqrt{d})-robustness, which is optimal for our setting.
>
> >**[Q2]** It is not an ask to prove the following. I am only curious to understand and know the authors thought. The fine-grained supervision setting can also be seen as learning a more expressive/complex classifier (learn k classes to do binary classification). Then, can the authors comment on the adversarial min-max training? Would similar feature decoupling, ie the weight of each neuron is aligned with one cluster feature, happen naturally if one does adversarial training of the binary class data?
>
> **[A2]**  Thank you for this excellent question, which provides an important perspective for understanding and extending our work. We conducted experiments on synthetic data (see the revised vision of our paper), and the results indicate that networks trained with adversarial training exhibit a tendency to learn feature decoupling solutions. However, the degree of decoupling is less pronounced compared to networks trained with fine-grained supervision. In terms of robustness, adversarial training does provide significant improvements over standard GD, but it remains slightly inferior to the robustness achieved through fine-grained supervision.

---

> > ### Comment · Reviewer_dDY4 · 2024-11-28
> >
> > Thank you for the response and clarifications. The additional experiments further show the strength of the theoretical results and so I retain my score and recommend acceptance of the paper.

---

> > > ### Author Response · Authors · 2024-11-28
> > > **Response to Review dDY4**
> > >
> > > Thank you very much for your kind review, and we are glad that you enjoyed our paper!

---

### Official Review · Reviewer_h25y · 2024-10-29

**Soundness:** 3
**Presentation:** 4
**Contribution:** 3
**Rating:** 6
**Confidence:** 2

**Summary:**

This paper analyzed the training dynamics of gradient descent on a simplified two-layer ReLU networks in a binary supervised classification task, where the feature vector is sample from a gaussian mixture model with equal probability for K>2 components and the label are masked as 2 categories. The main result in section 4 showed that, under some regulatory assumptions and choices of hyper-parameters, the learned network turns to a feature averaging network, which has low generalization error but is not robust to perturbation of dataset.

**Strengths:**

-This paper focuses on explaining the adversarial robustness of neural networks in classification problem, which of course is an important topic in ML/DL community.

-This paper is generally well-written and logically organized with clear notations, assumptions, and detailed references to prior works with comparisons

-This paper’s idea is intuitive: this paper characterizes the “approximate linearity” in networks under NTK assumptions as adversarial non-robustness in terms of Feature averaging.

**Weaknesses:**

- Too many strong and unrealistic assumptions are used. First, the number of neurons $M=2m$ assumed to be finite as the number of clusters of features in assumption 4.3. It is against the purpose of using the neural tangent kernel (NTK) theory to approximate infinite wide neural networks. Thus, the result in this paper is limited since “training deep neural network is a highly non-convex and over-parametrized optimization problem”. Second, as the solution to adversarial robustness, “fine-grained supervision” suggests including the original cluster labels for all data points, which is extremely unrealistic, because the response is simply assigned from the cluster labels. It is not surprising that there is an improvement in adverbial robustness as the exact latent information is provided. Feature decoupling is a simple result as the network can learn each cluster’s mean given the cluster labels.

- The evidence of feature averaging is shown Figure 2 as the strong correlations/cosine value between cluster’s mean and network weights, but the scale of correlations is not consistent. In particular, the results from CIFAR-10 are not convincing, where the average cosine value can be lower than 0.1.

**Questions:**

-Minor issues in notations: Big $O$ notation on line 185 is different from the others; $Theta$ notation is not defined.

-Why does strong correlations indicate the existence of feature averaging? Is there a rigorous proof? How do you define the threshold of having a strong correlation?

-Proposition 4.8 is used as an example of “adding more fine-grained supervision signals does not trivially lead to decoupled features and robustness”, but it does not follow the setup of your analysis, since $h=1$. How is this multi-class network obtained? Why is its parameter assumed to be the combination of $mu_s$?

---

> ### Author Response · Authors · 2024-11-23
> **Response to Review h25y (1/2)**
>
> We sincerely thank the reviewer for the positive feedback! We greatly appreciate the recognition of the novelty and significance of our contribution to the topic of adversarial robustness in the deep learning community, as well as the positive remarks on the clarity of our writing. We are very glad to address the questions and suggestions raised by the reviewer, which we believe will help further refine our work. Below are our responses to the questions and suggestions raised by the reviewer.
>
> >**[W1-1]**  First, the number of neurons  assumed to be finite as the number of clusters of features in assumption 4.3. It is against the purpose of using the neural tangent kernel (NTK) theory to approximate infinite wide neural networks. Thus, the result in this paper is limited since “training deep neural network is a highly non-convex and over-parametrized optimization problem”.
>
> **[E1-1]** We would like to clarify that we did not use the NTK assumption or NTK analysis in this paper. Unlike the infinite width (excessive over-parameterization) requirement of the NTK regime, we focus on a more realistic setup with moderately finite width (mild over-parameterization) and small initialization. This setup leads the neural network to the feature learning regime (i.e., more discriminative features are learnt during the training process, unlike in the NKT (also called lazy training) regime where the features barely change during training. Within this feature learning regime, we aim to understand the type of features (averaged features or decoupled features) learned by the GD algorithm based on the multi-cluster data distribution.
>
> Indeed, due to the non-linearity of the ReLU activation function, training two-layer ReLU networks with gradient descent method under this feature learning regime becomes a highly non-convex optimization problem. To address this challenge, we develop novel techniques for analyzing the full training dynamics by leveraging margin auto-balance (Lemma C.4 and Lemma E.7), which enables us to bound the loss derivative ratio (equation (6)) and analyze ReLU activation regions (Lemma C.6). See our detailed proof sketch in Appendix C.
>
> >**[W1-2]**  Second, as the solution to adversarial robustness, “fine-grained supervision” suggests including the original cluster labels for all data points, which is extremely unrealistic, because the response is simply assigned from the cluster labels. It is not surprising that there is an improvement in adverbial robustness as the exact latent information is provided. Feature decoupling is a simple result as the network can learn each cluster’s mean given the cluster labels.
>
> **[E1-2]** While it might intuitively appear that annotation refined to the cluster level could ensure that a multi-class neural network eventually learns a feature-decoupling solution, we emphasize that convergence to robust networks is nontrivial and requires implicit bias. In particular,  in Proposition 4.8, we provide a specific construction of multi-class network that achieves perfect clean accuracy but is non-robust. Indeed, due to non-linearity of ReLU activation function, it is also highly non-trivial to rigorously prove that the multi-class network converges to feature-decoupling solution, where we apply a margin auto-balance technique (Lemma F.10) similar to that in proof of feature-averaging regime to show it.
>
> >**[W2]** The evidence of feature averaging is shown Figure 2 as the strong correlations/cosine value between cluster’s mean and network weights, but the scale of correlations is not consistent. In particular, the results from CIFAR-10 are not convincing, where the average cosine value can be lower than 0.1.
>
> **[E2]**  In real-world datasets, such as the CIFAR-10 dataset, the data distribution is considerably more complex than the theoretical distributions assumed in our analysis. Consequently, even though we theoretically prove that the features exhibit exact uniform averaging, networks trained on real-world datasets often show feature averaging with moderately non-uniform coefficients.

---

> ### Author Response · Authors · 2024-11-23
> **Response to Review h25y (2/2)**
>
> >**[Q1]** Minor issues in notations: Big notation on line 185 is different from the others;  notation is not defined.
>
> **[A1]** Thank the reviewer for pointing out this typo. We have fixed it in the revision of our paper.
>
> >**[Q2]** Why does strong correlations indicate the existence of feature averaging? Is there a rigorous proof? How do you define the threshold of having a strong correlation?
>
> **[A2]** We would like to clarify that we consider the following decomposition: $\boldsymbol{w} \approx c_1 \boldsymbol{\mu}_1 + c_2 \boldsymbol{\mu}_2 + \cdots + c_k \boldsymbol{\mu}_k$, where $\boldsymbol{w}$ represents the weight vector, and $\boldsymbol{\mu}_i$ represents the features of the $i$-th cluster (the first half corresponds to positive clusters, and the remaining part corresponds to negative clusters). Assuming that the cluster features are nearly orthogonal, the feature averaging regime means that:
>
> 1) For positive neurons, $c_1 \approx c_2 \approx \cdots \approx c_{k/2} \gg c_{k/2 + 1} \approx \cdots \approx c_k \approx 0$.
>
> 2) For negative neurons, $c_{k/2 + 1} \approx c_{k/2 + 2} \approx \cdots \approx c_k \gg c_1 \approx \cdots \approx c_{k/2} \approx 0$.
>
> In our experiments, the coefficients $c_i$ are approximately computed based on the correlation between the weight and the cluster features. The term "strong correlation" here is a relative concept, indicating that the correlation between weight vectors and features of the corresponding classes is significantly stronger than the correlation with features of the other classes.
>
> >**[Q3]** Proposition 4.8 is used as an example of “adding more fine-grained supervision signals does not trivially lead to decoupled features and robustness”, but it does not follow the setup of your analysis, since . How is this multi-class network obtained? Why is its parameter assumed to be the combination of ?
>
> **[A3]**  We would like to clarify that we manually construct the multi-class network to show the existence of a multi-class network that achieves perfect clean accuracy but is non-robust, which shows that adding more fine-grained supervision signals does not trivially lead to decoupled features and robustness. We design this network, and it aligns with the setup. If $h=1$ represents an important distinction, then it is possible to replicate $h$ identical neurons.

---

> > ### Comment · Reviewer_h25y · 2024-11-26
> >
> > Thanks for the explanation. The paper is well-written and presents its ideas clearly. However, the results on CIFAR-10 is not entirely convincing for demonstrating that feature averaging exists more generally. Thus, I decided to keep my score.

---

> > > ### Author Response · Authors · 2024-11-27
> > > **Response to Review h25y**
> > >
> > > Thank you very much for your kind review, and we are glad that you enjoyed our paper!

---

### Official Review · Reviewer_DJdJ · 2024-11-02

**Soundness:** 3
**Presentation:** 3
**Contribution:** 3
**Rating:** 6
**Confidence:** 4

**Summary:**

This paper provides a theoretical analysis of the relation between feature dependency and adversarial robustness. The authors studied a two-layer ReLU network on the simulated data and derived that the gradient descent tends to learn averaged features instead of isolated/robust ones. The results were proved thoroughly. Then, the paper provides a test on the derived theory on real-world datasets, showing that by incorporating more precise training labels, the model can learn isolated features rather than mixed ones, therefore achieving better adversarial robustness.

**Strengths:**

The paper gives very thorough theoretical study of the problem. The hypothesis, assumptions and derivation are reasonable and sound. The results matched with the intuition well.

**Weaknesses:**

The theory developed in this paper is phenomenological. It can be further improved if systematic and quantifying criteria can be built to design an optimization method that can help improve the robustness.

**Questions:**

1. The feature average is clearly defined in two-layer networks - the linear input combinations. However, in the deeper layer, such a definition does not seem reasonable. How do we analyse features in this case? Can the authors provide one or more examples, conceptually or mathematically, about how features are averaged in deeper neural networks?

2. In image tasks, reasonable and robust features are often local and isolated. However, there are other scenarios where the features are required to have global or long-term dependency. Such as time series, and natural language. In those cases, it seems like the real interpretable and robust features are some "mixing" of the original data. Therefore:
 a) How should we define the feature mixing in those cases?
 b) Would the theory developed in the paper be effective in those circumstances? If so, how to measure or validate? If not, should there be possible modification can improve the adaptability of the theory?

3. There are researches that have out that, the neural network tends to learn low-frequency features (which are nonlocal and varied smoothly) more effectively than high-frequency features. Some connect such behaviour to the generalization capacity of over-parameterized neural networks. Do you think there is any relation between this theory and the ones developed in this paper? I want to hear about the comments of the authors.
I list a few papers for reference:
Xu, Zhi-Qin John, et al. "Frequency principle: Fourier analysis sheds light on deep neural networks." arXiv preprint arXiv:1901.06523 (2019).
Xu, Zhi-Qin John, Yaoyu Zhang, and Tao Luo. "Overview frequency principle/spectral bias in deep learning." Communications on Applied Mathematics and Computation (2024): 1-38.

---

> ### Author Response · Authors · 2024-11-23
> **Response to Review DJdJ (1/2)**
>
> We sincerely thank the reviewer for the encouraging and insightful feedback, and for highlighting the strength of our theoretical contributions and the clarity of our writing. We are very glad to address the questions and suggestions raised by the reviewer, which we believe will help further refine our work. Below are our responses to the questions and suggestions raised by the reviewer.
> >**[Q1]** The feature average is clearly defined in two-layer networks - the linear input combinations. However, in the deeper layer, such a definition does not seem reasonable. How do we analyze features in this case?
>
> **[A1]**  We thank the reviewer for the insightful suggestion about feature learning in deep networks. Indeed, we also believe a similar feature-averaging bias may emerge in more complex scenarios beyond our theoretical settings. Conceptually, although there may not be a predefined set of features in deeper networks, gradient descent could still have a tendency to combine many localized, semantically meaningful (and thus, more robust ([1][2])) features into a single discriminative but non-robust feature.
>
> Specifically, deep neural networks can be viewed as consisting of two parts: a feature extractor that is a mapping from the input space to the latent space, and a shallow classifier that predicts the classification results based on the extracted features (in the latent space). Feature averaging means that the shallow classifier mixes extracted features corresponding to different classes in the latent space. To verify this, we provide a CLIP-based experiment in the revised version of the paper, where we first apply the CLIP model to extract latent representation of the input image and then we train a two-layer network based on the latent representation. See the CLIP experiment results in Appendix H in the revised version of the paper for details.
>
> Indeed, if all layers (including the feature extractor layers) are trainable, the feature averaging bias may further influence the feature extracted in a more complicated way, and we leave the study for deeper networks as a challenging future direction.
>
> Finally, we would like to mention that similar “averaging” or “superposition” effects also observed in ref [3]. In particular, the authors observe that in CLIP models each neuron may encodes multiple, often unrelated concepts (e.g., ships and cars), and they leveraged such effect to construct  "semantic" adversarial examples. See Figure 1 in [3] for details.

---

> ### Author Response · Authors · 2024-11-23
> **Response to Review DJdJ (2/2)**
>
> >**[Q2]** In image tasks, reasonable and robust features are often local and isolated. However, there are other scenarios where the features are required to have global or long-term dependency. Such as time series, and natural language. In those cases, it seems like the real interpretable and robust features are some "mixing" of the original data.
>
> **[A2]** Thank the reviewer for giving the valuable suggestion, and we clarify that our study focuses on adversarial examples in the computer vision field. While extending to natural language processing or time series is very important, it falls beyond the scope of this work.
>
> Indeed, as the reviewer pointed out, many interpretable features in NLP have global or long-term dependency. However, only local attacks can be very harmful for NLP models.  For instance, the Greedy Coordinate Gradient (GCG) [4] attack only appends a suffix to the original prompt (which is a form of local attack) and it is sufficient to induce jailbreaks in large language models (LLMs). So far, the GCG attack continues to be challenging to defend against effectively. Hence, there seems to be a different mechanism for the non-robustness in NLP models and we leave it as an exciting and challenging direction for future research.
>
> >**[Q3]** There are researches that have out that, the neural network tends to learn low-frequency features (which are nonlocal and varied smoothly) more effectively than high-frequency features. Some connect such behaviour to the generalization capacity of over-parameterized neural networks. Do you think there is any relation between this theory and the ones developed in this paper?
>
> **[A3]**  We thank the reviewer for pointing out the related works. The papers mentioned by the reviewer refers to the phenomena that deep neural networks generally learn lower-frequency features first, and then the higher-frequency ones. This can be seen as a particular form of simplicity bias. The feature averaging bias studied in this paper is also a form of simplicity bias: under our theoretical setup, the simplicity refers to the linear combination of cluster features, and it is closely related to the approximate linearity of the decision boundary, as discussed in Appendix A. Hence, both studies assert that neural networks tend to favor simplicity during the initial stages of training, sharing a similar underlying spirit.
>
> **Reference**
>
> [1] Ilyas, A., Santurkar, S., Tsipras, D., Engstrom, L., Tran, B. and Madry, A. (2019). Adversarial examples are not bugs, they are features. Advances in neural information processing systems, 32.
>
> [2] Tsilivis, N. and Kempe, J. (2022). What can the neural tangent kernel tell us about adversarial robustness? Advances in Neural Information Processing Systems, 35 18116–18130.
>
> [3] Gandelsman, Y., Efros, A. A., & Steinhardt, J. (2024). Interpreting the Second-Order Effects of Neurons in CLIP. arXiv preprint arXiv:2406.04341.
>
> [4] Zou, A., Wang, Z., Kolter, J. Z., & Fredrikson, M. (2023). Universal and transferable adversarial attacks on aligned language models. arXiv preprint arXiv:2307.15043.

---

> > ### Comment · Reviewer_DJdJ · 2024-11-25
> > **Respond to authors**
> >
> > Thanks for your detailed feedback. It answers a lot of questions. Based on your feedback, this work belongs to a specific theoretical study of adversarial robustness in Imaging. It gives pretty good proof and experiments on simplified models, while not showing promising extensibility. But I still think this is a good work. I will keep my scores considering this.

---

> > > ### Author Response · Authors · 2024-11-25
> > > **Response to Review DJdJ**
> > >
> > > Thank you very much for your kind review, and we are glad that you enjoyed our paper!

---

### Author Response · Authors · 2024-11-23
**Global Response to Reviewers**

We sincerely thank all reviewers for the positive support and valuable feedback! We greatly appreciate the insightful review, and the recognition of highlighting the significance of our contribution and the strength of our theory and experiments, as well as the clarity of our writing. We thank their feedback and they have helped us to improve our manuscript. We have provided a revised manuscript (text highlighted in blue), and we explain our revision in this global response:

 1. In Section 1, we highlight the work of [1] and their conjecture in line 70-77 of the revision.

 1. In Appendix A, we have incorporated additional discussion about [2][3][4] based on the insightful suggestions and questions raised by the reviewers.

 1. In Appendix B,  we have incorporated an ablation study on equinorm and robustness-related performance tests of adversarial training algorithms on synthetic datasets.

 1. We have incorporated Appendix H. In the first part of this appendix, we present additional experiments based on the CLIP model, and in the second part we focus on experiments on real-world datasets (MNIST and CIFAR10) under the $\ell_{\infty}$-robustness case.

Thank all reviewers for their insightful and valuable reviews again.

**Reference**

[1] Min, H., & Vidal, R. (2024). Can Implicit Bias Imply Adversarial Robustness?. arXiv preprint arXiv:2405.15942.

[2] Zhi-Qin John Xu, Yaoyu Zhang, Tao Luo, Yanyang Xiao, and Zheng Ma. Frequency principle:
Fourier analysis sheds light on deep neural networks. arXiv preprint arXiv:1901.06523, 2019.

[3] Zhi-Qin John Xu, Yaoyu Zhang, and Tao Luo. Overview frequency principle/spectral bias in deep
learning. Communications on Applied Mathematics and Computation, pp. 1–38, 2024.

[4] Yossi Gandelsman, Alexei A Efros, and Jacob Steinhardt. Interpreting the second-order effects of
neurons in clip. arXiv preprint arXiv:2406.04341, 2024.

---

### Meta-Review · Area_Chair_4CWQ · 2024-12-19

**Metareview:**

This paper studies the convergence of gradient descent for a two-layer ReLU network on a data distribution characterized by localized and separated clusters, with binary labels. In particular, the authors show that GD provides a solution, for appropriate assumptions, where the network weights converge to the mean of the label-wise clusters, in a phenomenon they term "feature averaging". While this observation (even in this synthetic setting) is not new (including being presented as a conjecture in a recent work) this work constitutes the first proof of this behavior. Consequently, as pointed out by other works, this solution is susceptible to adversarial perturbations, and thus non-robust. In turn, the paper shows that extra granular supervision alleviates this issue, converging to a solution with improved robustness (and no feature-averaging). The theory is supported by experimental results.

The contribution of this paper is important, formally showing that GD can have an implicit bias that conveys a lack of adversarial robustness. Some weaknesses were noted by the reviewers, concerning mostly clarification on the connection to other closely related work, and on the technicalities of their setup.

In all, the reviewers have a consensus that this is a worthy publication, and I concur.

**Additional Comments On Reviewer Discussion:**

All reviewers had questions seeking for clarification from the authors. The most important points raised concerned on the specific connections to the works by [Vardi et al. (2022)], [Frei et al. (2024)] and [Min & Vidal (2024)], all of which share similarities in the studied problem, assumptions, and techniques. The authors have provided additional clarifications, showing that their work is indeed different from those works that study convergence by studying the KKT conditions of a max-margin classifier (as done in the former works), and their analysis resolves the conjecture by [Min & Vidal (2024)] which studies a very similar problem to the one considered here. These discussions led to clarifications in the paper that has improved its quality.

---

### Decision · Program_Chairs · 2025-01-22

Accept (Poster)